# Auricular malformations are driven by copy number variations in a hierarchical enhancer cluster and a dominant enhancer recapitulates human pathogenesis

Xiaopeng Xu[1,2,3,17], Qi Chen[4,17], Qingpei Huang[1,17], Timothy C. Cox [5,17], Hao Zhu[2], Jintian Hu[4], Xi Han[1], Ziqiu Meng[2], Bingqing Wang[4], Zhiying Liao[1], Wenxin Xu[1,6], Baichuan Xiao [2], Ruirui Lang[2], Jiqiang Liu[2], Jian Huang [2], Xiaokai Tang[2], Jinmo Wang[2], Qiang Li[7], Ting Liu[8], Qingguo Zhang[4], Stylianos E. Antonarakis [9,10,11], Jiao Zhang[12] ✉, Xiaoying Fan [1,3,13,14,15] ✉, Huisheng Liu [1,3,15] ✉ & Yong-Biao Zhang [2,16] ✉

Enhancers, through the combinatorial action of transcription factors (TFs), dictate both the spatial specificity and the levels of gene expression, and their aberrations can result in diseases. While a *HMX1* downstream enhancer is associated with ear malformations, the mechanisms underlying bilateral constricted ear (BCE) remain unclear. Here, we identify a copy number variation (CNV) containing three enhancers—collectively termed the positional identity hierarchical enhancer cluster (PI-HEC)—that drives BCE by coordinately regulating *HMX1* expression. Each enhancer exhibits distinct activity-location-structure features, and the dominant enhancer with high mobility group (HMG)-box combined with Coordinator and homeodomain TF motifs modulating its activity and specificity, respectively. Mouse models demonstrate that neural crest-derived fibroblasts with aberrant *Hmx1* expression in the basal pinna, along with ectopic distal pinna expression, disrupt outer ear development, affecting cartilage, muscle, and epidermis. Our findings elucidate mammalian ear morphogenesis and underscore the complexity of synergistic regulation among enhancers and between enhancers and transcription factors.

Investigating the intricate morphological changes during mammalian embryogenesis remains a pivotal focus in developmental biology, a discipline that has recently seen significant progress[1]. The mammalian ear, a complex structure originating from cranial neural crest cells (CNCCs) and all three germ layers, exemplifies the complexity of developmental biology. It showcases a unique interplay of developmental processes and evolutionary adaptation[2,3]. The outer ear,

essential for capturing sound waves and implicated in conductive hearing loss when dysfunctional, is less explored than its middle and inner counterparts[3]. This part, comprising the tragus and helix, is not only important for auditory function but also demonstrates considerable morphological diversity throughout evolution as well as in many types of diseases[4–7]. Unraveling the molecular mechanisms underlying this diversity is an ongoing field of investigation.

A full list of affiliations appears at the end of the paper. ✉e-mail: joycezhang1978@hotmail.com; fan_xiaoying@gzlab.ac.cn; liu_huisheng@gzlab.ac.cn; zhangyongbiao@gmail.com

Bilateral constricted ear (BCE) often presents as an autosomal dominant trait and primarily manifests as curling of the upper portion of the external ear, affecting structures such as the helix, scapha, and antihelix. Comparative genetic studies across species, including sheep, cows, mice, and rats, have identified a correlation between BCE-like phenotypes and a non-coding regulatory region adjacent to the *Hmx1* gene that contains an evolutionarily conserved region (ECR)[8,9]. *Hmx1* (Homeobox gene H6-like 1), alternatively designated as *Nkx5-3*, represents a fundamental member of the homeodomain-containing Nkx5 transcription factor family. Two other highly similar genes in this family include *Hmx2* and *Hmx3*, both of which show expression patterns in the central nervous system (CNS) and inner ear[10]. The deficiency of either *Hmx2* or *Hmx3* results in vestibular system malformation and abnormal inner ear development[11]. *Hmx1* is mainly expressed in the second branchial arch (ventral-caudal region), eye, trigeminal cranial nerve, and dorsal root ganglia, and its deficiency leads to the occurrence of Oculo-Auricular syndrome[12]. In humans, phenotypic presentations similar to those in concha-type microtia have been linked to genomic duplications encompassing the ECR near *HMX1*[13]. Notably, in rats, a 5.7-kb deletion that includes this ECR significantly diminishes *Hmx1* expression, leading to a dumbo ear phenotype, highlighting a critical regulatory role[14]. Furthermore, in human concha-type microtia, several genomic duplications exist that extend beyond the ECR, suggesting a complex and multifaceted genetic mechanism underlying outer ear development. The spectrum of pathogenic mutations involving the ECR, including both duplications and deletions, emphasizes the complex interplay of gene expression in determining ear morphology.

Elucidating the function of cis-regulatory elements in specific cellular and temporal settings is key to unraveling the complex mechanisms of ear development and regulatory variations that influence the external ear phenotype[15,16]. Recent research underscores CNCCs as vital in craniofacial development, including the outer ear[17–19]. CNCC positional identity is determined by a complex set of signaling factors that govern a distinct hierarchy of gene expression profiles. For example, positional identity along the anterior-posterior axis is influenced by bivalent gene expression, such as *Hoxa2* in the second pharyngeal arch and *Hand2* in the mandibular arch[20]. Along the proximo-distal axis, gene expression of *Dlx* homeobox family members, for example, governs patterning within the mandibular arch. Patterning along all axes is crucial for proper craniofacial morphogenesis[21]. This highlights the importance of precise CNCC positional identity[22]. For instance, ectopic *Hoxa2* expression in the pharyngeal arch 1 (PA1) results in mirror-image pinna duplication[7], while the loss of *Dlx5* and *Dlx6* in the mandibular arch results in the transformation of the lower jaw into a mirror image of the upper jaw components[23]. Single-cell RNA sequencing (scRNA-seq) studies of the mandibular arch further revealed a wide range of heterogeneous cellular domains (intra-arch identity), with disturbances leading to specific mandible and tooth defects[24]. Yet, the exact regulatory mechanisms of spatial gene expression in the craniofacial region, including *HMX1*, and within individual arches, are not fully understood. This knowledge gap limits our understanding of the genetic and epigenetic dynamics in human craniofacial morphogenesis.

This investigation presents a comprehensive analysis of human pedigrees with BCE, employing gene chips and next-generation sequencing to pinpoint a pathogenic locus, designated as the BCE locus on chromosome 4. This locus encompasses an enhancer cluster, comprising three synergistically acting enhancers identified in human craniofacial tissues and in vitro-derived human CNCCs (hCNCCs). Transgenic LacZ mouse experiments revealed that the spatial specificity of the *Hmx1* gene expression in pharyngeal arch 2 (PA2) during ear development is governed by a hierarchical interplay among these enhancers, forming what we term the Positional Identity Hierarchical Enhancer Cluster (PI-HEC). In-depth examination of the dominant enhancer in PI-HEC unraveled a complex network of transcription factors (TFs) modulating enhancer specificity and function. Gene-editing in mouse models elucidated the link between pinna development and altered *Hmx1* expression, exposing a dysregulated molecular network affecting pinna structures such as cartilage, muscle, and epidermis. Our findings significantly advance the understanding of the genetics and regulatory mechanisms behind human outer ear development, providing molecular insights into the etiology of pinna development and the presentation of BCE.

## Results

### Copy number duplications detected in BCE

We collected seven Chinese families presenting with the BCE phenotype, with an inheritance pattern consistent with complete autosomal dominance (Fig. 1a). Kinship analysis confirmed that these families are unrelated (PI-HAT < 0.1 for all sequenced founders). Affected individuals in these families exhibited the characteristic BCE phenotype (Fig. 1b), defined by malformations of the external ear, including a reduced helix size and the absence of the triangular fossa, scaphoid fossa, and antihelix, resulting in a distinctive shell-like appearance. We explored the genetic basis of BCE using classical linkage analysis with GeneChip genotyping. Linkage analysis on pedigrees 1, 3, and 6 yielded a significant combined parametric LOD score of 7.8 (Supplementary Fig. 1a), with individual scores of 2.7, 1.8, and 3.3 respectively (Supplementary Fig. 1b). This led to the mapping of the susceptibility locus to a 10.2 cM region on chromosome 4 (Supplementary Fig. 1b).

For fine mapping of this locus, target-capture sequencing was performed on 32 individuals across the three pedigrees. This detailed analysis revealed copy number duplications (copy number = 3) co-segregating in all sequenced patients from the three pedigrees (Supplementary Fig. 1c). The specific genomic ranges of these duplications were chr4:8691119-8795306, chr4:8679801-8750713, and chr4:8668379-8739905 (hg19) for each respective pedigree. Further investigations using chip-based methods and whole-genome sequencing in the remaining four pedigrees confirmed that probands in each of these pedigrees also carried copy number duplications in this non-coding region (Fig. 1c). Ultimately, our findings converged on a minimum genomic region of overlap of chr4:8691119-8728565 (termed the BCE core locus), consistently duplicated across all seven BCE pedigrees. Hi-C data from human eye cell line[25], revealed the TAD structure around the BCE core locus, suggesting that *HMX1* and *CPZ* are potential target genes (Fig. 1c).

### The BCE core locus, containing a cluster of enhancers, directly interacts with the *HMX1* promoter

Previous studies have identified a conserved enhancer within the BCE core locus that participates in the regulation of *Hmx1* gene expression across different mammals[8,14]. However, understanding of the full range of cis-regulatory elements at this locus, including their context-dependent characteristics and their role in controlling pinna development, remains incomplete. Given the spatiotemporal specificity of cis-regulatory elements, it is imperative to study them within embryonic contexts that are closely associated with pinna development. Studies in mice indicated that CNCCs are important for pinna organogenesis[7], and both murine studies[14] and single-cell sequencing data of human embryos[26] also indicated that *HMX1* is predominantly expressed in the craniofacial mesenchyme, including the PA1/2 and PA3/4 regions (Supplementary Fig. 2a). These data underscore the necessity of concentrating our research on CNCCs and craniofacial mesenchyme to thoroughly investigate cis-regulatory elements within the BCE core locus and their roles in pinna development (Supplementary Fig. 2a).

To delineate the cis-regulatory elements at the BCE core locus, we interrogated publicly available ChIP-seq data from both in vivo

embryonic human head tissues (CS13-CS17)[27] and in vitro-derived hCNCCs[28]. This approach revealed a cluster of potential enhancers (termed EC1, EC2, and EC3), the homologous sequence corresponding

to EC1 in mice contains the 594 bp ECR as mentioned in the introduction, within the BCE core locus across these datasets: EC1 and EC2 are both active in two datasets, EC3 was defined as an active enhancer

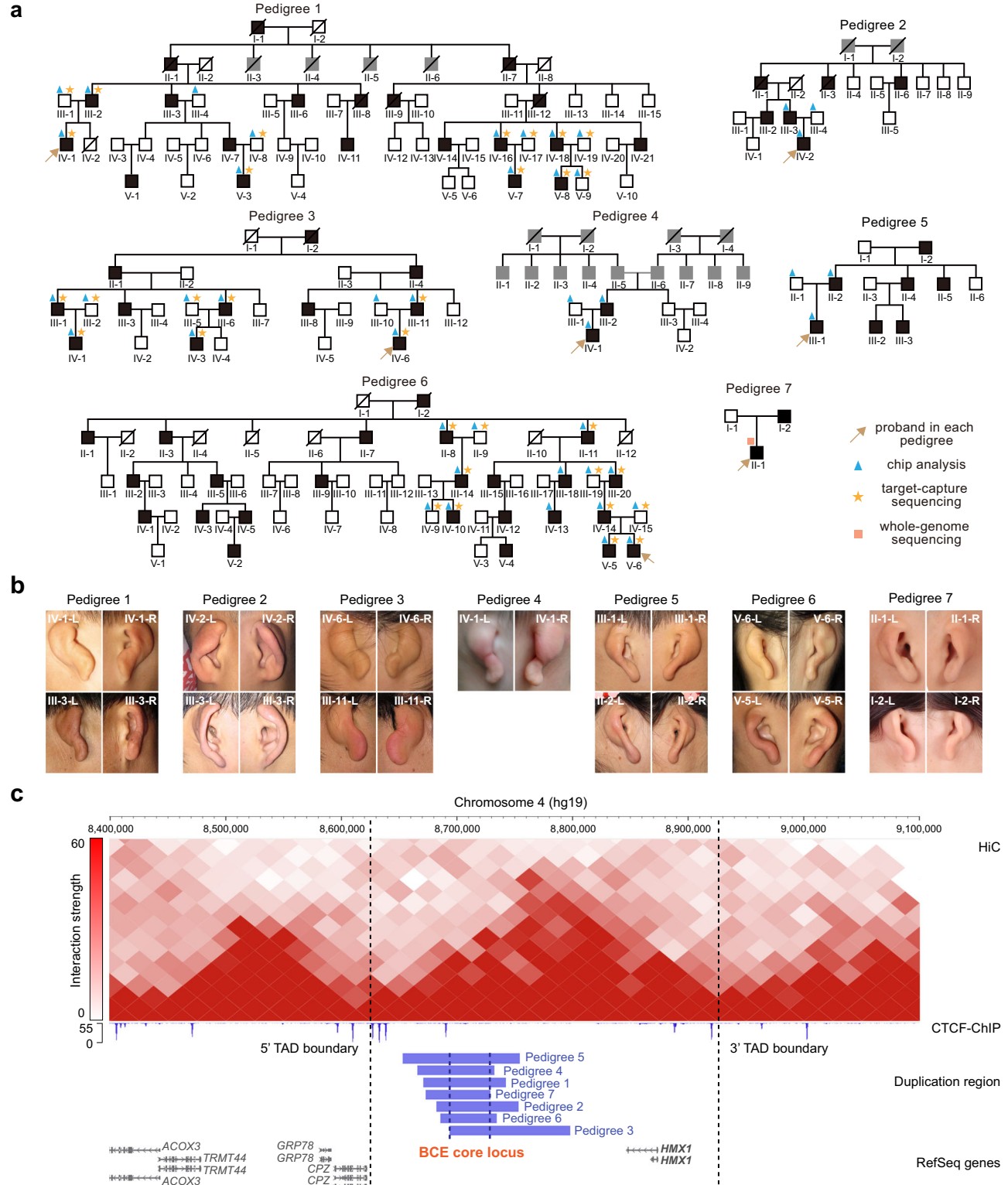

**Fig. 1 | Duplication of non-coding region downstream of *HMX1* as a pathogenic factor in Bilateral Constricted Ears (BCEs). a** Pedigrees of seven families with BCEs. Each individual's pedigree identity is indicated below their symbol. Black shading indicates affected individuals, while gray shading denotes unknown phenotype. **b** Representative images of affected ears from the proband patients and their families, in each pedigree. **c** Copy number duplication in each family, located in the intergenic region downstream of the *HMX1* gene, within the same TAD. The top panel displays Hi-C data (25 kb resolution) from human eye tissue[25], with the black dotted line marking the TAD boundary. In the middle panel, the CTCF signal (late hCNCCs), a marker frequently observed at TAD boundaries, is depicted across this region. The lower panel specifies the duplication regions for each family, highlighting the minimum overlapping genomic area (chr4: 8691119-8728565, termed BCE core locus).

in vivo but possesses weak histone signals in vitro (Fig. 2a). Consistent with findings in the human datasets, analogous sequences in publicly available mouse CNCCs (mCNCCs) data[20] were also identified as an

enhancer cluster, notably in PA2 (Supplementary Fig. 2b). Comparative analyses employing Promoter Capture Hi-C (PCHi-C) revealed that the enhancer cluster region actively interacts with the *HMX1* promoter in

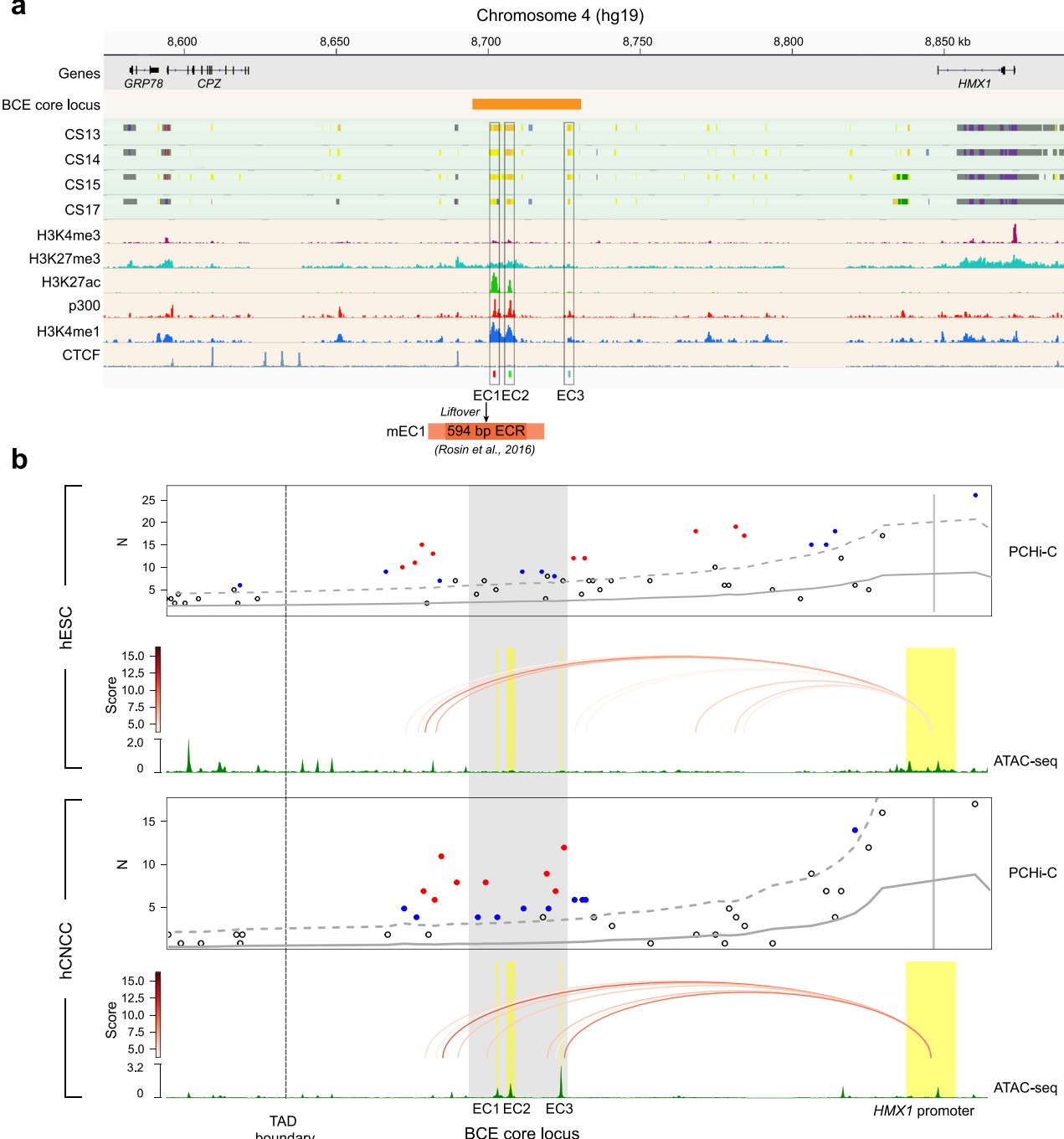

**Fig. 2 | Identification and characterization of candidate enhancers at the BCE core locus. a** Epigenomic profiling pinpoints enhancer elements at the BCE core locus. Integrated epigenomic analysis of the BCE core locus, combining in vivo data from human embryo craniofacial tissue at stages CS13-CS17 (including ear progenitor regions, upper part)[27] with in vitro data from hESCs-derived human Cranial Neural Crest Cells (hCNCCs, lower part)[31]. CNV region shared across seven pedigrees is outlined by the golden rectangle. The chromatin state is represented through color coding (gray, Repressed PolyComb; purple, Bivalent Promoter; light yellow, Weak Enhancer; dark yellow, Active Enhancer; green, Weak Transcription; red, Promoter Upstream TSS), as defined on the Cotney lab craniofacial epigenome website (https://cotney.research.uchc.edu/craniofacial/). The epigenetic marks H3K4me3, H3K27me3, H3K27ac, and the presence of the transcriptional coactivator p300, in addition to H3K4me1 and CTCF signals in hCNCCs, are displayed.

Candidate enhancers EC1, EC2, and EC3 are identified based on both datasets and are indicated by black rectangles. Previously reported mouse 594 bp ECR region resides in the mEC1 homologous sequence. **b** Differential chromatin interactions link CE enhancers to *HMX1* promoter. Promoter Capture Hi-C (PC-HiC) analysis demonstrates the differential physical interactions between the CE locus and the *HMX1* promoter in hESCs compared to late hCNCCs. For each cell line, the upper panel shows interaction profile (Gray full and dashed lines show expected counts and the upper bound of the 95% confidence intervals, respectively; Significant interactions with score ≥5 are shown in red, and sub-threshold interactions with 3 ≤ score < 5 are shown in blue); The middle panel shows called loops and the line color denotes the interaction strength; The bottom panel shows ATAC-seq track. The TAD boundary identified in Fig. 1c is depicted. ATAC-seq data and PC-HiC data for hESCs are sourced from GSE145327[28] and GSE86821[85], respectively.

hCNCCs, but not in human Embryonic Stem Cells (hESCs) (Fig. 2b and Supplementary Data 1). This observation suggests that the activity of the enhancer cluster is temporally regulated and cell-type specific, favoring *HMX1* and excluding interaction with the adjacent gene *CPZ*. Publicly available mouse PC-HiC data[29] further revealed that the BCE core locus interacts with the *Hmx1* promoter in PA2 tissue of E10.5 and pinna structures of E12.5 and E14.5 (Supplementary Fig. 2c). The collective evidence suggests a conserved role of the enhancer cluster in modulating *HMX1* expression in CNCCs, necessitating further clarification of how enhancers within this complex coordinate the spatiotemporal pattern of *HMX1* expression.

### Different intrinsic activities of the three enhancers at the BCE core locus during hCNCCs differentiation

To investigate the spatiotemporal modulating patterns of the individual candidate enhancers hEC1, hEC2, and hEC3 within the enhancer cluster, we further conducted a detailed analysis of epigenetic datasets of two developmental stages of in vitro derived hCNCCs[28,30]: early (D11), representing cells migrating out of the neural rosettes, and late (P4), representing post-migratory mesenchymal cells (Fig. 3a). Active enhancer markers, P300 and H3K27ac, revealed distinct profiles: hEC1

displayed significant activity in late hCNCCs; hEC2 was divided into two regions, hEC2.1 and hEC2.2, based on P300 signals; hEC2.1 showed strong H3K27ac activity at both stages, whereas hEC2.2 exhibited weaker H3K27ac activity, indicating a less active state at both stages; hEC3, despite showing strong accessibility in late hCNCCs and interaction with the *HMX1* promoter (Fig. 2b), lacked H3K27ac signals (Fig. 3a).

To evaluate the capacity of the candidate enhancers to drive reporter gene expression, we adapted a well-established in-vitro differentiation system[31], further incorporating retinoic acid (RA) at the late hCNCCs stage to induce transformation into PA (pharyngeal arch)-like hCNCCs characteristic (Fig. 3b and Supplementary Fig. 4a). *HMX1* gene expression initially rose approximately 4-fold at the early hCNCCs stage, and subsequently surged to about 140-fold at the late hCNCCs stage and 110-fold at the PA-like hCNCCs stage (Fig. 3c). In luciferase assays, none of the enhancers showed activity in H9 cells, consistent with the inaccessibility of these enhancers in hESCs (Fig. 3d and Fig. 2b). However, hEC2.1 and hEC2 become active in early hCNCCs, while hEC1 also initiates activation, albeit with very weak activity (Supplementary Fig. 4b). Furthermore, in late and PA-like hCNCCs, EC1 was a strong transcriptional activator (Fig. 3e and

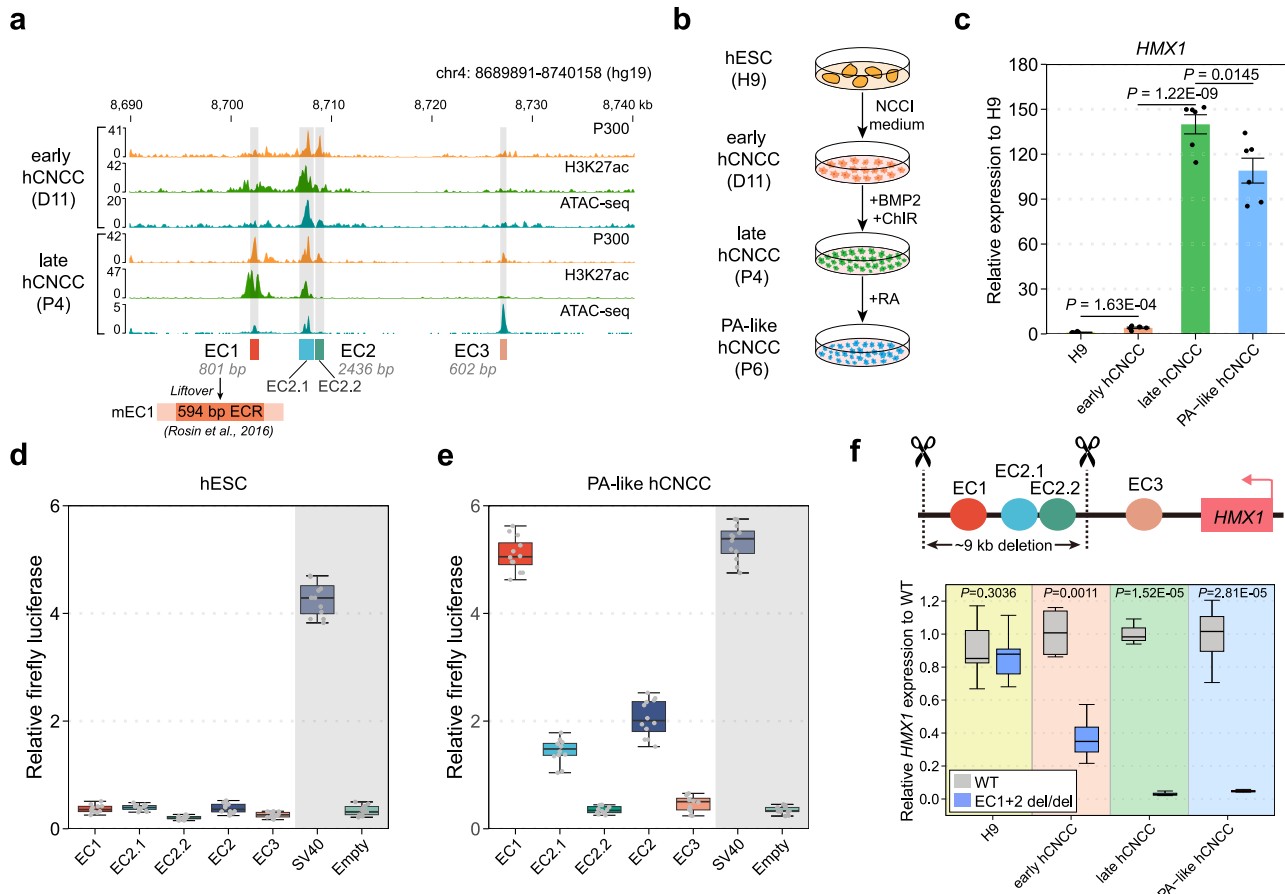

**Fig. 3 | The expression level of *HMX1* is correlated with the activities of the enhancers in the BCE core locus. a** Epigenetic profiling at the BCE core locus in early (D11) and late hCNCCs (P4). Previously reported mouse 594 bp ECR region resides in the mEC1 homologous sequence. **b** Schematic of hESCs differentiation to PA-like hCNCCs: The process involves transitioning H9 cell line hESCs to early hCNCCs using NCC induction (NCCI) medium, followed by BMP2/ChIR treatment to reach late hCNCCs stage, and finally achieving PA-like hCNCCs with retinoic acid (RA) introduction. **c** *HMX1* expression analysis across four cell states, presented as mean ± S.E.M. (*n* = 6). *P*-values were calculated using unpaired Student's *t*-test (two-tailed). **d, e** Luciferase assays evaluating candidate enhancers at the BCE core locus

in hESC (**d**), and PA-like hCNCC (**e**), including SV40 enhancer (positive control) and empty vector (negative control). Results from three independent experiments, each with four technical replicates (*n* = 12), are shown. The box plot displays the median (center line), 25th-75th percentiles (box bounds), and whiskers extending to 1.5×interquartile range from each quartile (10th-90th percentiles shown as reference), with outliers plotted individually. **f** CRISPR/Cas9 strategy to delete two active enhancers (hEC1 and hEC2, -9 kb), and gene expression comparison across four cell states in WT and knockout cell lines (*n* = 3). *P*-values were calculated using unpaired Student's *t*-test (two-tailed). The boxplot definition is same as **d, e**. Source data are provided as a Source Data file.

Supplementary Fig. 4c), while hEC2.1 and hEC2 demonstrated weaker activity than hEC1 (Fig. 3e and Supplementary Fig. 4c). Notably, hEC2.2 and hEC3 remained inactive in all tested contexts, consistent with their weak H3K27ac signals (Fig. 3d and Fig. 3e). These findings suggest that hEC1 is a strong enhancer, hEC2 is a weak enhancer and hEC3 is primed for activation but it is not currently active in this cellular context or it may serve as a structural element to facilitate the contact between the BCE core locus enhancer cluster and the *HMX1* promoter[32].

In a further experiment, we created knockout (KO) hESC lines with simultaneous deletions of hEC1 and hEC2 (Fig. 3f, upper panel, and Supplementary Fig. 5a). Both wild-type and KO hESC lines exhibited similar differentiation capacities into hCNCCs, and the expression levels of two marker genes, *SOX9* and *TWIST1*, were similar in both lines (Supplementary Fig. 5b, c). In the differentiation of hESCs to PA-like cells, *HMX1* expression remained similar between wild-type and KO cells at the H9 cell stage, but diverged noticeably at the early hCNCCs stage, becoming more pronounced in the late and PA-like

hCNCCs stages (Fig. 3f, bottom panel). In summary, our results establish the relationship between *HMX1* expression and enhancer activities in the BCE core locus in different stages of hCNCCs development.

## *Hmx1* gene patterning along pinna development is related to epistatic interactions of the three enhancers in the BCE core locus

We employed whole-mount in-situ hybridization (WISH, E9.5) and a newly generated transgenic Hmx1-P2A-EGFP mouse reporter line (E11.5 and E14.5) to elucidate the spatiotemporal dynamics of *Hmx1* expression during embryonic ear development (Fig. 4a and Supplementary Fig. 5d). The GFP reporter mouse we constructed accurately recapitulates the spatiotemporal expression pattern of the *Hmx1* gene in-vivo (Supplementary Fig. 5e, f, g). Specifically, we focused on three developmental stages: embryonic day 9.5 (E9.5), representing early CNCCs development stage in PAs; E11.5, reflecting CNCC-

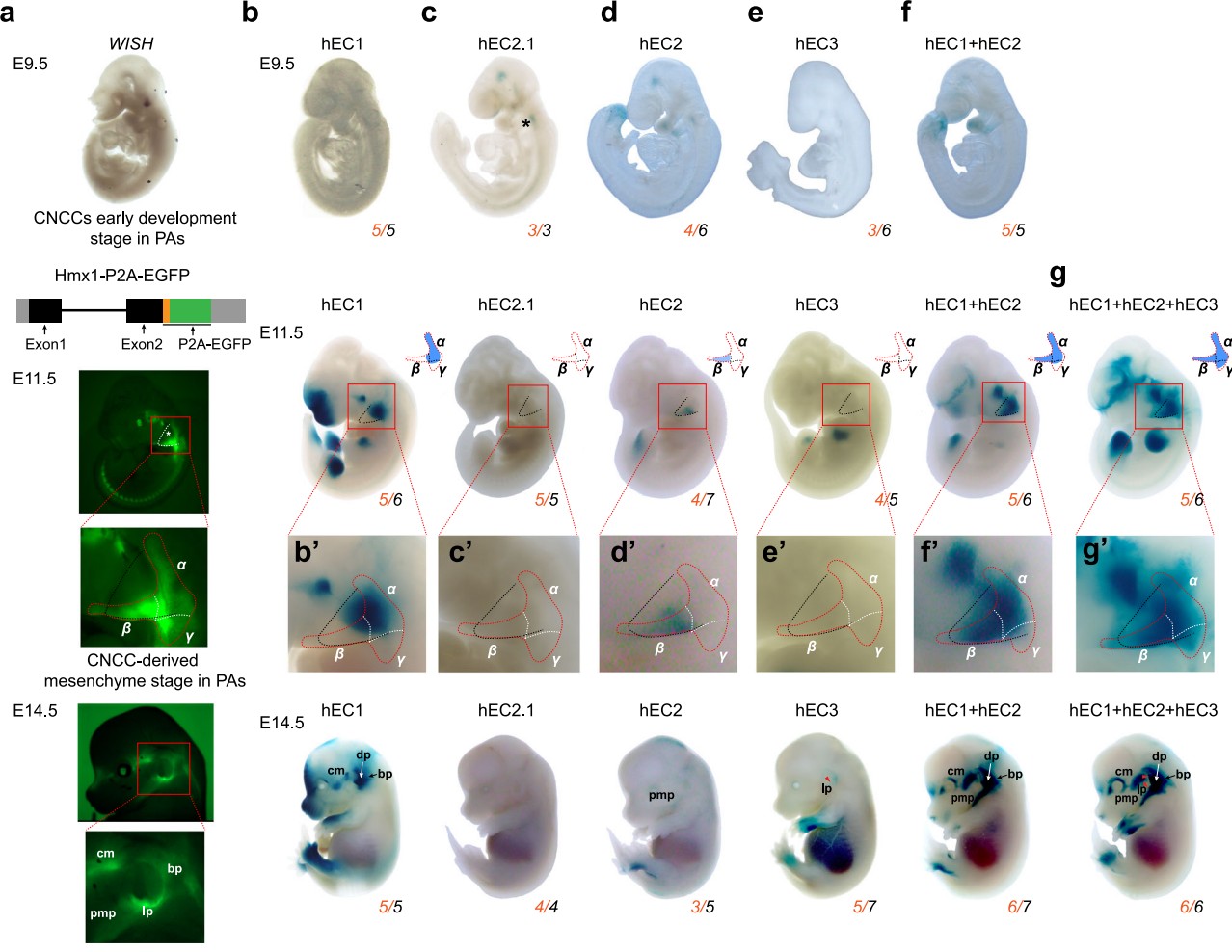

**Fig. 4 | Spatiotemporal regulation of *HMX1* expression by multipartite enhancers in-vivo. a** Spatiotemporal *Hmx1* expression from E9.5 to E14.5: WISH image and fluorescent images of Hmx1-P2A-EGFP transgenic mouse reporter line (Supplementary Fig. 4j); Schematic of the knock-in strategy of Hmx1-P2A-EGFP is presented. The expression of *Hmx1*, predominantly in the PA2 zone (region outlined by the white dashed box) and pinna region, was magnified for detailed observation, and the spatial location of *Hmx1* expression at E11.5 is manually divided into α (proximal), β (distal), γ (caudal to PA2) positions according to corresponding PA2 zones. cm, craniofacial mesenchyme; bp, basal pinna; lp, lower part of pinna; dp, distal pinna; pmp, the proximal region of the mandibular prominence. **b–g** LacZ assays during pinna development for various enhancers of hEC1 (**b**),

hEC2.1 (**c**), hEC2 (**d**), hEC3 (**e**), hEC1 + hEC2 (**f**), and hEC1 + hEC2 + hEC3 (**g**). The orange number in the bottom right corner of each embryo indicates the count of positive LacZ staining embryos showing a similar expression pattern, with the black number denoting the total count of transgenic embryos examined (Supplementary Fig. 4d–i). The black star on the E9.5 embryo of hEC2.1 marks weak activity at the basal of PA2. PA2 zones were magnified (**b'–g'**) for detailed comparison with endogenous *Hmx1* expression in the α (proximal), β (distal), γ (caudal to PA2). Two clearly visible staining positions in the upper distal part of the pinna and the lower pinna part adjacent to the proximal mandibular region on hEC1 + hEC2 + hEC3 E14.5 embryo, compared with other enhancer constructs, were marked as red triangles.

derived mesenchyme development stage in PAs; and E14.5, corresponding to the initial stages of pinna development (Fig. 4a and Supplementary Fig. 5f, g). We observed the initiation of *Hmx1* expression at E9.5 in the superior region of PA1. By E11.5, this expression expanded to the eye, craniofacial mesenchyme, part of PA2, and the dorsal root ganglia (DRG). By E14.5, *Hmx1* predominantly localized to the lower part of the outer ear (lp), the basal pinna zone (bp), craniofacial mesenchyme (cm), and the proximal region of the mandibular prominence (pmp) (Fig. 4a).

In the LacZ assays (a site-directed transgenic approach (enSERT) was adopted) assessing BCE enhancers' activity during mouse development, distinct patterns emerged for each enhancer. For hEC1, no activity was observed at E9.5. However, by E11.5, strong signals appeared in the proximal region of PA2, expanding to nearly the entire pinna, excluding the upper distal region, by E14.5 (Fig. 4b and Supplementary Fig. 6a). In contrast, hEC2.1 displayed only weak activity in the basal PA2 region and head at E9.5, with no detectable activity at later time points (Fig. 4c and Supplementary Fig. 6b). For hEC2, weak activity was noted in the distal PA1 and PA2 regions, head, and tail at E9.5. This activity persisted weakly in the distal PA2 region and hindlimbs at E11.5, and in the proximal region of the mandibular prominence, head, and hindlimbs at E14.5 (Fig. 4d and Supplementary Fig. 6c). For hEC3, no activity was observed at E9.5. However, it demonstrated two distinct patterns at E11.5: four transgenic embryos showed activity in the limbs (4/5 of samples), while one exhibited partial activity in proximal PA2 and a small area below PA2 (Fig. 4e and Supplementary Fig. 6d). At E14.5, the activity in the limbs persists, while relatively stable but weaker signals are observed in the ear region, specifically in the lower part of the pinna (lp). This pattern aligns with the activity detected in PA2 in only one embryo at E11.5 (Supplementary Fig. 6d). These results were somewhat similar to those using cellular models; thus, hEC2 appears to be active in early CNCC development in PAs, whereas hEC1 activity is initiated in CNCC-derived mesenchyme stage in PAs, with both maintaining activity through craniofacial development. In contrast, hEC3 activity is not active in early/late hCNCCs model, but appears to be active in later pinna development.

To enable a detailed comparison between the spatial specificity of the *Hmx1* gene expression at E11.5 and the LacZ staining patterns of three enhancers, we divided the *Hmx1* expression around the PA2 zone into three distinct regions: α (proximal PA2 region), β (distal PA2 region), and γ (caudal to PA2) (Figs. 4a, E11.5, zoomed region). Our analysis elucidated the hierarchical relationship among the three enhancers: hEC1 exerts a predominant influence on regions α and, to a lesser extent, γ; hEC2 is primarily associated with region β; and hEC3 appears to affect region γ (only one positive embryo), albeit with an inconsistent pattern (Fig. 4b', d', e'). This led us to hypothesize that hEC1, while driving significant transcriptional activity, may not fully convey positional information. In contrast, hEC2 and hEC3, despite their weaker transcriptional activities, could provide the additional positional information required for precise *Hmx1* localization within the PA2 zone (intra-arch identity). To investigate this particular gene patterning regulation, we constructed two additional vectors: hEC1 + hEC2 and hEC1 + hEC2 + hEC3, aiming to study the epistatic interactions among the enhancer cluster in the BCE core locus (Fig. 4f, g). Notably, the pattern exhibited by the hEC1 + hEC2 combination was distinct, showing reduced overall activity at E11.5 and a more defined spatial activity in the frontonasal prominence (FNP) and limbs. Crucially, it modulated the spatial position within PA2 (Fig. 4f' and Supplementary Fig. 6e), encompassing most *Hmx1* expression regions, including α, β, and part of γ (Fig. 4a). We found that hEC1 and hEC2 partially overlap in the upper right corner of the beta region (Fig. 4b' and 4d'), which suggests that the two may have some redundant characteristics. However, they are not interchangeable, as the spatial activity domain of hEC1 + hEC2 is larger than that of hEC1 or hEC2 alone

(Fig. 4f'). The hEC1 + hEC2 + hEC3 combination further extended this pattern, closely mirroring the endogenous *Hmx1* spatial expression across regions α, β, and γ (Fig. 4g' and Supplementary Fig. 6f). At E14.5, the staining pattern of hEC1 + hEC2 + hEC3 closely matched the endogenous spatial *Hmx1* expression around the pinna region, especially in the lower pinna part adjacent to the proximal mandibular region (Fig. 4g), correlating with areas of strong *Hmx1* expression (Fig. 4a).

These results indicate that spatial specificity of *Hmx1* expression around PA2 is finely tuned by the coordinated and synergistic enhancers within an enhancer cluster, which we term the Position Identity Hierarchical Enhancer Cluster (PI-HEC): hEC1 is responsible for the majority of the transcriptional output, while hEC2 and hEC3 provide supplementary positional information for the transcriptional activity in the PA2 region. However, they also act to restrict *Hmx1* activity in the frontonasal process and limbs. hEC1 and hEC2 likely function as synergistic enhancers, driving transcriptional activity both in terms of spatial specificity and expression levels, whereas hEC3 exhibits hierarchical characteristics when combined with hEC1 and hEC2.

## Regulatory dynamics of motif clusters within human EC1 responsible for activity and specificity

Although the three enhancers within the PI-HEC coordinately and synergistically regulate the spatiotemporal expression characteristics of *Hmx1*, we also found that hEC1 possesses the dual attributes of dictating the primary activity and proximal regional identity. Additionally, building on our previous research with the orthologous mouse sequence (mEC1) of hEC1, which highlighted a crucial 32 bp sequence regulated by the Hox-Pbx-Meis complex[14], we extended our analysis to hEC1. To delve deeper into the regulatory dynamics, we targeted five motif clusters (D1, D2, D3, D4, and D5 of hEC1), identified by TF-binding motif analysis using FIMO (Fig. 5a and Supplementary Data 2). The impacts of deleting these individual clusters were evaluated using both in vitro (PA-like hCNCCs) and in vivo (LacZ assay) methods.

In vitro luciferase assays showed that deletions of D2, D3, D4, and D5 significantly diminished hEC1 activity, with D4 almost completely eliminating this activity (Fig. 5b, top panel). The in vivo LacZ assay in mouse embryos revealed similar trends but also some notable differences (Fig. 5b, bottom panel). The D1 deletion, while not affecting activity in the cell line, resulted in almost complete loss of hEC1 activity in PA2 and other embryonic regions. Deletions of D2 and D5 led to substantial reductions in staining, similar to the luciferase results (Fig. 5b, bottom panel and Supplementary Fig. 7a). Interestingly, the D3 deletion, despite reducing activity in vitro, maintained visible LacZ staining across the embryo, with a spatial shift in the PA2 region from ventral to dorsal and ectopic PA1 staining. The D4 deletion eliminated staining throughout the embryo, in accordance with the cell line luciferase results.

These observations emphasize the complex regulatory mechanisms that control both the activity and specificity of the hEC1 enhancer. Concurrently, the distinct relationships of the five motif clusters for the activity and spatial specificity of gene expression provide a valuable model for elucidating the regulatory rules governing enhancers in pinna development[16].

## hEC1 activity and specificity is coordinately controlled by multiple TFs

To investigate the proteins that bind to hEC1, we performed in vitro DNA pull-down assays using E11.5 PA2 tissue nuclear proteins and biotinylated hEC1 (Fig. 5c). Proteomic analysis via LS/MS (Liquid Chromatography-Mass Spectrometry) highlighted homeodomain TFs (HD-TFs) such as Meis1, Pbx1/3, Dlx6 as the predominant hEC1-binding family of proteins (Fig. 5d and Supplementary Data 3). Considering the lower sensitivity of this in vitro experiment and the prominence of a 'Coordinator' motif, a 17-bp long DNA sequence encompassing basic helix-loop-helix (bHLH)- and HD-binding motifs in hCNCCs[31,33], we

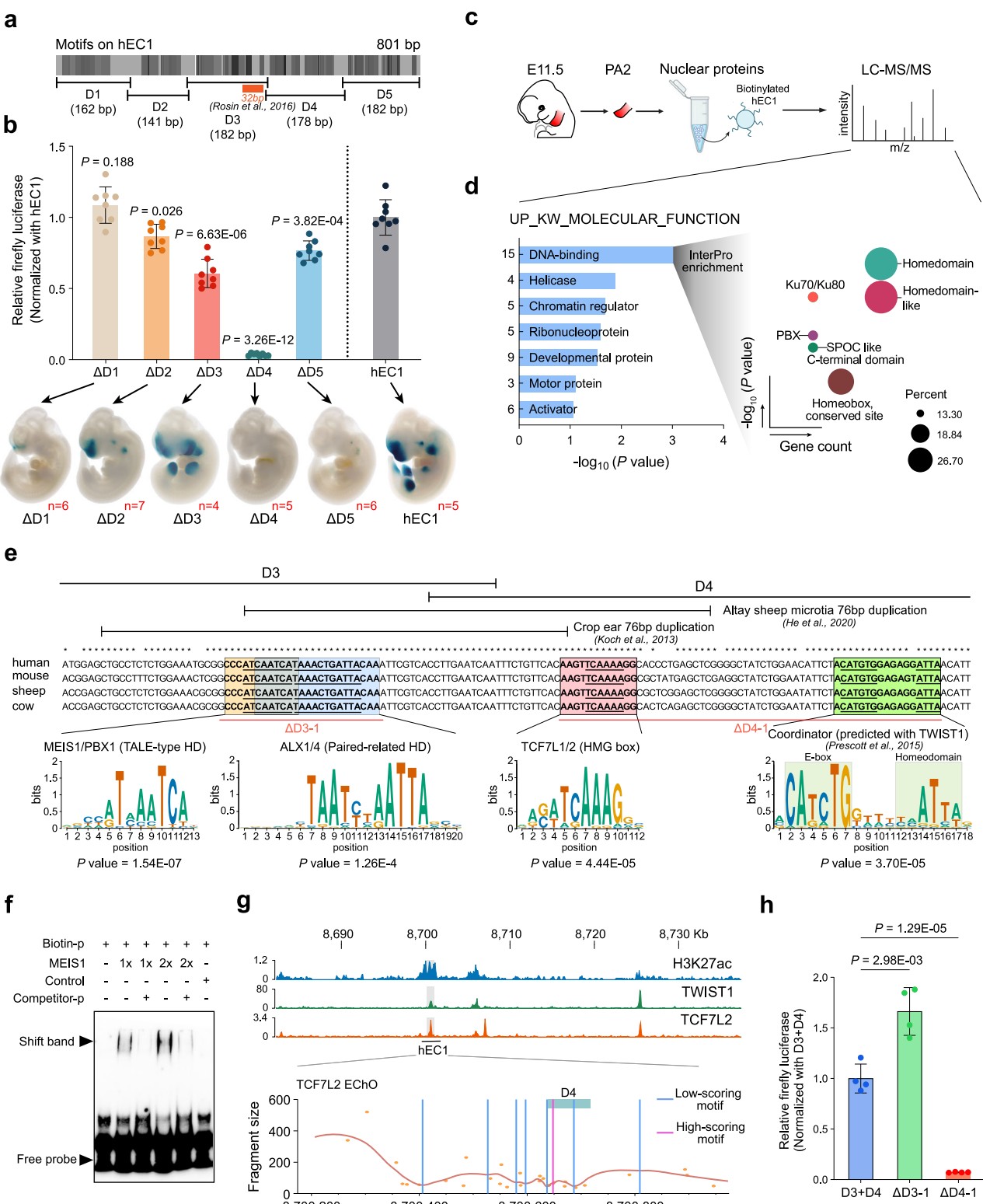

**f**

Biotin-p + + + + + +
MEIS1 – 1x 1x 2x 2x –
Control – – – – – +
Competitor-p – – + – + –

Shift band ▶

Free probe ▶

further examined our motif prediction near the HD-TFs, and found that a pattern containing HD, high mobility group (HMG) box and Coordinator (predicted with TWIST1 motif) sequences are present in the D1 (DLX1-HD) and D3-D4 region (MEIS1/PBX1-HD) (Fig. 5e and Supplementary Data 2). Therefore, the discrepancy between the luciferase assay in the cell line and the transgenic LacZ regarding the effect of D1 deletion on hEC1 activity may result from the difference in the *DLX* gene family expression between these two systems (Supplementary Fig. 4a).

Here, we focus on the conserved D3-D4 region, containing three high-affinity binding sites (*P* value ≤ 1E-04) including MEIS1/PBX1, TCF7L1/2 and Coordinator and one low-affinity binding site (*P* value > 1E-04) of ALX1/4, which are contained within the larger 76 bp duplications associated with auricular malformations in sheep and cow microtia (Fig. 5e). To validate the TF-binding events, we performed CUT&RUN assays in in vitro derived hCNCCs targeting TCF7L2 and reanalyzed one publicly available TWIST1 CUT&RUN dataset in hCNCCs[33], and confirmed their binding capacity (Fig. 5g and

**Fig. 5 | The grammar of the hEC1 enhancer is shaped by multifactorial transcription factors. a** Motif analysis on the hEC1 sequence. The color intensity represents the affinity of the predicted TFBSs, with the previously identified core 32 bp marked[14]. Five clusters of motifs, including D1 (162 bp), D2 (141 bp), D3 (182 bp), D4 (178 bp), and D5 (182 bp), are sequentially deleted. **b** hEC1 activity is evaluated using in vitro and in vivo strategies. Luciferase assay was performed across three independent experiments with two technical replicates each. Data are shown as mean ± SEM (n = 6). *P*-values were calculated using unpaired Student's *t*-test (two-tailed). The red number represents the number of positive LacZ staining embryos, see also Supplementary Fig. 7a. **c** A schematic process of DNA pull-down using PA2 tissue from E11.5 mouse embryos and biotinylated hEC1 is presented. **d** Gene enrichment analysis of proteins identified in the DNA pull-down assay. For fifteen DNA-binding proteins, InterPro enrichment was further performed to detect TF classification. *P* values were corrected by Benjamini-Hochberg method. **e** Dissection of motif structure across a core region linking D3 with D4. Two 76 bp

duplication regions previously reported in sheep and cow are shown, respectively. The middle panel displays DNA alignment across human, mouse, sheep and cow. Four TFBSs are highlighted and the underline marked the core motif sequence. The bottom panel shows motifs logo, *P* values were calculated using a dynamic programming algorithm to convert log-odds scores, assuming a zero-order background model. Two deletions, including ΔD3-1 and ΔD4-1 used in Fig. 5h are shown. **f** An EMSA assay reveals the binding of human MEIS1 protein to the predicted motif binding site in (**e**). Experiments were performed with two independent replicates. **g** CUT&RUN assay of H3K27ac, TWIST1 and TCF7L2 across the BCE core locus. Local minimal DNA protection by TF-DNA binding reveals multiple TCF7L2-binding sites. **h** Removal of ΔD4-1 containing HMG-box and Coordinator dramatically diminish the activity of hEC1. Data are shown as mean ± SEM (n = 4). *P*-values were calculated using unpaired Student's *t*-test (two-tailed). Source data are provided as a Source Data file.

Supplementary Fig. 7b, c). We then further applied enhanced chromatin occupancy (EChO) analysis, which identifies TF interaction foci[34], to explore TCF7L2 binding patterns in the hEC1 region. One high-scoring motif was found in the D4 region, consistent with the prediction result (Fig. 5g and Supplementary Data 4). The binding capacity of human MEIS1 protein in hEC1 was also validated using the EMSA method (Fig. 5f).

We further constructed two deletion regions, D3-1 (including HD motifs) and D4-1 (including HMG box and Coordinator motif) (Fig. 5e), with luciferase assay results demonstrating that removal of D4-1 dramatically suppressed the activity of hEC1 (Fig. 5h). *Meis1, Tcf7l2, Twist1/Tcf4*, that interacted with the D4-1 sequence, were all confirmed to be expressed in the PA2 region using public mCNCCs bulk RNA-seq[20] and *WISH* datasets (Supplementary Fig. 7d and 7e). Collectively, these findings suggest that the activity and specificity of hEC1 are modulated by a combination of different families of TFs. The underlying regulatory code, which resides within the characteristics of the enhancer sequences, offers a complex process for pinna development.

### Impact of spatial *Hmx1* expression on pinna development: insights from transgenic mouse models mimicking human BCE anomalies

To investigate human BCE development mechanisms, we created an endogenous/knock-in transgenic mouse model, *mEC1^dup/dup*, containing an additional copy of mEC1 sequence on each allele, ie. four copies of mEC1 (Fig. 6a, b and Supplementary Fig. 8a). LacZ reporter activity for mEC1 in the pinna region was similar to that of hEC1 but less intense (Figs. 4b, 6c and Supplementary Fig. 8b). *mEC1^dup/dup* mice showed significant pinna development abnormalities compared to wild-type (WT) mice (Fig. 6d, e, g). The helix of the pinna was notably affected, with a reduced area (Fig. 6e, f), and the pinna angle changed, suggesting altered cartilage or muscle formation (Fig. 6g, h). Older transgenic mice exhibited progressive eye dysgenesis, albeit less consistent than ear malformations (Supplementary Fig. 8c). WISH showed spatially expanded *Hmx1* expression in the PA2 region at E10.5 and in the upper pinna region at E14.5 in the *mEC1^dup/dup* mice (Fig. 6i, j), correlating well with the activity of mEC1 (Fig. 6c). These findings underscore the impact of enhancer copy number variation in developmental morphology, especially in pinna formation.

To explore the interplay between EC1 and *Hmx1* expression in CNCCs in ear development, we used multiple mouse models disrupting *Hmx1* function in various ways: *Wnt1::Cre;Hmx1^fl/fl* (conditional *Hmx1* knockout in mCNCCs), *mEC1^del/del*, and *dumbo* (mutation in exon 1 of *Hmx1* resulting in truncated protein after residue Gln65[12]) to test 0 to 4 copies of the mEC1 enhancer in transgenic animals (Fig. 6k and Supplementary Fig. 8d). The pinna morphology in these models exhibits similar low set, laterally protruding characteristics (Fig. 6k), suggesting that mEC1 is crucial for proper *Hmx1* expression in mCNCCs and normal pinna development. We did not observe external

ear malformations in *mEC1^del/+* (one copy of mEC1) or *mEC1^dup/+* (three copies of mEC1). To further investigate the contribution of mEC1 to *Hmx1* expression level, we evaluated the effects of mEC1 duplication and deletion on *Hmx1* expression levels. The results showed that mEC1 duplication led to a twofold increase in *Hmx1* expression, while mEC1 deletion caused a ninefold decrease (Fig. 6l). These findings highlight the critical role of mEC1 in regulating *Hmx1* expression in mice. Finally, performing micro-CT analysis on all mouse models revealed that the normal development of the paroccipital process, a conical prominence of bone adjacent to the outer ear and serving as an attachment point for certain neck muscles, was significantly impacted in the *dumbo*, *Wnt1::Cre;Hmx1^fl/fl* and *mEC1^del/del* mice but not in *mEC1^dup/dup* (Fig. 6m and Supplementary Fig. 8e-j).The reduced size of the paroccipital process may affect muscle attachment and, in turn, contribute to the lateral orientation of the pinna in these mutants (vs the more upright position in wildtype animals).

### EC1 duplication results in ectopic expression of *Hmx1*, leading to disrupted cell differentiation in ear development

To gain greater insight into the mechanisms underlying abnormal pinna development in the *mEC1^dup/dup* mouse model, we performed scRNA and bulk RNA sequencing at the E14.5 embryonic stage in the whole pinna prominence (Fig. 7a). scRNA-seq analysis revealed that the pinna prominence at this stage is primarily composed of various types of fibroblasts (clusters 1-8), chondrocytes (cluster 9), myogenic cells (cluster 10 and 11), endothelial cells (cluster 12), epithelial cells (cluster 13), melanocytes (cluster 14), neuron (cluster 15) and glial cells (cluster 16 and 17) (Fig. 7b and Supplementary Fig. 9a, b). The *Hmx1* gene is primarily expressed in fibroblasts, and the duplication of mEC1 leads to a significant expansion of *Hmx1* expression in these cells (Fig. 7c). Bulk RNA-seq analysis also revealed that the duplication of the mEC1 led to overexpression of *Hmx1*, but the majority of differentially expressed genes (DEGs) were downregulated. This suggests that increased expression of *Hmx1* plays a critical role in suppressing numerous genes during pinna development (Fig. 7d and Supplementary Data 5). However, the analysis of the promoters and cis-regulatory elements of these downregulated genes did not reveal an enrichment of the *Hmx1* motif, suggesting that the gene expression repression is likely indirect (Supplementary Data 6). Gene Ontology (GO) enrichment analysis of these downregulated DEGs showed their primary involvement in pathways essential to ear development, such as epidermis development, extracellular collagen organization, and myofibril assembly (Supplementary Fig. 9c and Supplementary Data 7). In addition, two cartilage development-related genes *Comp* and *Arhgap36* were also downregulated, although they did not rank high in the GO enrichment analysis (Fig. 7d). The proportions of these cell types, such as perimysial cells (cluster 6, closely related to muscle development), chondrocytes (cluster 9), myocytes (cluster 11), endothelial cells (cluster 12), and epithelial cells (cluster 13), have undergone significant changes (Supplementary Fig. 9d), which are consistent with the results of bulk RNA-seq.

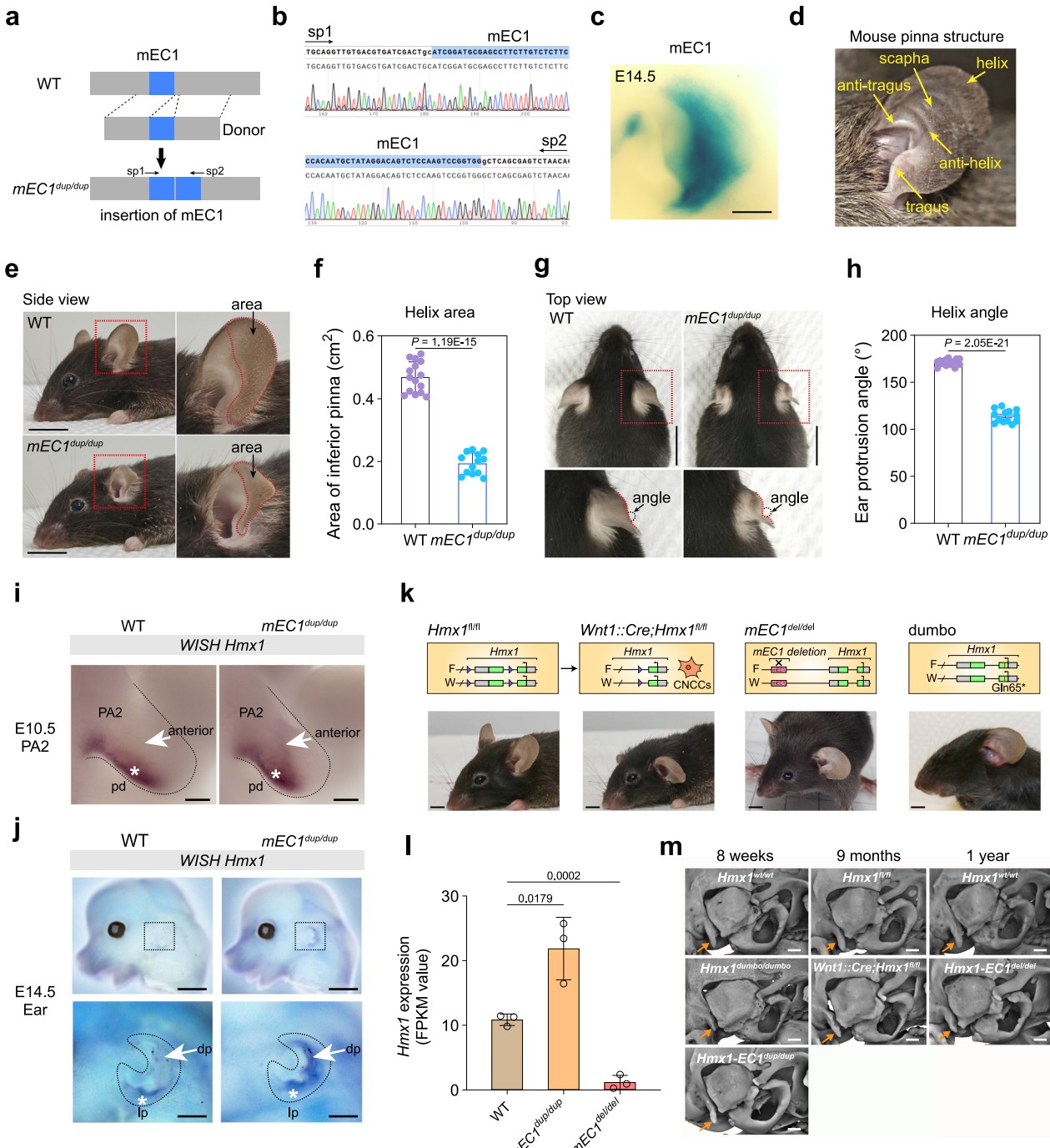

**Fig. 6 | mEC1 Knock-in in mouse recapitulates human constricted ear phenotypes, and mEC1-dependent *Hmx1* expression in mCNCCs is important for ear development. a** A CRISPR/Cas9 genome editing strategy was used to insert an extra copy of mEC1. **b** Sanger sequencing result of *mEC1^dup/dup^* transgenic model. **c** The expression pattern of mEC1 at E14.5 (n = 7; see also Supplementary Fig. 8b). Scale bar measures 500 μm. **d** Illustration of mouse pinna structure. (**e–h**) Side and top view comparison of WT and *mEC1^dup/dup^*. The red rectangle highlights the altered aera of inferior pinna (**e, f**) and wrinkled helix rim (**g, h**). Data are presented as mean ± S.E.M. (n = 15). *P*-values were calculated using an unpaired Student's *t*-test (two-tailed). Scale bars represent 1 cm. **i, j** Whole-mount in situ hybridization (WISH) showing *Hmx1* expression in WT and *mEC1^dup/dup^* embryos at E10.5 (**i**) and E14.5 (**j**). Three independent embryos were assessed at each age. The white star represents original *Hmx1* expression in the posterior-distal region (pd region) (**i**) and in the lower part of pinna (lp) (**j**); and the white arrow indicates expanded *Hmx1*

expression in the anterior PA2 region (**i**) and in the distal part of the outer ear (dp) (**j**). Scale bars are 50 μm (**i**) and 500 μm (**j**), respectively. **k** Loss of *Hmx1* expression through three different strategies results in a similar low-set and protruding ear phenotype. These strategies include *Hmx1^fl/fl^* (homozygous floxed allele), *Wnt1::Cre;Hmx1^fl/fl^* (conditional deletion of *Hmx1* in CNCCs), and *mEC1^del/del^* (homozygous deletion of mEC1). The 'dumbo' mutation, representing a missense mutation in exon 1 of Hmx1, results in a truncated protein. Scale bars measure 5 mm. **l** Expression of *Hmx1* (FPKM value) in different genotypes. Data are shown as mean ± SEM (n = 3). Three independent experiments were performed. *P*-values were calculated using unpaired Student's *t*-test (two-tailed). **m** Micro-CT analysis of ear-related structure in different mouse models, at eight weeks, nine months and one year. The gold arrow points to the paroccipital process. Scale bars measure 500 μm. Source data are provided as a Source Data file.

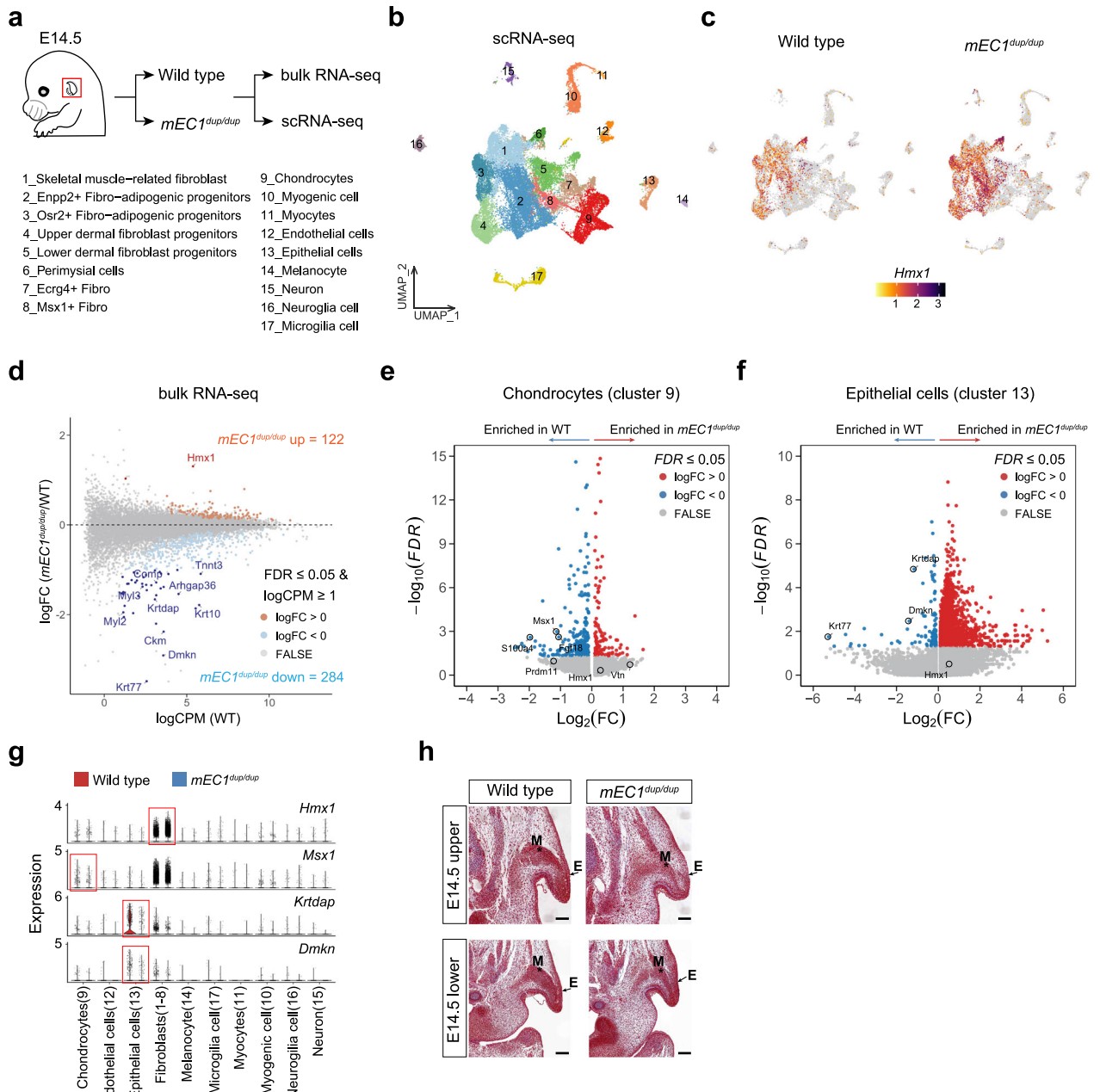

**Fig. 7 | *Hmx1* misexpression in mouse leads to widespread abnormal development in pinna structures including cartilage, fibroblasts, muscle, and epidermis. a** Diagram of micro-dissected pinna prominence at E14.5 for bulk and sc-RNA sequencing. **b** UMAP dimensional reduction visualizes seventeen cell clusters of pinna prominence at E14.5. **c** *Hmx1* expression is mainly expressed in fibroblasts and expanded in these cells in *mEC1^(dup/dup)* compared with wild type. **d** RNA-seq analysis compares the whole pinna structure between wild-type and E14.5 *mEC1^(dup/dup)* mice. Differentially expressed genes (DEGs) are defined as FDR ≤ 0.05, log2 CPM ≥ 1, and |log2 FC | 0. Up- and down-regulated genes are shown as gold and blue dots, respectively. Important DEGs including two previously reported cartilage development-related genes including *Comp*[85] and *Arhgap36*[86] were labeled. **e**, **f** Single cell differentially gene analysis in chondrocytes (**e**) and epithelial cell (**f**) cluster. DEGs are defined as FDR value ≤ 0.05 and |log2 FC | 0. Up- and down-regulated genes are shown as red and blue dots, respectively. Some genes that are critical for cell development are labeled. Source data are provided as a Source Data file. **g**, Violin plots show gene expression comparison between wild type and *mEC1^(dup/dup)* in each cell cluster. Red rectangle represents genes that are significant changed in corresponding cell clusters (*Hmx1* in fibroblast cluster, *Msx1* in chondrocyte cluster, *Krtdap* and *Dmkn* in epithelial cell cluster). **h**, Masson's trichrome staining at E14.5 (upper and lower pinna parts) is shown for WT and *mEC1^(dup/dup)*. Scale bars measure 100 μm. Experiments were performed with three independent embryos.

However, as revealed by the scRNA-seq results and previous studies, *Hmx1* is hardly expressed in these cells, and the *Hmx1* gene in *mEC1^(dup/dup)* mice has not expanded to these cells either (Fig. 7c). Thus, in conjunction with the WISH results (Fig. 6j), we hypothesize that the defects observed in the outer ear are due to ectopic expression of *Hmx1* in other pinna fibroblast zones, which further affects the development of other cells through intercellular interactions.

To confirm dysregulated genes discovered by bulk RNA-seq analysis and identify more disrupted genes that may not be detectable by bulk RNA-seq, we performed differential gene analysis in the fibroblasts (clusters 1-8), chondrocytes (cluster 9), myogenic cells (cluster 10-11) and epithelial cells (cluster 13). Differential gene analysis revealed significant downregulation of genes such as *Msx1*, *Fgf18*, and *S100a4* in the chondrocyte subpopulation (Fig. 7e), and *Cebpb* and

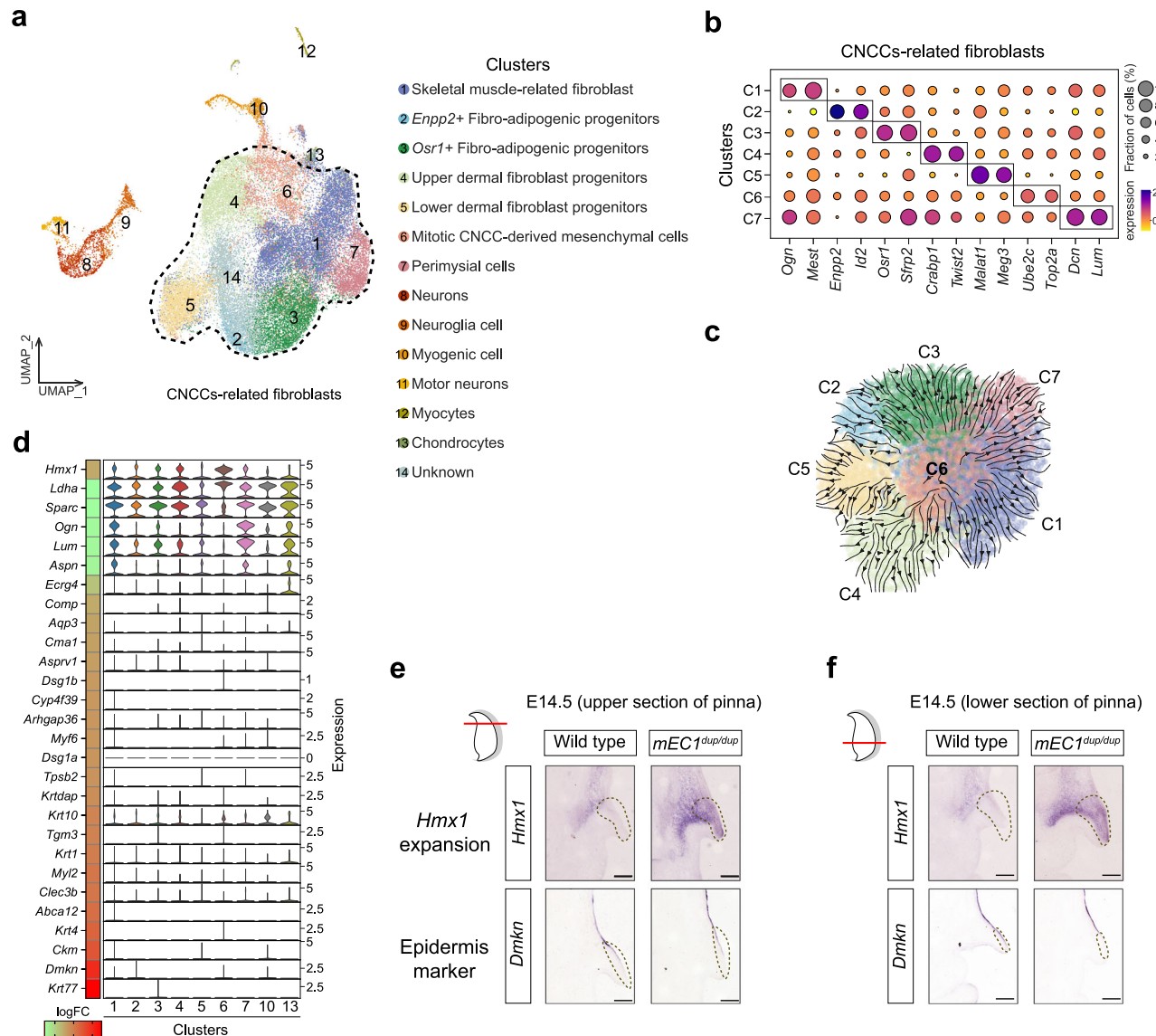

**Fig. 8 | Transcriptional trajectory analysis of GFP-positive cells. a** UMAP dimensional reduction visualizes *Hmx1+* expressed cells combined across E10.5, E12.5, and E14.5. CNCCs-related fibroblasts clusters are outlined with dotted lines for further analysis in **b** and **c**. **b** A dot plot displays two marker genes for each cluster identified within the CNCCs-related fibroblasts. The cell clusters of 1-7 are the same as in **a**. **c** Developmental trajectory analysis from E10.5 CNCC mesenchymal cells (cluster 6) to the fibroblasts defined in (**a**) is illustrated. The cell clusters of 1-7 are the same as in (**b**). **d** The expression of cartilage, fibroblast, muscle, and epidermis development-related downregulated DEGs across nine cell clusters (including 7 fibroblast, chondrocyte, and myogenic cell clusters) is shown. The heatmap on the left panel depicts the fold change of each gene. **e, f** In situ hybridization staining for *Hmx1* and *Dmkn* expression in wild-type and *mEC1dup/dup* embryos at upper (**e**) and lower (**f**) regions of the pinna. Scale bars measure 100 μm. Three independent replicates were performed.

*Ccl21a* in the fibroblast subpopulation (Supplementary Fig. 9e), which are closely associated with cell proliferation. The differential analysis of epithelial cells showed relatively high concordance with the bulk RNA-seq results, with both identifying genes such as *Krt77*, *Dmkn*, and *Krtdap* (Figs. 7f and 7g), which are closely associated with epidermal development. Although the differential gene analysis of the muscle subpopulation showed less consistency with the bulk RNA-seq results, it still identified genes such as *Pax3*, *Ets1*, and *Camk2a*, which are closely associated with muscle tissue development (Supplementary Fig. 9f). Masson's trichrome staining at E14.5 revealed delayed development of muscle and epidermis in the outer ear of *mEC1dup/dup* mice compared to that in wild-type mice, consistent with the bulk RNA-seq results (Fig. 7h).

To definitively characterize the cell types expressing the *Hmx1* gene within the outer ear, and to further elucidate the molecular mechanisms involved in BCE, we further used scRNA-seq to perform

transcriptional trajectory analysis of GFP-positive cells isolated from Hmx1-P2A-GFP embryos at three pivotal developmental stages: E10.5, E12.5, and E14.5 (Supplementary Fig. 10a). Primarily, the *Hmx1+* cells were identified as fibroblasts, with a small fraction defined as other cell types including neurons, myogenic cells, and chondrocytes also detected (Fig. 8a and Supplementary Fig. 10b). The fibroblasts were further categorized into seven subtypes based on distinct marker genes. These include skeletal muscle-related fibro (*Ogn, Mest*)[35], two fibro-adipogenic progenitors (*Enpp2+, Osr1+*)[36,37], dermal fibro progenitors (upper: *Crabp1, Twist2*; lower: *Malat1, Meg3*)[38], miotic CNCC-derived mesenchymal cells (*Ube2c, Top2a*)[39], and perimysial cells (*Dcn, Lum*)[35] (Fig. 8b and Supplementary Fig. 10c). In the developmental trajectory analysis with the cluster of CNCC-derived mesenchymal cells (C6) as the root, we demonstrated that it has the potential to differentiate into the other six fibroblast-related cell clusters (C1, C2, C3, C4, C5, C7) (Fig. 8c), suggesting an important role of the *Hmx1* gene

on pinna fibroblast development. We further noticed that genes that were substantially downregulated (logFC > 1) in $mEC1^{dup/dup}$ mice were predominantly not expressed in $Hmx1^+$ cells of WT mice, substantiating our earlier hypothesis that ectopic $Hmx1$ expression suppresses cell fate differentiation (Fig. 8d).

To provide conclusive evidence for our hypothesis that defects in the outer ear are attributable to aberrant $Hmx1$ expression, in situ section RNA hybridization ($Hmx1$, $Dmkn$) was performed on both upper and lower sections of the pinna in wild-type and $mEC1^{dup/dup}$ mice. The results demonstrated upregulated expression of $Hmx1$ gene in the basal pinna zone, and more notably, an expansion of $Hmx1$ expression from the basal pinna zone to the distal tip of the pinna, coupled with a downregulation of $Dmkn$ expression in these areas (Figs. 8e and 8f), which may result from a non-cell autonomous effect of $Hmx1$ expression on the tissue adjacent to the epidermis. Taken together, our data provide strong evidence for the critical role of $Hmx1$ in the observed abnormalities in the outer ear.

## Discussion

This study examines the genomes of seven human pedigrees, whose members present with constricted ears. All affected individuals showed a duplication of a genomic region termed the BCE locus near the $HMX1$ gene. We present compelling evidence that the duplication of regulatory elements in the BCE locus erroneously modulates the spatiotemporal expression characteristics of $HMX1$, leading to the ear developmental abnormality.

The evolutionarily conserved region (ECR) downstream of $Hmx1$ has been implicated in eye and external ear malformations across multiple species[8,9,14], and linked to human isolated bilateral concha-type microtia[13]. However, the explicit pathogenic mechanism in BCE patients remained unclear. This uncertainty was due to observations of larger genomic duplications extending beyond the single ECR element in previously reported pedigrees[13] and in our seven newly analyzed pedigrees. We have identified a minimal critical region, the 'BCE core locus', encompassing three distinct cis-regulatory elements. We demonstrate that this locus is the causative factor for bilateral constricted ear by employing transgenic cell and mouse models. In many CNV-related developmental abnormalities, distinct cis-regulatory elements are organized as a cluster in the mutation interval[15,28,30]. The deletion or duplication of these elements induces morphological variations, stemming from differential gene expression[15,40]. Our mouse models reveal that the morphology of the pinna is particularly sensitive to changes in $Hmx1$ gene expression due to variation in the copy numbers of enhancer sequences. Therefore, the variation in pinna morphology among pedigrees may also be attributed to differences in the number of cis-regulatory elements within the associated CNV. This hypothesis is primarily supported by our observation of additional weak enhancers (CS stages) located outside the defined BCE core locus (Fig. 2a), which may interact with the $HMX1$ promoter. On the other hand, distinct interaction patterns between these elements and the $Hmx1$ promoter may also influence pinna phenotype, which may require the study of patient-specific hiPSCs[30] for further analysis. Alternatively, this variation may reflect the contribution of other family-specific genetic variants or epigenetic effectors.

Enhancers activated within specific developmental windows are crucial for normal craniofacial development, as evidenced by the study of long-range enhancer clusters in hCNCCs in the PRS locus[28] and inter-TAD interactions controlling $Hoxa$ function[29,41]. Lineage tracing studies in mice have demonstrated that CNCCs play a crucial role in the development of fibroblasts and cartilage in the external ear[7,42]. CNCC models are also widely used to study craniofacial malformations[28,30,41,43,44], and many studies have highlighted the close association between CNCCs and external ear deformities, such as CFM and microtia[17,19,29,45]. Therefore, we selected CNCCs as our in-vitro validation model to complement and support the findings from our in-

vivo experiments. Our results also demonstrate that the enhancers in the BCE core locus are activated at different developmental stages of hCNCCs: hEC2 may be activated at early hCNCCs stage, and hEC1 is activated at late- and PA-like hCNCCs stage (Fig. 9a). The anterior-posterior positional identity of gene expression in CNCCs is pivotal for craniofacial development[20]. Likewise, optimizing the proximo-distal positional identity within the same pharyngeal arch is equally crucial[21]. In this study, we elucidate the contributions of both enhancer and motif cluster sequences in determining the developmental pattern of Hmx1 gene expression. We have delineated an intra-arch positional identity enhancer cluster, named PI-HEC, which includes hEC1, hEC2, and hEC3. While we reveal the critical role of hEC1 in pinna development and BCE in endogenous context, the functional contributions of hEC2 and hEC3 remain to be fully characterized in vivo. Redundancy within clusters of enhancers, or within super enhancer (SE) regions, is a prevalent phenomenon that has been observed in multiple genomic loci[46], providing phenotypic robustness[47,48], buffering gene expression against mutations[49] and transcriptional noise[50]. Similar additive effects of a multipartite enhancer cluster have also been reported in the $Ihh$ locus[15]. Besides these two modes, synergy (super-additive) of different enhancers is also observed in the EC1.45 of the PRS locus[28], the $Fgf5$ locus[51], and regulating gap gene-expression patterns in $Drosophila$ embryos[52] and coordinating cell fate determination[53]. Compared with aforementioned interaction modes of multipartite enhancers, the PI-HEC presents a distinct scenario in which they possess both coordinated and synergistic properties: hEC1 serves as the primary transcriptional activity driver in the proximal region, hEC2 enhances transcription weakly but contributes distal positional information, and hEC3 alone is a weak, unstable enhancer but compensates caudal positional information and becomes stable when combined with hEC1 and hEC2 (Fig. 9b). This collaborative mode is somewhat similar to the interaction between predominant sites and supportive sites studied in the estrogen receptor alpha binding sites (ERBS)[54] and recently reported non-classical enhancers (facilitators) in the mouse $\alpha$-globin cluster[55]. This study presents evidence for the synergistic and coordinated effects of multipartite regulatory elements on the spatial regulation of gene expression in a disease-related locus at the transcriptional reporter level. However, further in vivo studies are required to fully elucidate the contributions of hEC2 and hEC3 to these synergistic and coordinated effects. The underlying mechanism for this pattern may be explained by the spatial proximity of these enhancers. In this case, TFs or coactivators, responsible for driving transcription, recruited to hEC1 can spread to hEC2 and hEC3[56,57]. However, this possibility needs to be further investigated in its original three-dimensional genomic context[58]. Furthermore, whether this specific mode of interplay between multipartite enhancers is a common phenomenon in the regulation of intra-arch gene patterning in CNCCs also requires further exploration, but it potentially signifies a paradigm shift in our understanding of multipartite enhancers or SE dynamics[59].

The molecular basis of gene patterning is predominantly shaped by the interaction between tissue-specific TFs and enhancer structural organization[16,57,60–62]. While enhancers have been extensively studied for their role in governing spatial gene expression at the embryonic level[15,28,63–66] and across branchial arches[67], the precise mechanisms dictating gene expression within intra-arch regions remain unclear. Our investigation into the spatial specificity of $Hmx1$ expression highlights that gene patterning within specific sub-structures, such as the PA2 zone, is regulated by the highly restricted activity of enhancers. We have uncovered the vital role of TALE-type HD TFs, HMG box-containing TFs, and Coordinator-related TFs (TWIST1) in their interaction with hEC1, a key element for its functional integrity. TALE-type HD TFs are critical for the formation of the PA2 ground state[68], and our study also suggests a functional involvement in maintaining the proximal spatial position of hEC1. TCF7L2, a TF newly identified in this study as binding to hEC1, and the Coordinator motif in the D4 region,

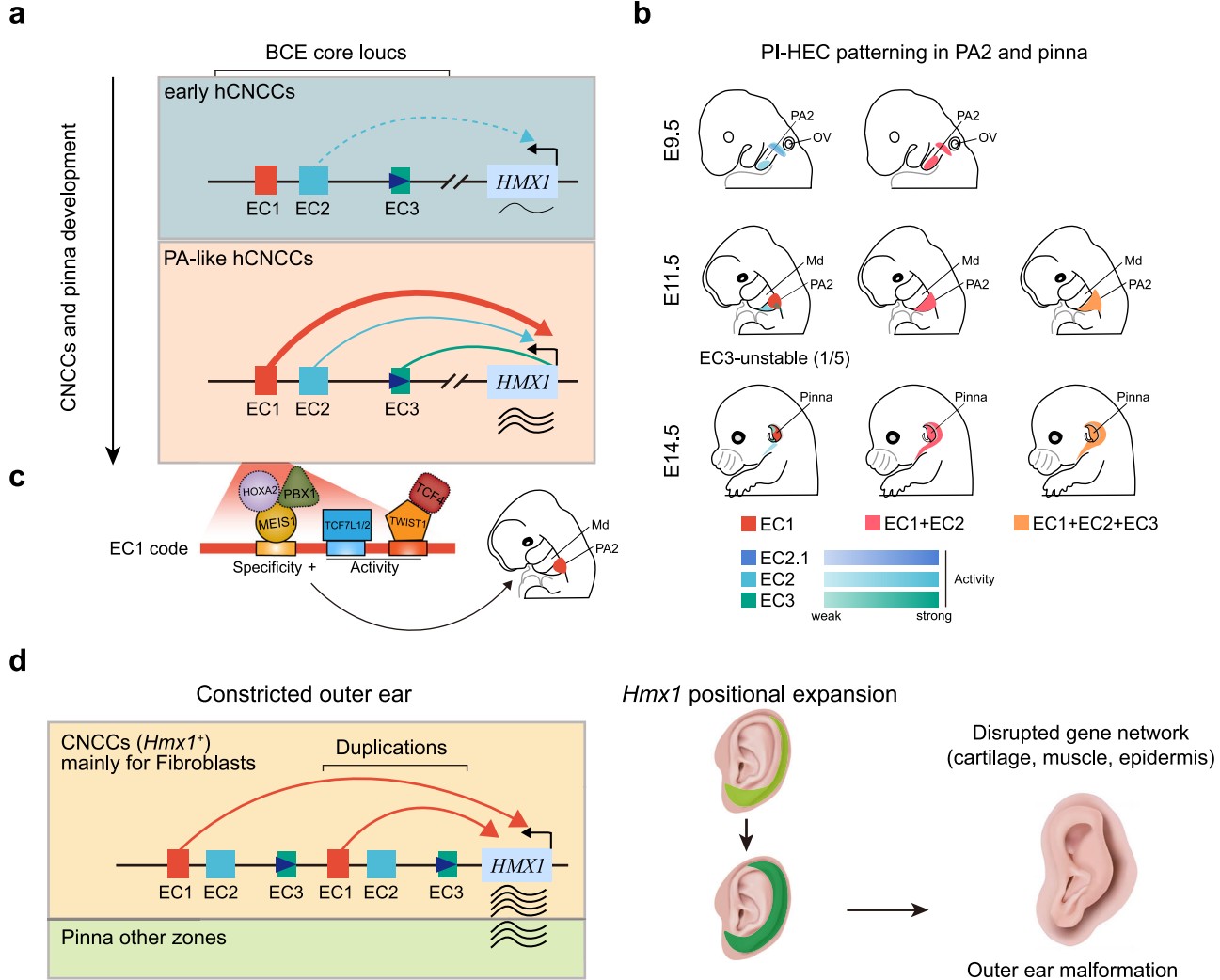

**Fig. 9 | Model. a** Stage-specific enhancer activities in hCNCCs in the BCE core locus. EC2 may be initiated at the early hCNCCs stage, driving weak expression of *HMX1*; EC1 drives most *HMX1* transcriptional output at the PA-like hCNCCs stage, and EC3 may serve as a structural element. **b** Coordinated and synergistic PI-HEC patterning in PA2 and pinna zones. EC2.1, EC2, and EC3, represented by varying color intensities, are weak enhancers. **c** A motif pattern consisting of TALE-type HDs (HOXA2/MEIS1/PBX1), HMG box (TCF7L1/2), and Coordinator (TWIST1/ TCF4) endows the dominant EC1 with specificity and activity. **d** A comprehensive disease model for BCE. Based on scRNA-seq and *mEC1^{dup/dup}* mouse model, we demonstrated that *Hmx1* is mainly responsible for CNCCs-derived fibroblasts development in pinna structure, and duplication of EC1 causes aberrant expression of *Hmx1* in fibroblasts and ectopic expression in the other zones of the pinna, disrupting the gene network in cartilage, muscle, epidermis, and leading to outer ear malformation.

are both crucial for the activity of hEC1 (Fig. 9c). TCF7L2 is a Wnt signaling effector, modulating the neural crest gene expression in a position-dependent manner[69]; whereas the Coordinator motif, which can be bound by many bHLH and HD TFs, is important for shaping facial morphology[33]. Our study on the hEC1 structural arrangement further revealed that a specific coordinator pattern, consisting of HD, HMG-box and Coordinator motifs (in the D1 and D3-D4 regions), is crucial for enhancer activity and specificity. Whether these regulatory rules exist in other craniofacial-specific enhancers remains to be investigated.

Duplication of genomic sequences, particularly those containing enhancers, can lead to neo-TAD formation at the *SOX9* locus and *KCNJ2* misexpression[70], precipitate ectopic gene expression[15] resulting in limb malformations, and even contribute to lung adenocarcinoma[71]. Here, the duplication of hEC1 also disrupts the spatial specificity of *Hmx1* expression, contributing to aberrations in the proximal helix and scapha in mice. These regions mainly originate from neural crest-derived mesenchymal cells[42,45,72], which aligns with our scRNA-seq data and underscores the pivotal role of *Hmx1* in the normal development

of CNCC-derived fibroblasts in the outer ear (Fig. 9d). CNCC-specific knockout of *Hmx1* in mice leads to a low-set, protruding ear phenotype, consistent with findings in mEC1 deletion mice and 'dumbo' mice[12]. This suggests that appropriate *Hmx1* expression in CNCCs is essential for pinna organogenesis[7], further substantiating the validity of employing CNCCs as a cellular model for the investigation of BCE pathogenesis. Notably, the ear malformations observed in most human BCE patients within our study pedigrees are more severe than those in our *mEC1^{dup/dup}* mouse model, as well as there not being the progressive eye dysgenesis in BCE patients as seen in the mouse. This discrepancy may arise from a broader expansion of *Hmx1* expression due to the extent of the duplication of the PI-HEC region, rather than solely the hEC1 duplication[15] and the difference between the extent of the CNV or *HMX1* expression pattern between BCE patients and *mEC1^{dup/dup}* mice. Additionally, it is worth noting that we did not observe external ear malformations in *mEC1^{del/+}* (one copy of mEC1) or *mEC1^{dup/+}* (three copies of mEC1), which may be due to the insensitivity of mice to *Hmx1* dosage and the activation of genetic compensation mechanisms, a phenomenon commonly observed in mouse models of

human diseases[73,74]. Despite these differences, the phenotypic hallmarks of the $mEC1^{dup/dup}$ mouse, including reduced ear area and a wrinkled helical rim, do grossly mimic the ear phenotype observed in patients.

In summary, our results reveal that CNCC-specific and precise spatial specificity of *Hmx1* expression is vital for the development of fibroblasts in the outer ear, with both loss and gain of *Hmx1* expression contributing to pinna malformation. The gene patterning of *Hmx1* across PA2 is intricately modulated by hEC1, hEC2, and hEC3, with the dominance of hEC1 as an enhancer being further refined by coordinator motif-transcription factor binding events. Our study reveals the complex interplay of gene regulatory activity in precise developmental space and time, and contributes to the understanding of the regulatory rules of gene expression.

## Methods

### Ethics statement
All procedures involving human samples were performed with the approval of the Ethics Committees of the School of Biological Science and Medicine Engineering at Beihang University (BM20210057) and the Plastic Surgery Hospital of the Chinese Academy of Medical Sciences (2017-07). The study has been registered and approved by China's Ministry of Science and Technology (project 2023-CJ0849). Written informed consent for participation in genetic and biological research was obtained from all subjects or their legal guardians. Consent for the publication of photographs was given after a separate explanation and request.

### Samples and recruitment
We recruited members of seven Chinese families (from five separate provinces of China) in which bilateral constricted ear was segregated in a Mendelian fashion. The phenotype in all pedigrees segregated in a manner consistent with autosomal dominant inheritance. All affected subjects presented with a bilateral constricted ear phenotype, as shown in both ears of the probands in Fig. 1b. This phenotype is characterized by a malformation of the external ear, notably reduced size of the helix and absence of the triangular fossa, scaphoid fossa, and antihelix, giving a shell-like appearance. Contrastingly, the crus helicis, concha cavity, tragus, and earlobe maintained their normal structure. No accompanying facial abnormalities were observed.

### Genotyping, target-capture sequencing, and whole-genome sequencing
We extracted DNA from venous blood samples using the TIANamp DNA Midi Kit (Tiangen Biotech, Beijing, China). Genotyping was performed on 40 samples using the Human Omni-Zhonghua chips (Illumina, CA, USA) for pedigrees 1, 2, 3, 4, 6, and 7. Target-capture sequencing was conducted on 32 samples from pedigrees 1, 3, and 7 using the Agilent SureSelect Kit (Agilent Technologies, Santa Clara, CA, USA). Whole-genome sequencing was carried out for the proband of Pedigree 5 using the DNBSEQ-T7 platform (Huada, Shenzhen, China).

For the chip data, we utilized the Genotyping and CNVPartition modules of GenomeStudio v2011.1 for genotype and CNV calling, achieving genotype call rates above 99.6%. The target-capture sequencing involved designing capture oligos for a 5.9 Mb region encompassing the susceptibility locus using the SureDesign tool (Agilent Technologies, Santa Clara, CA, USA). The DNA library was prepared following the manufacturer's specifications, and sequencing was executed on the Illumina HiSeq X10 system (Illumina, San Diego, CA, USA), yielding an average depth of 196X per sample.

Quality control for next-generation sequencing data was conducted using FastQC v1.1.0 and Cutadapt v1.15. Post quality control reads were mapped to the human reference genome (hg38) using BWA v0.7.16a[75]. PCR duplicates were removed with Picard v2.27.0, and BAM file sorting and indexing were performed using Samtools v1.18. SNP and small indel calling were done using the HaplotypeCaller model of GATK v4. For identifying causal variants, we applied criteria including cosegregation with affected and unaffected individuals in each pedigree, minor allele frequency below 0.05 in dbSNP138, ExAC project, and the 1000 Genomes database, and minimum coverage depth of 20 with an alternative allele depth above 5.

### Causal mutation screening
Our screening for causal mutations began with linkage analysis to pinpoint the suspected genomic locus for bilateral isolated constricted ear. We adhered to the quality control (QC) procedures outlined by Mao et al. for pruning SNPs in strong linkage disequilibrium, removing SNPs with Mendelian errors, and verifying sex specifications[18]. Parametric multi-point linkage analysis was conducted on Pedigrees 1, 3, and 7, both individually and collectively, using MERLIN v.1.1.2[76]. For consanguinity analysis, we used the same QC-approved data and performed pairwise IBD estimation for founders from the seven pedigrees using Plink v1.9[77]. A PI-HAT value greater than 0.1 between paired founders indicated a genetic relationship.

After examining all mutations within the coding region of the target area, we assessed copy number variations using differential coverage depth. We utilized the 'getSegmentReadCountsFromBAM' function in cn.mops for read coverage of genomic segments[78]. To detect CNVs, the read count matrix was normalized for each base relative to the mean coverage of the target-capture regions. We excluded segments with an average read count below 90 to minimize noise from untargetable regions. This threshold was based on the theoretical depth in cases where a deletion was homozygous in 22 sequenced cases and heterozygous in 10 controls, given a mean depth of 196 for target regions. We then calculated and displayed the ratio of the average read count of cases to controls for each family. Furthermore, to illustrate the CNV of each case at the target-capture region, we calculated and plotted the case-to-control read count ratio for each family (Supplementary Fig. 1c). The expected ratios for CNV = 3, CNV = 2, and CNV = 1 were approximately 1.5, 1, and 0.67, respectively.

### Culture of human embryonic stem cells (hESCs)
Female H9 (WA09) hESCs were cultured on six-well plates pre-coated with hESC-qualified Matrigel matrix (Corning, catalog no. 354277). The cells were maintained in mTeSR media (Stem Cell Technologies, catalog no. 85850), which was replenished daily. For routine passaging every 4-6 days, hESC were diluted at ratios between 1:6 and 1:50 using 0.5 mM EDTA-PBS (Thermo Fisher Scientific, catalog no. 15575020). All experiments involving human embryonic stem cells in this study were reviewed and approved by the Institutional Review Board of Guangzhou National Laboratory.

### Generation of CRISPR/Cas9 genome-edited cell lines
For the deletion of specific regions in H9 hESCs via CRISPR/Cas9, six guide RNA (gRNA) sequences targeting the hEC1 and hEC2 regions (approximately 9 kb) were designed based on sequences from the Ensembl Genome Browser (Supplementary Fig. 4a). The gRNA sequences were as follows: g2 - CCAGACCCCTATGGAGCACGGGG, g4 - GCGGATCCAAGCTCACCTCAGG, g5 - CGATCTTAGTGCCTTCACCGCGG, g6 - GGGGTATTCTGCTGGCCGTAGGG, g1 - AGGAGTGTAGGAGCCGATGGTGG, and g3 - TCTGCTGTGACTCACGGTTGAGG. Confluent hESC colonies were dissociated into a single-cell suspension, and $3.0 \times 10^6$ cells were electroporated using the P3 Primary Cell 4D-Nucleofector L kit and 4D-Nucleofector system (Lonza Group Ltd., Basel, Switzerland), following the manufacturer's protocol (1320 v, 30 ms, 2 pulses). Post-transfection, the cells were plated in 6-well plates using mTeSR media supplemented with 10 μM Y-27632 (Stem Cell Technologies, catalog no. 72304) for 48 hours. Cells with high

viability were then selected in media containing puromycin (1 μg/ml) for 2-3 days. Drug-resistant wells were subsequently processed for single-cell isolation using Accutase (Sigma-Aldrich, catalog no. A6964), diluted, and re-plated into a 96-well plate. Pool cleavage efficiency was assessed using the EZ-editor Genotype Analysis System (UBIGENE). After two weeks, colonies were picked, split, and screened for enhancer deletion using specifically designed primers (Supplementary Fig. 4a and Supplementary Data 8). Positive clones with confirmed homozygous deletions were expanded and prepared for cryopreservation for downstream analysis.

## hCNCCs and PA-like hCNCCs derivation and culture

hESCs were differentiated into cranial neural crest cells (hCNCCs) following a previously described protocol with minor modifications[31]. Initially, confluent hESC colonies were treated with 2 mg/mL collagenase (Gibco, catalog no. 17104019) for 30-60 minutes until colony edges began to lift. These detached colonies were then cultured in neural crest cell induction (NCCI) medium, forming clusters of 100-200 cells in ultra-low attachment petri dishes. The NCCI medium comprised a 1:1 mixture of DMEM F-12 medium (Thermo Fisher Scientific, no. 10565018) and Neurobasal medium (Thermo Fisher Scientific, no. 21103049), supplemented with 0.5× N2 (Thermo Fisher Scientific, no. 17502048), 0.5× B27 (Thermo Fisher Scientific, no. 17504044), 20 ng/mL bFGF (Peprotech, catalog no. AF-100-18B), 20 ng/mL EGF (Peprotech, catalog no. AF-100-15), and 5 μg/mL bovine insulin (Gemini Bio, catalog no. 700-112 P). After about 24 hours, embryoid bodies (EBs) formed and were maintained with bi-daily changes of NCCI medium. On day 4, EBs were transferred to new tissue culture-treated dishes for approximately three days to initiate sphere attachment. Following attachment, media was changed daily, allowing cranial neural crest cells to migrate from the neural rosettes over about four days. Neuroepithelial spheres were then manually removed, and the remaining neural crest cells were harvested and enriched using CD271 MicroBead Kit (Miltenyi Biotec, catalog no. 130-099-023), and passaged onto fibronectin (7.5 μg/mL)-coated plates (Millipore, catalog no. FC010-100MG) in early hCNCCs medium. This medium included a 1:1 DMEM F-12/Neurobasal mixture, 0.5× N2 supplement, 0.5× B27 supplement, 20 ng/mL bFGF, 20 ng/mL EGF, 1 mg/mL BSA (Gemini Bio, catalog no. 700–104 P), and 1× antibiotic-antimycotic (Gibco, catalog no. 15240062). After two additional passages, the medium was changed to late hCNCCs medium by supplementing the early hCNCCs medium with 3 μM CHIR 99021 (Selleckchem, catalog no. S2924) and 50 pg/mL BMP2 (Peprotech, catalog no. AF-120-02) to enhance cell proliferation and reduce migration[31]. Following two more passages, late-stage hCNCCs were induced to transition into pharyngeal arch (PA)-like cell states by adding 100 nM retinoic acid (RA) (Sigma-Aldrich, catalog no. R2625) to the late hCNCCs medium for additional two passages.

## Immunofluorescence analysis

hCNCCs were cultured on fibronectin-coated (7.5 μg/mL) coverslips placed in 6-well plates. Upon reaching confluency, cells were fixed with 4% paraformaldehyde (PFA) for 10 minutes at room temperature and permeabilized with 0.1% Triton X-100 in PBS for 15 minutes. The cells were then incubated with primary antibodies targeting p75 (Abcam, catalog no. ab245134), NR2F1 (Abcam, catalog no. ab181137), AP2 (Santa Cruz Biotechnology, catalog no. sc-12726), and SOX9 (Millipore, catalog no. AB5535), each diluted in blocking buffer, and left overnight. After washing thrice for 5 minutes each with PBS, the cells were incubated for 1 hour in the dark with anti-mouse Alexa Fluor 488 antibody (Thermo Fisher Scientific, catalog no. A28175) and anti-rabbit Alexa Fluor 594 antibody (Thermo Fisher Scientific, catalog no. A-11012). Following another series of three 5-minute washes in PBS, the coverslips were mounted on slides using DAPI Fluoromount-G (SouthernBiotech, catalog no. 0100-20). Imaging was performed on a Carl Zeiss LSM 980 microscope, and the acquired images were processed using ImageJ software.

## RNA extraction, cDNA preparation, and quantitative real-time PCR

For each of the three independent hCNCCs differentiation experiments, approximately $2×10^6$ cells per well were washed with cold PBS and lysed using Buffer RLT (Qiagen, catalog no. 79216). RNA extraction was performed according to the manufacturer's protocol (RNeasy Mini Kit, Qiagen, catalog no. 74104). Subsequently, 1-2 μg of total RNA from each sample was reverse transcribed to cDNA using RevertAid H Minus Reverse Transcriptase (Thermo Fisher Scientific, catalog no. EP0452). The resulting cDNA samples were diluted, and qRT-PCR was conducted on a Bio-Rad CFX96 real-time PCR machine using SYBR Green PCR Master Mix (Thermo Fisher Scientific, catalog no. 4309155), following the manufacturer's instructions. The delta-delta Ct method was employed to calculate the relative expression levels.

## Promoter-Capture Hi-C in late hCNCCs

The preparation of Promoter Capture Hi-C libraries was conducted following established protocols[79,80]. Cells were first fixed with 2% formaldehyde for 10 minutes at ambient temperature, followed by quenching with 0.2 M glycine for 5 minutes. Post-fixation, cells were lysed, and endogenous nucleases were inactivated with 0.3% SDS. Chromatin was digested using 100U of HindIII (New England Biolabs, catalog no. R0104L), labeled with Biotin-14-dCTP (Invitrogen, catalog no. 19518018), and then ligated using 50U of T4 DNA ligase (New England Biolabs, catalog no. M0202L). After reversing the cross-links, DNA was isolated using the QIAamp DNA Mini Kit (Qiagen, catalog no. 51306), in strict accordance with the manufacturer's instructions. Subsequent steps involved fragmenting the DNA to 300-600 bp, end-repair, A-tailing, and adapter ligation using SureSelect adaptors. Biotinylated fragments were isolated through streptavidin-affinity pull-down and PCR amplified. Promoter-specific Capture Hi-C was performed using the SureSelect XT Library Prep Kit ILM (Agilent Technologies), which included a custom-designed biotinylated RNA bait library and paired-end blockers. Post-capture PCR enrichment was followed by final purification of the libraries with AMPure XP Beads (Beckman Coulter, catalog no. A63881). Library quality was evaluated using Bioanalyzer profiles (Agilent Technologies), and high-throughput sequencing was performed on the Illumina NovaSeq 6000 platform.

## Luciferase assay

The luciferase assay was conducted as previously described[28]. In the case of hESCs, small clusters of 4-10 cells each were passaged into a 24-well plate with ROCK inhibitor (Y27632), and transfection was performed the following day. For pharyngeal arch-like hCNCCs, cells were transfected two hours post-seeding at approximately 160,000 cells per well in 24-well plates. Each assay consisted of four technical replicates. The transfection mix included 0.5 μg of pGL3-enhancer plasmids with firefly luciferase gene sequences driven by an SV40 promoter (Promega, catalog no. E1761), 0.02 μg of pRL-CMV Renilla luciferase (Promega, catalog no. E2261) as an internal control, and 1.5 μL of FuGENE HD (Promega, catalog no. E2691), all diluted in Opti-MEM I Reduced-Serum Medium (Gibco, catalog no. 31985062). The enhancer sequences were cloned into the cut site between KpnI and XhoI, and relevant plasmid construction primers are listed in Supplementary Data 8.

Twenty-four hours post-transfection, cells were harvested for measurement using the Dual-Luciferase Assay System (Promega, catalog no. E4550), following the manufacturer's instructions, on a CLARIOstar Plus machine (BMG LABTECH). Two or three independent biological replicates were conducted, and the relative firefly luciferase activity was calculated by normalizing the firefly signal values to the renilla signal values across the reaction period.

## Enhancer-reporter assays in mouse embryos

Human enhancer sequences (hEC1-chr4: 8700174-8700974; hEC2.1-chr4: 8705075-8706475; hEC2-chr4: 8705075-8707510; hEC3-chr4: 8725074-8725674, hg38 version) were synthesized or PCR-amplified from H9 cell line genomic DNA. Combinations of these sequences, including (hEC1 + hEC2) and (hEC1 + hEC2 + hEC3), along with deletion tiling variants of hEC1 (ΔD1-ΔD5), were assembled using NEBulider HiFi DNA Assembly Mix (New England Biolabs, catalog no. E2621L). These constructs were then cloned into PCR4-Shh:LacZ-H11 plasmids (Addgene plasmids #139098), placing them upstream of a minimal *Shh* promoter and LacZ reporter gene. Similarly, the orthologous mouse enhancer, mEC1 (chr5: 35623357-35624110, mm39 version), was amplified from C57BL/6 J mouse genomic DNA and cloned using the same method. The detailed primers used are in the Supplementary Data 8.

CRISPR/Cas9 microinjection protocol was employed to site-specifically integrate these enhancer sequences into the H11 locus of the FVB mouse strain, which served as surrogate mothers. $F_0$ embryos were harvested at developmental stages E9.5, E11.5, and E14.5. Only embryos carrying the transgenic reporter at the H11 locus were processed further for LacZ staining. Genotyping primers for each construct are listed in Supplementary Data 8. Whole mouse embryo LacZ staining was performed as previously established protocols[81]. For each enhancer and developmental time point, a minimum of three independent positive embryos were analyzed. Imaging was conducted using a Carl Zeiss Axio Zoom V16 microscope.

## Enhancer DNA pull-down assay

Biotin-labeled enhancer DNA sequences were prepared via PCR and verified through 1% agarose gel electrophoresis. Approximately 50 mouse embryos at E11.5 were microdissected to isolate PA2 tissues, which were then washed with cold PBS and homogenized using a Dounce homogenizer. Nuclear proteins were extracted using NE-PER nuclear and cytoplasmic extraction reagents (Thermo Fisher Scientific, catalog no. 78833), following the manufacturer's protocol. Protein concentrations were quantified using BCA protein assay kits (Thermo Fisher Scientific, catalog no. 23225).

The DNA probe and nuclear proteins were pre-mixed and incubated with Bioeast Mag-SA (Streptavidin, SA, Bioeast, catalog no. M2800S) for 1 hour at 4 °C. Negative control is a mix of nuclear protein and Bioeast Mag-SA, sample is a mix of Biotin-11-dUTP labeled probe, nuclear protein and Bioeast Mag-SA, and we also include a probe control with unlabeled probe. The magnetic beads were then isolated using a magnetic stand and washed thrice with cold PBS. The eluted proteins were analyzed by SDS-PAGE and silver staining to confirm binding. Finally, protein identification was performed using liquid chromatography-mass spectrometry (LC-MS).

## CUT&RUN experiment in hCNCCs

The CUT&RUN assay was conducted in hCNCCs using the ChIC/CUT&RUN Kit (Epicypher, catalog no. 14-1048), incorporating several modifications from previously reported methods. Approximately 500,000 cells were harvested using Accutase and subsequently incubated with activated ConA Beads for 10 minutes at room temperature. Primary antibodies were then added to each 50 μL reaction: IgG (Epicypher, catalog no. 13-0042k, 0.5 μg), H3K27ac (Active Motif, catalog no. 39133, 0.5 μg), TCF7L2 (Cell Signaling, catalog no. 2569S, 1 μg), and the mixture was incubated overnight on a nutator at 4 °C.

On the second day, pAG-MNase and cold calcium chloride were used to activate the cleavage of target DNA. E. coli Spike-in DNA (0.1 ng) was also added for normalization in downstream analyses. DNA purification involved phenol/chloroform extraction and ethanol precipitation. Library preparation for histone proteins was performed using the CUT&RUN Library Prep Kit (Epicypher, catalog no. 14-1001). For TFs, modified procedures, as previously reported (https://doi.org/ 10.17504/protocols.io.bagaibse), were employed to enhance the preservation of small DNA fragments. Sequencing was carried out using paired-end reads (2 × 150 bp) on the Illumina NovaSeq 6000 platform.

## Expression of recombinant proteins and electrophoretic mobility shift assays (EMSA)

The coding sequences for MEIS1 protein was cloned into the pET-His6-MBP-TEV-LIC vector between BamHI and XhoI sites, incorporating a tobacco etch virus (TEV) protease cleavage site (ENLYFQ | G) between the maltose-binding protein (MBP) and the MEIS1 protein. BL21 (DE3) competent cells were transformed with this construct. Positive bacterial cultures were grown at 37 °C until the optical density at 600 nm (OD600) reached 0.5 and then induced with 0.2 mM isopropyl β-d-1-thiogalactopyranoside (IPTG). Bacterial cells were lysed by sonication in buffer A (50 mM Tris-HCl, 500 mM NaCl, 10 mM 2-mercaptoethanol, 10% (v/v) glycerol, pH 7.5) and centrifuged to collect the supernatant. This supernatant was then applied to a Ni-NTA Superflow Cartridge (Sangon, catalog no. C600792), washed with buffer A supplemented with 50 mM imidazole, and eluted with an imidazole gradient from 100 mM to 500 mM in buffer A. The eluted protein was treated with TEV protease and dialyzed overnight in buffer A at 4 °C. The protein mixture was re-applied to a Ni-NTA Superflow Cartridge and eluted with buffer B (50 mM Tris-HCl, 1 M NaCl, 10 mM 2-mercaptoethanol, 10% (v/v) glycerol, pH 8.0) containing 50 mM imidazole. Finally, the MBP-free protein was further purified using size-exclusion chromatography (Superdex 200 Increase 10/300 GL; Cytiva, catalog no. 28990944) and concentrated to approximately 1 mg/mL in buffer C (50 mM Tris, 200 mM NaCl, 10 mM 2-mercaptoethanol, 10% (v/v) glycerol, pH 7.5).

EMSA was performed using the LightShift Chemiluminescent EMSA kit (Thermo Fisher Scientific, catalog no. 20148) with the purified TF protein. Biotin-labeled probe (Supplementary Data 9) containing predicted MEIS1-binding DNA sequence was synthesized and incubated with the MEIS1 following the manufacturer's protocol. The resultant gels were visualized using a Tanon 5200 system.

## Mice and ethical statement

All animal experiments were performed under the approval of the Animal Care Committee at Beihang University (BM20210057). The breeding protocol was conducted as follows: one male mouse was paired with 1-2 female mice per cage for mating, with daily vaginal plug checks. Noon on the day of plug detection was considered the embryonic developmental time point (E0.5). All mice used for breeding met the following criteria: minimum age is more than 8 weeks, maximum age is less than 6 months. All mice were housed in a temperature-controlled environment (20 ± 2 °C) with regulated humidity (50-60%) and standard 12-hour light/dark photoperiod. Animals had continuous access to food and water throughout the experimental period.

No sex- and gender-based analyses have been performed, as patients with bilateral constricted ear (BCE) show no sex differences, and gender differences in our mouse model do not impact disease occurrence.

## Generation of genome-edited mouse models via CRISPR/Cas9

This study utilized four distinct mouse models constructed using the CRISPR/Cas9 technique: mEC1 deletion (*mEC1^del/+*), mEC1 duplication (*mEC1^dup/+*), *Hmx1^fl/+*, *Hmx1^wt/EGFP*. gRNAs were designed using CHOPCHOP to target specific regions: *mEC1^del/+*: AGGACAGTCTCCAAGTCCGG-TGG; *mEC1^dup/+*: TGTGACGTGATCGACTCCAT-CGG; *Hmx1^fl/+*: gRNA1- CCCA-GATTCAGGGCGTACAA-GGG, gRNA2- ACTGACTTGTTCCTACCTAC-AGG; *Hmx1^wt/EGFP*: gRNA1-ATGCCGGGGGCTAGTGTGAGC-CGG, gRNA2-GCGCCGGCTCACACTAGCCC-CGG. A combination of sgRNAs, Cas9 protein, and donor vectors were co-injected into fertilized mouse embryos at the single-cell stage. F0 founder mice were identified via PCR and confirmed by Sanger sequencing. These founders were bred with

wild-type mice for germline transmission testing and F1 heterozygous mouse generation. Different homozygous genotypes were subsequently produced by intercrossing heterozygous mice.

The generation of *dumbo* mouse lines has been previously described[14]. *Wnt1::Cre;Hmx1^{fl/fl}* mice were generated by crossing *Wnt1::Cre;Hmx1^{fl/+}* male mice with *Hmx1^{fl/fl}* female mice, and the offspring were validated using PCR. All mice were maintained on a C57BL/6J genetic background, housed in a Specific Pathogen Free (SPF) environment, and handled according to ethical standards. These mouse lines are available upon request from the corresponding author.

### Sample preparation, RNA isolation and bulk RNA-seq library preparation

Forming pinna prominences from E14.5 embryos (both wild type and *mEC1^{dup/dup}*) were micro-dissected, washed with cold 1× PBS, and homogenized using a Tissue-Tearor. Total RNA extraction followed the same protocol as for cell RNA extraction. RNA-seq libraries were then prepared using the Illumina TruSeq RNA Library Prep Kit v2 (Illumina, catalog no. RS-122-2001), adhering to the manufacturer's instructions. For each genotype, five independent biological replicates were processed.

### Sample preparation, RNA isolation and single cell RNA-seq library preparation

Single cell RNA-seq samples were prepared as previously described[29]. Briefly, micro-dissected forming pinna prominences from *Hmx1^{wt/EGFP}* embryos were enzymatically dissociated using 0.5% trypsin/1× EDTA for 10 minutes at 37 °C (for E10.5 and E12.5 stages) and papain digestion mix for 7 minutes (for E14.5 stage). The treated tissues were then rinsed in ice-cold 1× DMEM, filtered, and GFP⁺ cells were enriched using FACS (Sony, MA900). These collected cells were resuspended and diluted in 1× DPBS (Cytiva, catalog no. SH30028.FS) to a concentration of $1 × 10^6$ cells per ml. Gene expression libraries were prepared using GEXSCOPE Single Cell RNA Library Kits v.2 (Singleron, catalog no. SD-4180022). The single-cell suspensions were loaded onto the GEXSCOPE microchip, which facilitates automated single cell capture, cell lysis, cellular mRNA capture, and molecular labeling via the Singleron Matrix instrument. The barcoded cDNA was amplified and utilized for constructing single cell NGS libraries. The quality and quantity of the resulting libraries were assessed using Qubit 4.0 (Thermo Fisher Scientific) and Qseq100 (Bioptic). For gene expression analysis, samples were sequenced on an Illumina NovaSeq platform to a depth of 500 million reads per channel.

### Whole-mount in situ hybridization (WISH)

The WISH procedure was carried out on whole-mount embryos according to previously established protocols[29]. Initially, mouse embryos were carefully dissected to remove extraembryonic membranes and then immersed in cold 1× PBS. For fixation, embryos were submerged in 4% paraformaldehyde in 1× PBS and gently rocked for 6 hours or overnight at 4 °C in a sealed vial. Post-fixation, embryos were thoroughly washed three times with PBST (1× PBS with 0.1% Tween-20), dehydrated through a graded series of methanol-PBST solutions, and finally placed in 100% methanol for storage at −20 °C until they were processed further.

On the day of hybridization, the embryos were first rehydrated and then washed twice for 5 minutes each in PBST. They were incubated for 1 hour in 6% hydrogen peroxide/PBST and subsequently washed again with PBST at 4 °C. Next, the embryos were treated with 10 µg/mL proteinase K (Roche, catalog no. 3115887001) in PBST for a duration of 10 to 30 minutes, varying according to different embryonic stages, at room temperature. The proteinase K reaction was halted by adding 2 mg/mL glycine in PBST. For pre-hybridization, embryos were incubated for 1 hour at 68 °C in hybridization buffer, which comprised 50% deionized formamide, 5× SSC buffer, 1% SDS, 100 µg/mL yeast tRNA, and 50 µg/mL heparin. After this, digoxigenin-labeled probes

(concentration ranging from 0.5 to 1 µg/mL) were added ((Supplementary Data 9)), and the embryos were incubated overnight at 68 °C.

On the following day, embryos underwent a series of extensive washes. They were initially washed three times for 30 minutes each in Wash 1 (50% regular formamide, 5× SSC buffer, 1% SDS). This was followed by a 10-minute wash in a 1:1 mixture of Wash 1 and Wash 2 (0.5 M NaCl, 10 mM Tris HCl pH 7.5, 0.1% Tween-20), and then two 30-minute washes in Wash 2 containing 100 µg/mL RNaseA and 100 units/mL RNase T1 at 37 °C. Further, embryos were washed three times for 30 minutes each in Wash 3 (50% regular formamide, 2× SSC buffer) at 65 °C, and then three times for 10 minutes each in TBST (140 mM NaCl, 2.5 mM NaCl, 25 mM Tris pH7.5, 0.1% Tween-20) at room temperature. The embryos were then blocked for 2.5 hours at room temperature in block buffer (1% blocking reagent in TBST) and incubated overnight at 4 °C with anti-Digoxigenin-AP-conjugated antibody (dilution range: 1:1000 to 1:5000, Roche, catalog no. 11093274910) diluted in the same buffer.

On the third day, the embryos were subjected to five 10-minute washes, followed by five 60-minute washes in TBST, and then left to wash overnight at 4 °C. Lastly, the embryos were washed twice for 20 minutes each in NTMT buffer (100 mM NaCl, 100 mM Tris pH 9.5, 50 mM MgCl2, 0.1% Tween-20) at room temperature. The color reaction was developed using NBT/BCIP (Roche, catalog no. 11681451001) in NTMT buffer for 0.5 to 2 hours in the dark. The reaction was stopped with three 10-minute washes in PBST, and the embryos were subsequently photographed using a Carl Zeiss Axio Zoom V16 microscope, following the manufacturer's instructions. Two or three independent biological embryos for each genotype were conducted.

### Frozen section in situ hybridizations

Mouse embryos (wild type and *mEC1^{dup/dup}*) at E14.5 were dissected and washed with cold 1× PBS before being fixed in 4% paraformaldehyde (PFA). The embryos underwent a dehydration process using a series of sucrose/PBS solutions (10%, 20%, 30%). This was followed by an equal volume mixture of OCT compound, with rotation for 30 minutes at room temperature. Embryos were then embedded in OCT on dry ice and stored at −80 °C until required. Cryostat sections of 20 µm thickness were prepared in a horizontal orientation using a Leica CM1860. Before hybridization, frozen sections were thawed to room temperature and allowed to air dry for at least one hour. They were then re-fixed using 4% PFA/PBS for 15 minutes, followed by three 5-minute washes in 1× PBS. Acetylation was carried out in an acetic anhydride solution (1.35% (v/v) triethanolamine, 0.175% (v/v) HCl, 0.25% (v/v) acetic anhydride) for 10 minutes at room temperature, and the sections were subsequently washed three times for 5 minutes each in 1× PBS. Prehybridization was carried out at room temperature for 2 hours. Digoxigenin (dig)-labeled probes were diluted to 1 µg/mL in hybridization buffer (50% deionized formamide, 5× SSC buffer, 1% SDS, 100 µg/mL yeast tRNA, 50 µg/mL heparin), then heated to 80 °C for 5 minutes. Approximately 400 µL of hybridization buffer was applied to each slide, covered with parafilm, and hybridized at 70 °C for 16-18 hours. On the second day, the parafilm was removed by immersing slides in 5× SSC. This was followed by a series of washes: 30 minutes in 5× SSC/50% formamide, 20 minutes in 2× SSC, and twice for 20 minutes each in 0.2× SSC at 65 °C. Subsequently, the sections were washed three times for 5 minutes each in MABT buffer at room temperature. Blocking was performed for 1-2 hours in 2% Blocking reagent/MABT buffer. Anti-Dig-AP diluted at 1:2000 in blocking buffer was applied to the sections, which were then incubated overnight at 4 °C. On day 3, the sections were washed eight times for 5 minutes each in MABT, twice for 10 minutes each in NTMT buffer, followed by application of BM purple (Roche, catalog no. 11442074001) in the dark for 2-6 hours, depending on the signal development. The reaction was terminated by immersing the slides in MABT for two 10-minute washes, then post-fixing in 4% PFA/PBS for 30 minutes, and washing three times for 5 minutes each in 1× PBS. The sections were mounted with VectaMount

AQ aqueous mounting medium (Vector Laboratories, catalog no. H-5501-60). Imaging was performed using a LEICA Aperio Versa 8 microscope, and images were processed with ImageScope x64 software (version 12.4.6.5003). Two independent biological embryos for each genotype were conducted.

## Micro-computed tomography (Micro-CT) imaging and quantitative analysis of ear structures

Mice were euthanized and whole heads imaged using a SkyScan 1276 micro-computed tomograph (Bruker, Belgium). Scanning was performed at 18-micron resolution using the following parameters: 55 kV, 200 uA, 0.5 mm Al filter; 0.4° rotation step over 360°, with 3-frame averaging. All raw scan data were reconstructed into multiplanar slice data using NRecon V1.7.3.1 software (Bruker). Reconstructed data were then rendered in 3D with consistent thresholding parameters using Drishti V3.0 Volume Exploration software (GitHub.com/nci/Drishti) for gross visual assessment of the craniofacial skeleton. Representative rendered images from animals of each genotype were captured and processed using Photoshop (Adobe Creative Cloud).

For quantitative assessment of affected structures, three distinct interlandmark measurements were taken from each scan (as shown in Supplementary Fig. 8). The measurements included: 1) the exterior or posterior length of the paroccipital process (as seen from the posterolateral view of the process, 2) the interior or anterior length of the paroccipital process (as viewed after placement of a clipping plane to remove bone obstructing a posterior-facing view of the processes), and 3) the height of the posterior aspect of the periotic capsule, which resides adjacent the paroccipital process. All measurements were taken for both the left and right side structures of each skull. All data were graphed using SuperPlotsofData shiny app (https://huygens.science.uva.nl/SuperPlotsOfData/).

## Computational analysis

### RNA-seq data and GO enrichment analysis.
RNA sequencing reads were initially processed using fastp for trimming of the paired-end data, with quality assessment performed via FASTQC (version 0.12.1). Subsequently, the reads were aligned to the mouse reference genome (GRCm39) using the STAR aligner (version v2.7.11a). Quantification of the alignment files was carried out using featureCounts (version 2.0.6), incorporating the gene model parameter. Differential gene expression analysis was conducted utilizing the limma (version 3.56.2), Glimma (version 2.10.0), and edgeR (version 3.42.4) packages. Significantly downregulated genes, defined by criteria of an FDR ≤ 0.05, log fold change (logFC) < 0, and log counts per million (logCPM) ≥ 1, were selected for Gene Ontology (GO) enrichment analysis. This analysis was performed using the clusterProfiler package (version 4.8.3), with results visualized via the treeplot function.

### ChIP-seq and CUT&RUN data analysis.
Publicly available ChIP-seq datasets for this study were obtained from the GEO database, including datasets for d11hCNCC: GSE28874[82]; P4hCNCC: GSE145327[28] and GSE70751[31]; mNCC: GSE89435[20]; Pinna: GSE211900[29]. Sequencing reads were trimmed using fastp and aligned to either the mouse (GRCm39) or human (GRCh38) reference genome using bowtie2 (version 2.4.0). Duplicate reads (marked by Picard toolkits), non-unique, and low mapping quality reads were filtered out using sambamba. BigWig files, with signals normalized using the RPGC method, were generated for visualization using deepTools2.0 (version 3.4.2). Peak calling against the INPUT sample was performed using Macs2 software (version 2.2.7.1). Additionally, human embryonic craniofacial epigenetic data from stages CS13, CS14, CS15, and CS17 were accessed from https://cotney.research.uchc.edu/craniofacial/[27].

CUT&RUN sequencing data were analyzed employing the CUT&RUNTools 2.0 pipeline, as previously reported[83]. For histone peak calling, the fragment size filter was deactivated, while for TFs, fragments larger than 120 bp were filtered out. Normalized bigwig files were generated using an alignment scale factor based on spike-in reads from *E. coli* K12 strain MG1655. Two replicates for independent biological experiments were generated for each antibody (IgG, H3K27ac, and TCF7L2).

### ATAC-seq data analysis.
ATAC-seq datasets used in this study were sourced from existing databases (d11hCNCC: GSE108517[30]; hESC and P4hCNCC: GSE145327[28]; mNCC: GSE89436[20]; Pinna: GSE211899[29]). The adapter sequences were trimmed using the bbduk.sh script and the reads were subsequently aligned to the reference genome using bowtie2. Low-quality reads were filtered out following the methodology described in the ChIP-seq data analysis section. The alignmentSieve function with the –ATACshift parameter was employed to adjust for the transposase binding pattern. For visualization purposes, bigwig files were generated utilizing the bamCoverage function in deepTools 2.0 (version 3.5.2).

### Hi-C and PCHi-C data analysis.
For the capture Hi-C data of human cranial neural crest cells (hCNCCs), a genome digest file for HindIII was created using hicup_digester. The HiCUP software (version 0.9.2) was then employed to generate alignment files. To prepare files for the Chicago R package (version 1.30.0), scripts such as bam2chicago.sh and makeDesignFiles.py from chicagoTools were utilized. The Chicago package produced washU_text format results, showcasing interactions between two genomic positions. Interactions with a score ≥5 were considered high-confidence and visualized as arcs using pygenometrack (version 3.9). Other PCHi-C datasets in this study (hESC: GSE8682[84]; Pinna: GSE211901[29]) were downloaded and processed as same as hCNCCs.

For the human eye Hi-C dataset, we obtained the interaction heatmap profile from the Epigenome Gateway website (Hub: HiC interaction from Juicebox, Track label: 3PNAS_2016_1RPE1_control_HIC005[25] of human eye). The file (GSM1847524) was converted to cool format using hicConverFormat function in HiCExplorer (version 3.7.2) with –resolutions 25000, then hicFindTADs was applied for calling TAD regions with parameters: -thresholdComparisons 0.05, -delta 0.01, -correctForMultipleTesting fdr. CTCF signals of hCNCCs were further integrated to display sub-TAD structure.

### scRNA-seq analysis.
The scRNA-seq reads were processed using the CeleScope pipeline, details of which are available at https://github.com/singleron-RD/CeleScope. This processing generated a 10x Genomics-like raw feature-barcode matrix suitable for downstream filtering and data integration using Seurat (version 4.3.0). For cell filtering, we assumed that both the log10-transformed gene counts and UMI counts per cell for each sample followed a normal distribution. We calculated the mean ± 1.96 times the standard deviation, defining the 95% confidence interval, to determine the lower and upper thresholds. These thresholds were then converted back to the original scale (as the power of 10 of the log values) to establish criteria for excluding low-quality cells. Furthermore, cells exhibiting a mitochondrial gene expression ratio exceeding 5% were also categorized as low quality.

Data integration was accomplished using the CCA2-based pipeline of the Seurat package. The SelectIntegrationFeatures function was utilized to identify variable features across datasets for integration. Subsequently, the FindIntegrationAnchors function was employed to identify anchor pairs between datasets, followed by the IntegrateData function to create an integrated assay, which included the newly integrated expression profiles. Based on the results of integration, we performed PCA dimensionality reduction using RunPCA on the variable features employed for integration. This was followed by UMAP dimensionality reduction using RunUMAP with parameters *-dims 1:20, -n.neighbors 50*. For unsupervised clustering, we executed the

FindNeighbors function using default parameters, and FindClusters with -resolution 0.4.

**PAGA developmental trajectory analysis.** The integrated expression profiles were imported into SCANPY (version 1.9.3) for the construction of the fibroblast differentiation trajectory employing the Partition-based Graph Abstraction (PAGA) algorithm. For this analysis, a total of 4994 cells, equally proportioned from each sample and cell type, were subsetted for trajectory construction. The scanpy.tl.paga function was utilized to construct a PAGA graph, which facilitated the visualization of cell population structures. This visualization was achieved using sc.tl.draw_graph, producing a force-directed graph drawing. The root cell for the trajectory was designated as a cell from the Mitotic CNCC-derived mesenchymal cell population. Diffusion pseudotime was calculated using scanpy.tl.diffmap and scanpy.tl.dpt functions. Additionally, the cellrank.kernels.PseudotimeKernel function from the CellRank package (version 2.0.2) was employed to construct the cellrank pseudotime kernel. This kernel was crucial for computing the cell-cell transition matrix. The pseudotime results were ultimately visualized as embeddings on the force-directed graph, providing insights into the developmental trajectory of fibroblast differentiation.

**Motif analysis.** Motif prediction on hEC1 sequence was performed using FIMO software in the MEME Suite (version 5.5.3). The motifs used in the prediction process were downloaded from HOCOMOCO database (v12) containing a collection of 1443 motifs, and the threshold $P$-value of 1E-4 was used to identify high-affinity binding sites. After the preliminary results, we further used the single cell expression datasets from human embryos (GSE157329) to filter TFs that are not expressed (TFs must be expressed in cells expressing the *HMX1* gene in no less than 10% of the cells) with the *HMX1* gene. Additionally, through the ALX1/4 prediction $P$-value is larger than 1E-4, we also present it in Fig. 3 considering the very close distance to the very high MEIS1/PBX1 binding site and potential influence. Finally, we further confirm the binding pattern of HD, HMG-box, and Coordinator in the D3-D4 region using sequence motif location tool (MoLoTool).

**EChO analysis.** The Enhanced Chromatin Occupancy (EChO) analysis was conducted following previously established protocols[34,69]. The process began with the generation of two bed files using bedtools: one representing the region file (TCF7L2 occupied peaks on hEC1 extended by ±500 bp) and the other being the fragment file. Detailed bedtools procedures for EChO can be found at https://github.com/FredHutch/EChO. Utilizing the region file, the foci mode of EChO was implemented to identify single-base-pair foci as the centers of matrices. Additionally, the matrix mode was executed over a 400 bp window for each focus. This approach generated a matrix of base pair-resolution EChO fragment size values, facilitating subsequent visualization. For the analysis of high and low motif scores, TCF7L2 matrix profiles (including MA0523.1 and MA0523.2,) were retrieved from the JASPAR database. These profiles were scanned across a 40 bp window surrounding each focus. Motifs with scores exceeding 80% were categorized as high-scoring, while those above 70% were deemed low-scoring motifs.

**Proteomic analysis of LS/MS results.** Raw MS files were converted to MGF format using MM File Conversion software and subsequently analyzed via MASCOT software (version 5.5.3) against the UniProt database (https://www.uniprot.org/taxonomy/10090). The search parameters are: Fixed modifications--Carbamidomethyl, Variable modifications--Oxidation, Enzyme--Trypsin, Maximum Missed Cleavages--1, Peptide Mass Tolerance--20ppm, Fragment Mass Tolerance--0.6 Da, Mass values--Monoisotopic, Significance threshold--0.05. The matched proteins were filtered based on a peptide expectation value threshold (pep_expect <0.05), and the resulting protein data are listed in the Supplementary Data 3. Then, significantly identified proteins were enriched using UP_KW_MOLECULAR_FUNCTION implemented in the DAVID website (https://davidbioinformatics.nih.gov/). For fifteen DNA-binding proteins, InterPro enrichment was further performed to detect TF classification. $P$ values were corrected by Benjamini-Hochberg method.

**Reporting summary**
Further information on research design is available in the Nature Portfolio Reporting Summary linked to this article.

## Data availability
All raw sequencing data and processed data in this study have been deposited in the NCBI Gene Expression Omnibus (GEO) database under accession code GSE263084 (CUT&RUN, https://www.ncbi.nlm.nih.gov/geo/query/acc.cgi?acc=GSE263084), GSE263085 (PC-HiC, https://www.ncbi.nlm.nih.gov/geo/query/acc.cgi?acc=GSE263085), GSE263086 (bulk RNA-seq, https://www.ncbi.nlm.nih.gov/geo/query/acc.cgi?acc=GSE263086), GSE263087 (scRNA-seq, https://www.ncbi.nlm.nih.gov/geo/query/acc.cgi?acc=GSE263087), GSE293403 (scRNA-seq, https://www.ncbi.nlm.nih.gov/geo/query/acc.cgi?acc=GSE293403). The mass spectrometry proteomics data have been deposited to the ProteomeXchange Consortium via the PRIDE partner repository with the dataset identifier PXD063373. The raw genomic data are available under restricted access for patients' confidentiality, access requires a brief project description and a signed data-use agreement restricting downstream data sharing and limiting use to the requesting investigator. In addition, we used public sequencing datasets: ChIP-seq datasets: GSE28874 (d11hCNCC, https://www.ncbi.nlm.nih.gov/geo/query/acc.cgi?acc=GSE28874), GSE145327 (P4hCNCC, https://www.ncbi.nlm.nih.gov/geo/query/acc.cgi?acc=GSE145327), GSE70751 (P4hCNCC, https://www.ncbi.nlm.nih.gov/geo/query/acc.cgi?accacc=GSE70751), GSE89435 (mCNCC, https://www.ncbi.nlm.nih.gov/geo/query/acc.cgi?acc=GSE89435), GSE211900 (Pinna, https://www.ncbi.nlm.nih.gov/geo/query/acc.cgi?acc=GSE211900). ATAC-seq datasets: GSE108517 (d11hCNCC, https://www.ncbi.nlm.nih.gov/geo/query/acc.cgi?acc=GSE108517), GSE145327 (hESC and P4hCNCC, https://www.ncbi.nlm.nih.gov/geo/query/acc.cgi?acc=GSE145327), GSE89436 (mCNCC, https://www.ncbi.nlm.nih.gov/geo/query/acc.cgi?acc=GSE89436), GSE211899 (Pinna, https://www.ncbi.nlm.nih.gov/geo/query/acc.cgi?acc=GSE211899). scRNA-seq dataset: GSE157329 (single-cell transcriptomes of 4- to 6-week human embryos, https://www.ncbi.nlm.nih.gov/geo/query/acc.cgi?acc=GSE157329). Additionally, human embryonic craniofacial epigenetic data from stages CS13, CS14, CS15, and CS17 were accessed from https://cotney.research.uchc.edu/craniofacial/. Source data are provided with this paper.

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

## Acknowledgements

We would like to thank the patients and their families for their participation in the study. In addition, we would like to thank Weiqiang Li, Peng Xiang (from Sun Yat-sen University) and Sahin Naqvi (from Stanford University) for helping in neural crest cell culture and Kenny Zhang for helping in mouse line experiments. We also would like to thank Yanxiao Zhang (from Westlake University) for advice on our LacZ experiments. This study was supported by grants from the National Natural Science Foundation of China (32470644 and 82171844 to Y-B.Z., 82202047 to X.X., and 81671933 to J.H), China Postdoctoral Science Foundation (2022M722228 to X.X.), and the Special Research Fund for Plastic Surgery Hospital, Chinese Academy of Medical Sciences and Peking Union Medical College (YS202041 to Q.C.), the Key Research and Development Program of Guangzhou (202206060002 to X.F.), the Guangdong Provincial Pearl River Talents Program (2021QN02Y747 to X.F.) and a Stowers Family Endowment to T.C.C. The funders had no role in study design, data collection and analysis, decision to publish, or preparation of the manuscript.

## Author contributions

Y-B.Z., X.X., and H.L. designed and supervised the study. Q.C., Q.L., Jin H., B.W., Q.Z., T.L., and Y-B.Z. recruited case samples. X.X., X.F., Q.H., Z.M., H.Z., Z.L., J.Z., J.L., and Q.C. planned and conducted laboratory experiments. X.X, X.H., B.X., W.X., R.L., Z.M., X.T., J.W., Jian H., and T.C.C. analyzed data. X.X., Y-B.Z., J.Z., X.F., T.C.C., S.E.A., and H.L. drafted and revised the manuscript. All authors have reviewed and contributed to the manuscript.

## Competing interests

The authors declare no competing interests.

## Additional information

¹Guangzhou National Laboratory, Guangzhou 510320 Guangdong, China. ²School of Bioengineering Medicine, Beihang University, Beijing 100191, China. ³Bioland Laboratory, Guangzhou 510320 Guangdong, China. ⁴Department of Ear Reconstruction, Plastic Surgery Hospital, Chinese Academy of Medical Sciences, Beijing 100144, China. ⁵Departments of Oral & Craniofacial Sciences, School of Dentistry, and Pediatrics, School of Medicine, University of Missouri-Kansas City, Kansas City, USA. ⁶Division of Life Sciences and Medicine, University of Science and Technology of China (USTC), Hefei 230000, China. ⁷Department of Plastic Surgery, Affiliated Hospital of Xuzhou Medical University, Xuzhou 221000, China. ⁸Department of Ophthalmology, Daping Hospital, Army Medical University, Chongqing, China. ⁹Department of Genetic Medicine and Development, University of Geneva Medical Faculty, Geneva 1211, Switzerland. ¹⁰Medigenome, Swiss Institute of Genomic Medicine, 1207 Geneva, Switzerland. ¹¹iGE3 Institute of Genetics and Genomes in Geneva, Geneva, Switzerland. ¹²Shandong collaborative innovation research institute of traditional Chinese medicine industry, Jinan 250000 Shandong, China. ¹³The Fifth Affiliated Hospital of Guangzhou Medical University, Guangzhou 510320 Guangdong, China. ¹⁴GMU-GIBH Joint School of Life Sciences, Guangzhou 510320 Guangdong, China. ¹⁵School of Biomedical Engineering, Guangzhou Medical University, Guangzhou 510320 Guangdong, China. ¹⁶Key Laboratory of Big Data-Based Precision Medicine, Beihang University, Ministry of Industry and Information Technology, Beijing 100191, China. ¹⁷These authors contributed equally: Xiaopeng Xu, Qi Chen, Qingpei Huang, Timothy C. Cox. ✉e-mail: joycezhang1978@hotmail.com; fan_xiaoying@gzlab.ac.cn; liu_huisheng@gzlab.ac.cn; zhangyongbiao@gmail.com

