## [Transparent Peer Review file · Nature Communications]

Auricular Malformations Driven by Copy Number Variations in a Hierarchical Enhancer Cluster and Dominant Enhancer Recapitulates Human Pathogenesis

Corresponding Author: Professor Yong-Biao Zhang

Version 0:

Reviewer comments:

Reviewer #1

(Remarks to the Author)

In the study of Zhang et al. the authors tackle the genomic mechanisms and molecular/cellular defects underlying bilateral ear constriction (BCE) which represents an auricular disorder. The authors start by performing linkage analysis and target capture sequencing for fine mapping of BCE in seven Chinese families, which identifies duplications of a non-coding, CNCC enhancer-containing interval near the HMX1 transcription factor, a gene implicated in outer ear development and related malformations. The authors then use state-of-the art transgenesis and genome editing to generate advanced in vitro (human) and in vivo (mouse) models to validate and functionally dissect the enhancer modules within the identified genomic interval. The authors thereby characterize the spatiotemporal activity patterns of three enhancer modules and combinations thereof (hEC1, hEC2, hEC3) in context with transcriptional control of Hmx1 expression in CNCCs essential for auricular/pinna development. While these experiments provide novel insights into synergistic and hierarchical enhancer interactions based on transgenic reporter output, it remains to be noted that the mouse orthologous region of the human EC1 enhancer has been previously identified in Rosin et al., 2016 (PMID: 27287804). EC1 appears to be clearly the strongest enhancer, while hEC2 drives significantly more restricted expression in PA CCNCs, with additional repressive/modulating roles. The role of EC3 in this process remains less clear. The authors then focus their study on EC1 and perform an in-depth functional characterization by first addressing the functional characteristics of EC1-subregions using a tiling deletion approach (based on site-directed reporter transgenesis in mouse embryos). This allows to pinpoint functionally relevant enhancer subregions and TF motifs. The authors then generate an EC1 duplication model in mice to convincingly show that EC1 enhancer duplication is the likely mechanistic determinant underlying the BCE phenotype observed in the initially studied human pedigrees. In turn, the authors also demonstrate that EC1 deletion, as well as Wnt1::Cre-mediated conditional Hmx1 deletion, in mice leads to the BCE-opposing dumbo pinna phenotype (along with a defective paroccipital process). In a final part, the authors perform whole-mount transcriptome analysis to identify DEGs resulting from ectopically expressed Hmx1 (in muscle and epidermis tissue of the pinna) as a result of EC1 duplication. Single-cell analysis in Hmx1-P2A-GFP reporter embryos (in wildtype context) further elucidates on the role of Hmx1 in this process.

Overall, this is a comprehensive and impressive study going the whole distance from mapping a human (auricular) disorder to resolving the (regulatory) mechanism at the genomic level and thereby defining BCE as an enhanceropathy. In addition to valuable in vitro experiments on the basis of hESC-derived CNCCs, the authors made consistent use of a novel site-directed transgenic approach for more accurate enhancer-reporter transgenesis and generated four different mouse alleles using CRIPR-Cas9 to consolidate their results and to ensure high-quality in vivo data. I consider this study suitable for Nature Communications and of high interest for the readership, given the following points are taken into account:

Major points:

1. As the authors cite in the introduction, Rosin et al., 2016 (PMID: 27287804) reports the identification of a distal 594 bp ECR that specifies Hmx1 expression in pinna and lateral facial morphogenesis and is regulated by the Hox-Pbx-Meis

complex. Here, the authors reference that hEC1 maps to the “mouse 594bp ECR” (here termed mEC1) only later in the Results section. However, this circumstance should already be described in the beginning of the Results (when the hEC1 enhancer is defined) and the position of the “mouse 594bp ECR” (mEC1) should also be indicated in the locus schemes of Figures. 2a and 3a.

2. Despite the significantly improved efficiency and accuracy of enhancer-driven reporter signals in site-directed transgenic reporter assays such as enSERT, at least n=2 replicates need to be shown to corroborate a result, as a random integration event with position effects cannot be excluded in multi-copy (“tandem”) integrations. Therefore, the n=1 for hEC3 that the authors show in panel I of Figure 3 is not sufficient to demonstrate PA activity and at least n=2 embryos with overlapping signal in the respective tissue are required for a robust result.

In relation to this, is there a reason why the authors analyzed hEC3 only at E11.5? For consistency with hEC1 and hEC2 and to evaluate potential earlier/late craniofacial activities, the authors should also provide information about hEC3 activity at E9.5 and E14.5.

3. In Fig. 3 (panels h-m) it is not entirely clear how the authors extrapolate from the LacZ pattern observed in the whole mount embryos at E11.5 (images) to the regional distribution of reporter activity shown in alpha, beta and gamma subregions in the schemes. Ideally, the authors add close-ups as shown in panel G (E11.5) to better visualize staining in these areas.

4. Line 221/222: “These results were somewhat similar to those using cellular models..”. Could the authors describe better which aspects of the interacting enhancers are synergistic and which are following a hierarchical logic (as described e.g. in Long et al., 2016, PMID: 27863239)? Related to this, in line 246 the authors state: “These results indicate that spatial specificity of Hmx1 expression around PA2 is finely tuned by the coordinated and synergistic enhancers within an enhancer cluster”. Could the authors better describe the “synergistic” relationship(s) based on the patterns driven by individual and combined hECs?

5. Line 330: “...development of the paroccipital process, a conical prominence of bone adjacent to the outer ear and serving as an attachment point for certain neck muscles, was inhibited in the dumbo, Wnt1::Cre;Hmx1fl/fl and mEC1del/del mice but not in mEC1dup/dup (Fig. 5l)” In Fig. 5l the difference of paroccipital process phenotypes in Wnt1::Cre;Hmx1fl/fl versus mEC1dup/dup appears not that obvious at first sight. Ideally measures for quantification and replicates are added.

6. While for mEC1dup/dup convincing expression data is shown in fig. 5i and 5j, it remains unclear to what extent Hmx1 expression is downregulated in mEC1del/del embryos/mice. Such information is particularly relevant for functional interpretation of enhancer relationships implied by the title and subject in the Discussion (e.g., enhancer redundancies). Ideally the authors show qualitative or quantitative Hmx1 expression along with the mouse ear phenotypes for each genotype in Fig. 5k.

7. The single cell RNA-seq results from sorted Hmx1-P2A-GFP shown in Fig. 6d-g is focusing mostly on specifying the role of Hmx1 at the (sub)cell type level. In line 365, the authors state that “genes that were substantially downregulated (logFC > 1) in mEC1dup/dup mice were predominantly not expressed in Hmx1+ cells of WT mice” which is an interesting finding. To define effectors downstream of Hmx1-overexpression (due to duplicated mEC1) beyond the Comp and Arhgap36 genes ideally the authors would perform comparative scRNA-seq from mEC1dup/dup (Fig. 5l).

8. In the discussion (line 408) the authors state: “In this study, we elucidate the mechanism of Hmx1 gene patterning related to both the enhancer and motif cluster sequences. We have delineated an intra-arch positional identity enhancer cluster, named PI-HEC.” While the authors convincingly reveal the role of mEC1/hEC1 in context with pinna development and BCE, they do not address the functional contributions of the other enhancers (hEC2, hEC3) identified as part of the PI-HEC. While the authors demonstrate the activities of all three enhancer regions at the “transcriptional reporter level”, the lack of in vivo analysis for “synergistic” and “coordinated” effects implicating hEC2 and hEC3 have to be taken into account in their statement in line 429 “However, this study present, for the first time, the synergistic and coordinated effects of multipartite regulatory elements on the spatial regulation of gene expression in a disease-related locus.”

9. Directly related to the above comment, the title of the study seems to be somewhat misleading, as the second part of the study is about the duplication/deletion of only mEC1/hEC1. The authors do not perform in vivo deletions of EC2 and/or EC3 which would be essential to delineate synergistic or redundant relationships at the functional level, as implied in the title (“Dysregulation in Spatiotemporal Expression of HMX1 Coordinated by Multifaceted Enhancers...”).

Minor points:

1. In the main text (line 211) it should be mentioned that a site-directed transgenic approach (enSERT) has been used. The authors should also clearly describe in which embryonic subregions hEC1 and hEC2 overlap and elucidate potential redundancies.

2. Given an overlapping fraction of TFs in GRNs orchestrating limb and craniofacial development, and in light of the reproducible limb activity of hEC3, it would be interesting to analyze epigenomic marks of developing limbs. Are limb-enhancer signatures present at hEC1, 2 and 3?

3. Line 201: “We employed the whole-mount in-situ hybridization (WISH, E9.5) and transgenic Hmx1-P2A-EGFP mouse

reporter line (E11.5 and E14.5) to elucidate the spatiotemporal dynamics of Hmx1 expression during embryonic ear development”: Here the authors should highlight in the main text that the Hmx1-P2A-EGFP line is a new mouse line they generated and also provide information about the exon targeted (in Fig. 4j), and whether Hmx1 levels and/or functions are affected by addition of the P2A-EGFP moiety.

4. Line 249: “hEC1 is responsible for most transcriptional output, while hEC2 and hEC3 provide the supplementary positional information for the transcriptional output”. What about repressive sites, given that it seems that hEC1 activity is suppressed in hEC1+hEC2 (+hEC3) embryos?

5. The results shown in panels 4h and 4i are based on or reflecting publicly available data of, in my opinion, only supporting value. These panels should therefore be moved to the supplementary data section. In line 305 the source of these data should also be referenced.

6. Line 311: “To investigate human BCE development mechanisms, we created a transgenic mouse model, mEC1dup/dup, containing four copies of mEC1 sequences”: To separate the mouse line generated clearly from LacZ reporter transgenics (also mentioned in the subsequent sentence), the authors should refer here to an “endogenous/knock-in mouse model” instead.

7. Ext Data Fig. 4j should be placed before panels d-l (enhancer-reporter transgenesis) to align with the narrative in the main text/figures.

8. In the Methods section (line 617): “hEC2.1-chr4: 8705075-8706475; hEC2-chr4: 8705075-8706475”: coordinates of elements are identical (?)

Reviewer #2

(Remarks to the Author)

The authors present a comprehensive study of a specific group of *cis*-regulatory elements that modulate bilateral constricted ear (BCE) malformation, an outer-ear phenotype within the overall area of auricular dysregulation. The authors present a compelling collection of results that fine map the specific enhancer cluster involved in outer ear malformation in affected pedigrees of East Asian ancestry, demonstrating with *in vivo* and *in vitro* assays the spatiotemporal activity of this cluster in a relevant cell type, namely CNCCs, identify the specific transcription factors bound to the most important enhancer of the cluster, and model the human defect in mice upon targeted enhancer element duplication. In this process, the authors establish *HMX1*, a transcriptional repressor important for craniofacial development including the eye and ear, as the target gene of this enhancer cluster and subsequently perform a multifaceted interrogation of *HMX1* activity and regulation during early animal development which guides them to pinpoint the specific pharyngeal arch and cell-type, namely fibroblast, that could be important for the phenotype in an affected human population.

This work augments the authors' previous work [(Quina, L, et al. *Disease Models & Mechanisms* 2012) and (Rosin, J, et al. *Development* 2016)], and the work of others [for example (Schorderet et al. *AJHG* 2008)] in solidifying the enhancer cluster and *HMX1* as important genetic and epigenetic contributors of auricular dysregulation. The results from this work are noteworthy for several reasons. First, it could lead to establishing a low-cost prophylactic PCR strategy for genetic screening of the disorder in affected human populations. Second, it positions the authors to take the next steps in characterizing the molecular mechanisms by which *HMX1* hyperactivation leads to craniofacial defects that extend to the inner ear and eyes.

Major comments

1. It is questionable if the BCE locus defined by the authors is defined correctly or as inclusively enough to include other enhancers within the cluster, besides EC1/2/3, that could contribute to BCE malformation. This question emerges from Figure 5K, where the outer-ear phenotypes observed from *Wnt1::Cre;HMX1^{fl/fl}* conditional knockout animals look much more severe, specifically in the helix area, compared to *mEC1^{dup/dup}* mice. This suggests that other enhancers, maybe *mEC2* and *mEC3* or potentially other distal-acting enhancers outside the author-defined BCE locus, may contribute to the phenotype. There is motivation for this in the authors' other figures. First, as suggested in Promoter-Capture HiC enhancer-promoter interaction data of Figure 2B and Extended Figure 2C, there exist other distal-acting enhancers that influence *HMX1*'s expression. Second, in Figure 1C, the authors show that some pedigrees contain copy number duplication outside of the authors' defined BCE locus. The current BCE locus definition by the authors appears to be the common intersection of all pedigree-specific BCE regions.

2. Are “PA-like CNCCs” truly the most relevant and representative cell type for performing *in vitro* assays? The authors demonstrate by single-cell RNA-seq of the second pharyngeal arch, that *HMX1⁺* cells are enriched within fibroblast cells and to a lesser extent, in chondrocytes. Differentiating hCNCCs to these two lineages is feasible (for example, myofibroblast in (Lee, G. et al. *Nat Biotechnology* 25, 1468-1475 (2007)) and chondrocytes in (Long, H.K. et al. *Cell Stem Cell* 27, 765-783 e714 (2020)). It seems plausible that these terminal cell-type derivatives may be a better *in vitro* model for luciferase, LC/MS, ChIP-, and RNA-seq assays conducted in this study rather than the loosely defined “PA-like CNCCs”.

3. The “PA-like CNCC” of this study should be better characterized. I am skeptical if the authors have gotten the right “PA-like CNCCs” that are most representative of PA2, the specific prominence in which the BCE-associated enhancer cluster is

active and where *HMX1* is expressed. This questioning stems from the incongruence observed for hEC1's D1 and D3 deletion between *in vitro* luciferase assay done in "PA-like CNCs" and *in vivo* LacZ transgenic mouse (see Figure 4A). Retinoic acid treatment has been a common morphogen used for posteriorizing NCCs along the body axis, such as for enteric or trunk locales (see Frith, T.J.R. et al. *Stem Cell Reports* 15, 557-565 (2020), and Frith, T.J. et al. *Elife* 7 (2018); reviewed in Williams, A.L. & Bohnsack, B.L. *Genesis* 57, e23308 (2019)). This 'posteriorization' process is dependent on the dose and timing of the morphogen delivery. The authors should rationalize clearly their usage of 100nM RetA treatment and should explain if higher or lower concentrations were tested. At this concentration and administration window, which pharyngeal arch are the cells pushed toward (PA2, PA3, PA4, or more posterior)? The authors show a panel of genes in Ext Figure 3D and Figure 4H of PA2 enriched genes, however, the authors should show a detailed characterization of Hox gene expression, either through RT-qPCR or RNA-seq. They should include *Hoxa3*, *Hoxb3*, *Hoxd3*, and *Hoxd4*.

4. The mEC1 duplication animal contains a 2-copy number duplication of mEC1, which may not be a faithful model of the human pedigrees' copy number variation. The authors do not comment on the copy number variability of hEC1 segregating within affected individuals of their cohort. This information should be noted in Figure 1 or within a supplementary table. If the copy number of affected individuals is greater than two, then the authors are encouraged to generate mice with duplications at the upper bound of copy number duplication seen in the human cohort or explore the full spectrum of cohort-specific copy number duplications (i.e 2, 3, ..., etc).

5. The paper focuses heavily on EC1, to a lesser extent on EC2, and seems to introduce and then abandon EC3 in their study. However, the authors try to convince the reader that duplications of the BCE locus (i.e. all 3 enhancers in the PI-HEC) are what is responsible for spatiotemporal dysregulation of *HMX1* and ultimately pinna malformation. Specifically, in the latter half of their paper (in figures 4-6) they diverged entirely to a focus of EC1. Given that EC2 also contributes to *HMX1* expression in the proximal (alpha) position of the PA2/pinna region (Figure 3C), the authors should show comparable motif analysis and mouse phenotype due to mEC2 duplication. Finally, the authors say on line 188 that hEC3 is a structural element. What evidence is there to support this? Are there CTCF motifs or CTCF-ChIP enrichment at hEC3?

Minor Comments:

1. *HMX1*, also known as *NKX5-3*, is closely related in homology to the broader family of homeodomain *NKX5* family members. It could be helpful to briefly introduce the similarities and dissimilarities of *HMX1* from other *NKX5* paralogs (i.e. *HMX2*, *HMX3*, etc.) early in the paper regarding its function, developmental role, and spatiotemporal expression.
2. Are there SNPs associated with normal-range ear morphology variation in humans that intersect with the BCE locus? For example, see [(Adhikari, K. et al. *Nature Communications* 2015) or (Li, Y. et al. *PLoS Genetics* 2023)] for ear morphometry genome-wide association studies.
3. In Figures 3D and 3E, the luciferase reporter assay of the 3 enhancers, and control sequences should be conducted at D11 and P4 hCNCs. Such figure(s) can be added as a supplement. This information can provide insight into the time at which these enhancers start becoming active in an *in vitro* context.
4. Line 301- Figure 4G indicates that hEC1-D3 could be a repressor. That is, deletion of D3 compared to wild-type hEC1-D3+D4 results in an increase in reporter activity. Why is this?
5. On all figures or figure legends, please specify the p-value, or the p-value scale attributable to each asterisk, and the statistical test used. This applies specifically to Figure 3C, 3F, 4A, 4G, 5F, 5H, and corresponding extended figures.
6. Line 336- *HMX1* ectopic activation in mEC1 duplication leads to suppression of numerous genes. Do you find enrichment of the *HMX1* motif within the *cis*-regulatory elements of these suppressed targets?
7. Figure 3C and 3F- "Relative Expression" is relative to which gene or condition? Please describe clearly in the figures' y-axes or legends.
8. Line 1192- Correct Figure 5g to Figure 4g

Reviewer #3

(Remarks to the Author)

This manuscript by Xu and colleagues presents significant progress to the understanding of how non-coding variants can alter phenotypes with relevance to human disease. Overall, this is a large and impressive body of work, which should be well viewed by the Developmental Biology and Gene Regulatory Fields. For further impact in Genetics and Medical fields additional supporting cases in other genetic backgrounds may be warranted.

Major comments:

1. The authors should consider better controls for many of their experiments. One particular stand out is the proteomics approach which is poorly described and has no control included at all. The protein IDs from this experiment mainly include

novel unknown proteins, but without a negative control what does the data really mean.

2. It is unclear what the authors mean by single cell lineage tracing. Analysis of scRNAseq data from multiple ages may infer expression trajectory but has no spatial context. This needs better description.

Minor comments:

1. Better description of the pedigrees.
2. Labels 5l.
3. Spatial in situ of HMX1 is mentioned but only E9.5 shown.

Reviewer #4

(Remarks to the Author)

In this work, Xu and colleagues dissect the Hmx1 regulatory regions responsible for gene expression in the pharyngeal arch and pinna. They characterize in detail three regulatory modules that synergistically create an Hmx1 expression pattern. They link duplications of these modules, found in seven families, to bilateral constricted ears. To achieve this, they produce a mouse model that accurately reproduces the human phenotypes and perform further analyses to understand the transcriptional differences between normal and altered pinna development.

This work elegantly provides a well-described molecular mechanism linking non-coding variants to congenital ear malformation. The interplay between in vitro and in vivo approaches robustly supports all points made by the authors. Nevertheless, I have a few concerns for the authors to address. Pending these changes, I strongly endorse the publication of this work in Nature Communications.

1. Generally, chromatin structure data should be presented differently:

- In Figure 1C, the Hi-C track spans a much broader region than the one under investigation, and the lines indicating the region of interest are difficult to discern initially. I would suggest the authors directly zoom in on the region of interest. Additionally, the displayed TAD structure (black and red arrows) is not well-supported by the data presented. Indeed, a Hi-C dataset of higher quality, even from a different tissue, could be more informative than the one presented here. The display of H3K26me3, a marker of transcribed gene bodies, also seems inappropriate.
- To further describe the locus's 3D organization, a CTCF ChIP-seq track could be very informative in Figures 1C and/or 2B.
- In Figure 2B, the Promoter Capture Hi-C interaction profiles should be displayed atop the called loops.

2. In Figures 3H-K, the dotted black line drawn on the facial structures of E11.5 LacZ transgenic embryos is neither defined in the text nor in the figure legend. Additionally, the correspondence between the upper right schematic of pinna structure and the embryo staining is not straightforward. The authors should consider providing a clearer description or a zoomed-in view of the region of interest, similar to the Hmx1-GFP fluorescence images.

3. The description of the hEC3 enhancer as "primed" is unclear, particularly as there is no visible staining shown at later developmental stages. Can the authors clarify what they mean? Additionally, is this region bound by CTCF?

4. In the discussion, the absence of eye-related phenotypes in patients, in contrast to what is observed in mice, should be explored. Could this be attributed to the differences between homozygosity and heterozygosity in mouse models and human patients?

Minor points:

L61: "Developmental biology, a discipline that has recently seen significant progress." While we agree with this statement, reference #1 (Atchley et al., 1991) may not be the most current example to demonstrate recent advancements in the field.

L118: Suggest using "GeneChip" instead of "genechip".

L154: "Promoter Capture Hi-C (PC-HiC)." The acronym PChi-C is commonly used in the field, as in L158.

L211: "In the LacZ assays assessing enhancer activity..." This sentence could be reformulated to clarify the identity of the enhancers tested. For example: "In the LacZ assays assessing BCE enhancers' activity."

In Figure 2A, the color code of CS13 to CS17 tracks should be described rather than merely referenced on a website.

Figure 3A-F: The colors used to represent the three different EC enhancers, the three stages of hCNCCs sampled, and the two genotypes tested are all relatively similar and complicate the graphical understanding of Figure 3F.

Figure 3G: An "r" is missing in "CNCCs early development."

Figure 6E-F: Could the authors specify in the figure how the seven clusters from panels E-F were selected from panel D? Our understanding is that they are found E10.5, but this should be acknowledged in the main figure.

Version 1:

Reviewer comments:

Reviewer #1

(Remarks to the Author)

The authors have convincingly addressed all major points raised by this reviewer and included substantial additional experimental data and analysis, leading to an overall improvement of the manuscript which I consider ready for publication in Nature Communications.

I would however recommend to still consider the following points:

- The new title should be more accurate and clear. The authors found that HMX1 is controlled by hierarchical and synergistic interactions among the clustered enhancers in the wildtype situation. The title does however not reflect that a mutation leads to the ear malformation.
- Figure 3 (panels h-m) would still gain from clearer indication that the embryos in vertical order are the result of the same construct (at different timepoints) with indications of the timepoints on the left of the h panel. It is not immediately clear that the timepoints shown in g also account for panels h-m.
- Line 283: The sentence should include that "...hEC1 and hEC2 likely function as synergistic enhancers..." as the result is based on transgenic reporter analysis out of endogenous context. Therefore, the actual endogenous enhancer properties could still be subject to additional modification.
- Line 369: The reference (Fig. 5l) should be placed after the previous sentence describing the result. The respective figure legend of panel 5l (line 1328) should mention what type of expression analysis this is.

Reviewer #2

(Remarks to the Author)

First, I must thank the authors for addressing each point of my critique thoroughly with additional figures that have been made using pre-existing data, and by conducting further experiments. These revisions add to the monumental amount of evidence already presented prior and strengthen the authors' manuscript.

Second, the manuscript's narrative is clearer, especially in the presentation of the ssRNA-seq data. Specifically, adding ssRNA-seq data from mEC1^{dup/dup} strengthens the authors' spatiotemporal determination of mEC1's effects in fibroblast cells. I also appreciate the authors' work on performing RetA titration and gene expression analysis of marker genes to demonstrate the posteriorizing of CNCC to a PA2/PA3 locale. Their presentation of this data is well rationalized, and I commend the authors for the inclusion of these experiments in the revised manuscript.

Finally, the authors have satisfactorily addressed each of my critique points and I would recommend the publication of this research article in Nature Communications. In my response to their rebuttal, I have raised a few additional questions. The authors are not compelled to address these in the manuscript, but I would be delighted to see them do so.

Major Comment #1 response to rebuttal:

I am pleased that the authors acknowledge the other enhancers outside the BCE core locus. I agree with the authors that these enhancers are indeed weak based on the chromatin tracks presented. These may become active in some unknown developmental context that is beyond the scope of the study. Though not necessary for further revision, it would be worthwhile in later studies to test these elements in luciferase assay conducted in late CNCCs and PA2-line CNCCs. Noteworthy, in future luciferase and lacZ experiments, it may be worthwhile replacing the LUC or LacZ reporter's promoter, which may be minSV40 or HSP90 promoters, with the *HMX1* basal promoter to determine if there is improved reporter response due to enhancer-promoter compatibility.

Major Comment #2 response to rebuttal:

In their rebuttal, the authors demonstrate that *HMX1* is not expressed in *in vitro* derived cranial chondrocytes, nor do hEC1/2/3 elements show accessibility in *in vitro* chondrocyte ATAC-seq data. This is concordant with the authors finding in their ssRNA-seq data that "the *HMX1* gene is primarily expressed in fibroblasts, and the duplication of mEC1 leads to a significant expansion of *HMX1* expression in these cells (Fig. 6c)" (line 384-386). Yet, the authors say in lines 409-411 that "differential gene analysis revealed significant downregulation of genes such as *Msx1*, *Fgf18*, and *S100a4* in the chondrocyte subpopulation" (see Figure 6e). However, Figure 6c shows very little *HMX1* expression in chondrocytes (cluster 9). If there are *HMX1*-expressed cells in cluster 9, then they must be rare because I can only see a few red points. Therefore, I am confused by the discrepancy between the paucity of HXM1 expression in chondrocytes and the authors' finding of DEG

in chondrocytes. Is this due to organismal and/or *in vitro* vs *in vivo* differences? That is, the CNCC *in vitro* data is human and the ssRNA-seq *in vivo* data is mouse? If there is a tiny subpopulation of chondrocytes in Figure 6c that is driving the DEGs shown in Figure 6e, I find it exciting to know what these cells are.

Major comment #3 response to rebuttal:

First, I agree with the authors that the *in vitro* CNCC model is not a drop-in replacement for the *in vivo* mouse system. However, the readership can benefit from an explanation of why 100uM RetA was chosen in this protocol. The RT-qPCR results showing regionalized gene expression of marker genes are crucial, and I thank the authors for this addition.

Second, the authors say “In fact, based on our WISH results and transgenic GFP mouse model, *HMX1* expression domain extends partially from PA2 into the region that develops from PA3”. I examined the updated Figure 3g, Figures 5i,j, and Extended Figure 5f but it is not obvious where *HMX1*'s expression expands into the PA3. The only figure that shows this expanded expression to PA3 is in the scRNA-seq analysis presented in Extended Figure 2A, however, showing this in an animal image is preferred. Please note this expanded PA3 expression by highlighting the expression domain in the WISH or HXM1-GFP reporter animals presented in the main or supplementary figures. It could be worthwhile to show a zoomed-in version of the mouse if the PA3 region is unclear from the whole animal image.

Major comment #4 response to rebuttal:

I appreciate the authors' specification of the n=3 BCE copy number in their affected cohort and for noting in their rebuttal that they are indeed testing 0 to 4 copies of the mEC1 enhancer in their transgenic animals. This information could also be helpful for the reader, and should be written in the results section titled “Impact of spatial *HMX1* expression on pinna development: insights from transgenic mouse models mimicking human BCE anomalies.” I was puzzled why 3 copies did not show ear defects, and the authors should note in the discussion section about their hypothesis. That is, “this may be due to the insensitivity of mice to HMX1 dosage and the activation of genetic compensation mechanisms, a phenomenon commonly observed in mouse models of human diseases^{17, 18}”.

Major Comment #5 response to rebuttal

I agree with the authors that EC2 and EC3 need to be explored more in *in vivo* experiments, as noted in the discussion, but the authors have adequately shown that it is the EC1 element that contributes the most to BCE malformation.

The authors write “while hEC3 shows weak activity in both LUC and LacZ experiments independently, it significantly enhances the gene-driving capability when combined with hEC1+hEC2, similar to enhancers described in the Molecular Cell paper.” As noted in comment #1, in future luciferase lacZ experiments, it may be worthwhile replacing the LUC or LacZ reporter's promoter with the *HMX1* promoter to see if EC3 does show activity due to promoter compatibility. In light of the absence of CTCF motifs and chip-seq peaks, it is exciting to understand how EC3 functions as a structural element or if it could indeed be a facilitator.

Additional Minor comments:

Figure 3 h-i: “The orange number in the bottom right corner of each embryo indicates the count of positive LacZ staining embryos”. Does this number include only LacZ-positive embryos, likely measured from PCR, or does it also include animals that show the same expression domain as the representative animal shown? Neither the methods section nor the legend makes this clear. For reproducibility, an expression domain should be consistent between at least 2 LacZ positive embryos.

Text is difficult to read in Figure 6D. I would recommend labeling only a few genes.

Superscript text is difficult to read in Figure 5M. I would recommend making the font larger.

Reviewer #3

(Remarks to the Author)

The authors have made significant improvements to the original manuscript addressing all of my original concerns. This publication is expected to be well received by a general audience.

Reviewer #4

(Remarks to the Author)

The authors have satisfactorily addressed all my points.

I would, however, ask them to provide more detail in the legend of Figure 2B regarding the promoter capture Hi-C tracks. Specifically, they should clarify the meaning of the full and dotted lines, the points, and the color of the different points. This is particularly important as the current display of the data is not intuitive.

Dear Editors and Reviewers,

We would like to express our sincere gratitude for your thorough review and the detailed comments provided by the four reviewers regarding our manuscript titled "Dysregulation in Spatiotemporal Expression of *HMX1* Coordinated by Multifaceted Enhancers Drives an Auricular Disorder" (NCOMMS-24-24495-T).

The insightful feedback from the reviewers has been instrumental in enhancing both the quality and clarity of our manuscript. We have carefully considered each comment and made substantial revisions in response. All modifications are highlighted in the revised manuscript for your convenience.

In the following sections, we provide detailed responses to each reviewer's comments and summarize the major changes we have implemented. We believe these revisions have significantly strengthened our manuscript and hope they address all concerns raised during the review process.

REVIEWER COMMENTS

Reviewer #1 (Remarks to the Author):

In the study of Zhang et al. the authors tackle the genomic mechanisms and molecular/cellular defects underlying bilateral ear constriction (BCE) which represents an auricular disorder. The authors start by performing linkage analysis and target capture sequencing for fine mapping of BCE in seven Chinese families, which identifies duplications of a non-coding, CNCC enhancer-containing interval near the *HMX1* transcription factor, a gene implicated in outer ear development and related malformations. The authors then use state-of-the art transgenesis and genome editing to generate advanced in vitro (human) and in vivo (mouse) models to validate and functionally dissect the enhancer modules within the identified genomic interval. The authors thereby characterize the spatiotemporal activity patterns of three enhancer modules and combinations thereof (hEC1, hEC2, hEC3) in context with transcriptional control of *Hmx1* expression in CNCCs essential for auricular/pinna development. While these experiments provide novel insights into synergistic and hierarchical enhancer interactions based on transgenic reporter output, it remains to be noted that the mouse orthologous region of the human EC1 enhancer has been previously identified in Rosin et al., 2016 (PMID: 27287804). EC1 appears to be clearly the strongest enhancer, while hEC2 drives significantly more restricted expression in PA CCNCs, with additional repressive/modulating roles. The role of EC3 in this process remains less clear. The authors then focus their study on EC1 and perform an in-depth functional characterization by first addressing the functional characteristics of EC1-subregions using a tiling deletion approach (based on site-directed reporter transgenesis in

mouse embryos). This allows to pinpoint functionally relevant enhancer subregions and TF motifs. The authors then generate an EC1 duplication model in mice to convincingly show that EC1 enhancer duplication is the likely mechanistic determinant underlying the BCE phenotype observed in the initially studied human pedigrees. In turn, the authors also demonstrate that EC1 deletion, as well as Wnt1::Cre-mediated conditional Hmx1 deletion, in mice leads to the BCE-opposing dumbbopinna phenotype (along with a defective paroccipital process). In a final part, the authors perform whole-mount transcriptome analysis to identify DEGs resulting from ectopically expressed Hmx1 (in muscle and epidermis tissue of the pinna) as a result of EC1 duplication. Single-cell analysis in Hmx1-P2A-GFP reporter embryos (in wildtype context) further elucidates on the role of Hmx1 in this process.

Overall, this is a comprehensive and impressive study going the whole distance from mapping a human (auricular) disorder to resolving the (regulatory) mechanism at the genomic level and thereby defining BCE as an enhanceropathy. In addition to valuable in vitro experiments on the basis of hESC-derived CNCCs, the authors made consistent use of a novel site-directed transgenic approach for more accurate enhancer-reporter transgenesis and generated four different mouse alleles using CRISPR-Cas9 to consolidate their results and to ensure high-quality in vivo data. I consider this study suitable for Nature Communications and of high interest for the readership, given the following points are taken into account:

Major points:

1. As the authors cite in the introduction, Rosin et al., 2016 (PMID: 27287804) reports the identification of a distal 594 bp ECR that specifies Hmx1 expression in pinna and lateral facial morphogenesis and is regulated by the Hox-Pbx-Meis complex. Here, the authors reference that hEC1 maps to the “mouse 594 bp ECR” (here termed mEC1) only later in the Results section. However, this circumstance should already be described in the beginning of the Results (when the hEC1 enhancer is defined) and the position of the “mouse 594 bp ECR” (mEC1) should also be indicated in the locus schemes of Figures. 2a and 3a.

Response: We sincerely appreciate your valuable suggestions. We have added the description of the “mouse 594bp ECR,” which is contained in the homologous mouse sequence of hEC1 on lines 167-168 of the revised manuscript. At the same time, as you suggested, we also modified the locus schemes of **Fig. 2a** and **Fig. 3a** and added the description in the corresponding figure legends.

2. Despite the significantly improved efficiency and accuracy of enhancer-driven reporter signals in site-directed transgenic reporter assays such as enSERT, at least n=2 replicates need to be shown to corroborate a result, as a random integration event with position effects cannot be excluded in multi-copy (“tandem”) integrations. Therefore, the n=1 for hEC3 that the authors show in panel I of Figure 3 is not sufficient to demonstrate PA activity and at least n=2 embryos with overlapping signal in

the respective tissue are required for a robust result.

In relation to this, is there a reason why the authors analyzed hEC3 only at E11.5? For consistency with hEC1 and hEC2 and to evaluate potential earlier/later craniofacial activities, the authors should also provide information about hEC3 activity at E9.5 and E14.5.

Response: We sincerely appreciate your suggestions. We previously generated transgenic embryos for hEC3 at E11.5 twice; however, none of them reproduced the PA activity observed in the single positive embryo shown. We speculate that hEC3 may have very weak or unstable activity at this time point, consistent with the observation that hEC3 lacks the epigenetic markers P300 and H3K27ac (**Fig. 3a**), which are canonical signals associated with transcriptional activity. Alternatively, hEC3 alone may be insufficient to drive transcriptional activity, and the activity we originally observed may have resulted from a specific integration site. As correctly pointed out, two replicates are required to validate the result. Therefore, we have included the majority of LacZ results in **Fig. 3k**, and the sole PA pattern has been moved to the supplementary results (**Extended Data Fig. 6d**). Additionally, we revised our description of the hEC3 E11.5 pattern in PA2 in the revised manuscript (lines 243-248).

Additionally, we generated E9.5 and E14.5 transgenic hEC3 embryos for LacZ staining. The results showed that hEC3 activity was not detected at E9.5 but was observed in the forelimbs and hindlimbs at E14.5, consistent with its expression pattern in the limbs at E11.5 (**Figure R1a**). We also observed relatively stable but weaker signals in the ear at E14.5, specifically in the lower part of the pinna (**lp**; **Figure R1b**). This explains the signals in the lp region when hEC3 was introduced alongside hEC1 and hEC2 (**Fig. 3m**). These findings further demonstrate the positional specificity of the three enhancers' activity in the ear, consistent with our overall conclusions. We have incorporated this new result into **Fig. 3k** and added a description of this finding in the revised manuscript on lines 243-248.

Figure R1: enSERT assay result of hEC3 at E9.5 and E14.5. a, LacZ staining in hRE3-E9.5; b, LacZ staining in hRE3-E14.5, red arrow shows lower part of pinna (lp).

3. In Fig. 3 (panels h-m) it is not entirely clear how the authors extrapolate from the LacZ pattern observed in the whole mount embryos at E11.5 (images) to the regional distribution of reporter activity shown in alpha, beta and gamma subregions in the schemes. Ideally, the authors add close-ups as shown in panel G (E11.5) to better visualize staining in these areas.

Response: We are very grateful for your expert recommendations. To clearly describe the regional distribution of LacZ activity in the PA2 zone, we have reorganized the layout of **Fig. 3 h–m** and incorporated magnified views (**Fig. 3, h'–m'**) to highlight the signal patterns. Additionally, we have included schematic regional subdivisions (alpha, beta, and gamma) comparable to those shown in panel G.

4. Line 221/222: “These results were somewhat similar to those using cellular models..”. Could the authors describe better which aspects of the interacting enhancers are synergistic and which are following a hierarchical logic (as described e.g. in Long et al., 2016, PMID: 27863239)? Related to this, in line 246 the authors state: “These results indicate that spatial specificity of Hmx1 expression around PA2 is finely tuned by the coordinated and synergistic enhancers within an enhancer cluster”. Could the authors better describe the “synergistic” relationship(s) based on the patterns driven by individual and combined hECs?

Response: Thank you for your insightful comments and for pointing out the need for a more detailed description of the synergistic and hierarchical relationships among the interacting enhancers. We have carefully revised the manuscript to address these points, using the valuable reference paper you provided (Long *et al.*, 2016). We have expanded the discussion to better describe the aspects of synergy and hierarchical logic among the interacting enhancers, as shown in **Figure R2**.

Figure R2. Crosstalk between enhancers within the same cis-regulatory domain. (original figure from long *et al.* 2016¹)

In terms of regional activity, hEC1 is primarily active in the alpha region of the PA2 zone, while hEC2 shows activity mainly in the beta region. When combined, hEC1 and hEC2 exhibit activity in both the alpha and beta regions, with significantly stronger staining intensity compared to hEC2 alone, indicating that these two enhancers are synergistic both in spatial activity and expression levels. In contrast, hEC3 alone does not show any activity in the PA2 zone. However, when hEC3 is present together with hEC1 and hEC2, the combined enhancers (hEC1 + hEC2 + hEC3) exhibit activity in the alpha, beta, and gamma regions, with an expanded staining pattern compared to hEC1 + hEC2 alone. This demonstrates that hEC3 contributes synergistic and hierarchical characteristics to the interaction between hEC1 and hEC2.

We have modified our description in the revised manuscript (lines 280-285). Thank you again for your suggestions, which have helped us better describe the complex interactions among these enhancers.

5. Line 330: "...development of the paroccipital process, a conical prominence of bone adjacent to the outer ear and serving as an attachment point for certain neck muscles, was inhibited in the dumbo, *Wnt1::Cre;Hmx1^{fl/fl}* and *mEC1^{del/del}* mice but not in *mEC1^{dup/dup}* (Fig. 5I)" In Fig. 5I the difference of paroccipital process phenotypes in *Wnt1::Cre;Hmx1^{fl/fl}* versus *mEC1^{dup/dup}* appears not that obvious at first sight. Ideally measures for quantification and replicates are added. Response: Thank you for highlighting this. To further demonstrate that the development of the paroccipital process is inhibited in the *Wnt1::Cre;Hmx1^{fl/fl}* mice, we crossed *Wnt1::Cre;Hmx1^{fl/+}* with *Hmx1^{fl/fl}*. However, consistent with our previous matings, we encountered significant difficulties in obtaining viable homozygous mice that survived through birth to adulthood. Over the past several months, we successfully obtained only one additional specimen suitable for micro-CT analysis. We have nevertheless revisited each scan and done three separate measurements on each rendered cranoskeleton, including for background controls, the *Wnt1::Cre;Hmx1^{fl/fl}* samples, some dumbo animals as well as EC1-KI, KO and KD adults. We specifically measured both the exterior and interior aspect of the paroccipital process, measuring from the distal top of the process to separate landmarks on each side, as well as the height of the periotic capsule adjacent to the paroccipital process (**Figure R3a**).

Although the specimens were of various adult ages (4 weeks, 8 weeks, 9 months and 1 year), it is important to note that in control mice both the paroccipital process and periotic capsule did not change in size (see below) between 4 weeks and 1 year consistent with each having reached their final adult size by 4 weeks of age(**Figure R3b**). The measurements also clearly highlight the significant paroccipital process hypoplasia in both dumbo and *Wnt1::Cre;Hmx1^{fl/fl}* mice, and to a lesser degree in the EC1-KO mouse (**Figure R3c,d**). The EC1-KD and EC1-KI mice have normal size paroccipital processes. The periotic capsule height was not significantly different in any strain when compared to controls (**Figure R3c,d**). Finally, the left and right measurements were graphed individually to better assess variation. This showed that the *Wnt1::Cre;Hmx1^{fl/fl}* mice showed

considerable variation between the left and right paroccipital processes (**Figure R3e**). In further support of the conclusion that the *Wnt1::Cre;Hmx1^{f/f}* phenotype recapitulates the dumbo phenotype, we also include a rendered microCT image of the pinna showing the duplicated proximoventral cartilage as previously reported for the dumbo mice (Rosin *et al.*, 2016)(**Figure R3f**)². This is not seen in EC1-KI mice (**Figure R3f**) or the EC1-KD mice (not shown).

These data have been combined into **Extended Data Fig.8 e-j**. A statement on the quantification results has been added to the manuscript (lines 372-375 and 863-869).

Figure R3: Morphometric analysis of paroccipital processes and periotic capsules in control and mutant mice across age groups. **a**, The exterior and interior length of each paroccipital process and the height of the periotic capsule on each side were measured on 3D rendered scans using the 3D coordinate landmarks denoted by the yellow dots. **b**, The exterior and interior lengths of the paroccipital processes and the height of the periotic capsule in control mice measured at 4 weeks, 8 weeks, 9 months and 1 year of age. The measurements suggest that the paroccipital processes have reached their near-final adult length by 4 weeks of age. **c**, Paroccipital process and periotic capsule measurements were graphed based on age, showing the distinct genotypes. **d**, The average of the left and right side paroccipital process measurements and the periotic capsule measurements were graphed to display the respective genotype-specific sizes. The dots represent separate measurements. The distribution of each genotype is also shown to the right of the individual measurements for each. **e**: Left and right side paroccipital process and periotic capsule measurements were graphed separately to assess lateral variability. Left side (purple dots/distribution), right side (teal dots/distribution) based on genotypes. The dots represent separate measurements. The distribution is shown to the right of the individual measurements for each side. **f**, 3D renderings of the soft tissue

on heads of *Wnt1::Cre;Hmx1^{fl/fl}* and EC1-KI mice. White arrow indicates the pinna cartilage duplication, which is also seen in *dumbo* mice.

6. While for mEC1dup/dup convincing expression data is shown in fig. 5i and 5j, it remains unclear to what extent *Hmx1* expression is downregulated in mEC1del/del embryos/mice. Such information is particularly relevant for functional interpretation of enhancer relationships implied by the title and subject in the Discussion (e.g., enhancer redundancies). Ideally the authors show qualitative or quantitative *Hmx1* expression along with the mouse ear phenotypes for each genotype in Fig. 5k.

Response: Thank you for your suggestions. To be transparent, we currently do not have mice available to assess *Hmx1* expression levels in *mEC^{del/del}* embryos, as this mouse model was unfortunately lost in the year of 2022 due to disruptions caused by the COVID-19 pandemic. We did not preserve embryos through cryopreservation, but we retained photographs and some adult samples (fixed in 4% PFA) for micro-CT analysis.

Fortunately, we conducted RNA-seq experiments at that time using ear tissue from postnatal day 0 (P0) mice, including WT, *mEC^{del/del}*, and *mEC^{dup/dup}* genotypes. We believe this RNA-seq dataset addresses your question regarding “to what extent *Hmx1* expression is downregulated in *mEC^{del/del}* embryos/mice.” The results showed that *Hmx1* expression in *mEC^{del/del}* decreased to approximately one-ninth compared to WT (**Figure R4**). This significant reduction nearly eliminated the majority of *Hmx1* expression, indicating that mEC1 plays a dominant role in regulating *Hmx1* expression in the ear. We have included this result in **Fig. 5I** of the revised manuscript (lines 365-369).

Figure R4: RNA-seq on mEC mice models. Left panel shows PCA analysis of RNA-seq samples from WT, *mEC^{del/del}*, and *mEC^{dup/dup}* genotypes. Right panel shows FPKM values of WT, *mEC^{del/del}*, and *mEC^{dup/dup}* genotypes.

Currently, we do not have enough samples to assess *Hmx1* expression in the *Wnt1::Cre;Hmx1^{fl/fl}* pinna at P0. Regarding Dumbo mice, the phenotype is primarily caused by a mutation in the first exon (Gln65*), which results in premature termination of protein translation and subsequently affects *Hmx1* function. We sincerely appreciate your questions, which have helped us improve the quality of the manuscript.

7. The single cell RNA-seq results from sorted *Hmx1*-P2A-GFP shown in Fig. 6d-g is focusing mostly on specifying the role of *Hmx1* at the (sub)cell type level. In line 365, the authors state that

“genes that were substantially downregulated ($\log_{2}FC > 1$) in *mEC1dup/dup* mice were predominantly not expressed in *Hmx1*⁺ cells of WT mice” which is an interesting finding. To define effectors downstream of *Hmx1*-overexpression (due to duplicated *mEC1*) beyond the *Comp* and *Arhgap36* genes ideally the authors would perform comparative scRNA-seq from *mEC1dup/dup* (Fig. 5l).

Response: Thank you for this thoughtful suggestion. To provide stronger evidence that *mEC1* drives the expansion of *Hmx1* expression while identifying the affected cell types and downstream genes, we performed single-cell sequencing experiments using the pinna prominence of WT and *mEC1^{dup/dup}* mice at E14.5 (**Fig. 6a-6g and Extended Data Fig. 9**). We have included this new result in the revised manuscript (lines 380-386, lines 397-417).

We believe our new scRNA-seq results can provide more detailed insights into the gene regulatory changes across different cell types caused by the expansion of the *Hmx1* gene.

8. In the discussion (line 408) the authors state: “In this study, we elucidate the mechanism of *Hmx1* gene patterning related to both the enhancer and motif cluster sequences. We have delineated an intra-arch positional identity enhancer cluster, named PI-HEC.” While the authors convincingly reveal the role of *mEC1/hEC1* in context with pinna development and BCE, they do not address the functional contributions of the other enhancers (*hEC2*, *hEC3*) identified as part of the PI-HEC. While the authors demonstrate the activities of all three enhancer regions at the “transcriptional reporter level”, the lack of *in vivo* analysis for “synergistic” and “coordinated” effects implicating *hEC2* and *hEC3* have to be taken into account in their statement in line 429 “However, this study present, for the first time, the synergistic and coordinated effects of multipartite regulatory elements on the spatial regulation of gene expression in a disease-related locus.”

Response: We sincerely appreciate your careful attention to the overstatements in our manuscript. We fully acknowledge that comprehensive validation of enhancer cluster functionality would require knock-in and knock-out experiments for *hEC2* and *hEC3*. While our primary objective was to elucidate the molecular mechanisms underlying human BCE disease, for which our current mouse models provide substantial support, we recognize the need for more extensive functional validation. We plan to generate additional transgenic mouse models in future studies to fully decipher how this sophisticated enhancer cluster orchestrates precise gene regulation *in vivo*.

Here, we initially evaluated the activity of *mEC2* using enSERT technology to determine whether it exhibits transcriptional activity consistent with *hEC2*. The staining results revealed that *mEC2* primarily drives gene expression in the dorsal root ganglia and lacks the ability to drive gene expression in PA2 (**Figure R5a**), which is a key feature of *hEC2*. This discrepancy may be attributed to the low conservation of sequences between humans and mice (**Figure R5b**), indirectly reflecting the increased complexity of human auricular regulation. Furthermore, based on the findings from Rosin *et al.* (2016), the staining pattern of *mEC1* in mice appears to correspond to the combined effects of *hEC1* and *hEC2* in humans (**Figure R5c**), indicating that *mEC1* in mice has a broader

regulatory role². Therefore, future *in vivo* functional studies may require the development of more targeted mouse models, such as large-fragment insertions, to better simulate the copy number variation phenomena observed in humans.

Figure R5: Comparative LacZ staining and conservation analysis of human and mouse enhancers. a, LacZ staining comparison between hEC2 and mEC2; **b,** Conservation analysis of EC1 and EC2; **c,** LacZ staining comparison between hEC1+hEC2 and mEC1 (Rosin *et al.*, 2016)².

Accordingly, we have revised our claims (lines 486-491, lines 516-520) and modified the potentially misleading title to “**Hierarchical and Synergistic Interactions of Enhancers within Clusters Lead to Ear Malformations: A Dominant Enhancer Mimics Human Phenotypes**” to ensure scientific accuracy. We are grateful for your rigorous review, which has greatly contributed to maintaining high standards of scientific research.

9. Directly related to the above comment, the title of the study seems to be somewhat misleading, as the second part of the study is about the duplication/deletion of only mEC1/hEC1. The authors do not perform *in vivo* deletions of EC2 and/or EC3 which would be essential to delineate synergistic or redundant relationships at the functional level, as implied in the title (“Dysregulation in Spatiotemporal Expression of HMX1 Coordinated by Multifaceted Enhancers...”).

Response: Please see the response to your comment 8.

Minor points:

1. In the main text (line 211) it should be mentioned that a site-directed transgenic approach (enSERT) has been used. The authors should also clearly describe in which embryonic subregions hEC1 and hEC2 overlap and elucidate potential redundancies.

Response: Thank you for your suggestions. We have added the description, "a site-directed transgenic approach (enSERT) was adopted," when referring to the LacZ assay in the revised manuscript (lines 243-248). Additionally, we have also described the overlap region and potential redundancies in the revised manuscript: "We found that hEC1 and hEC2 partially overlap in the upper right corner of the beta region, which suggests that the two may have some redundant characteristics. However, they are not interchangeable, as the spatial activity domain of hEC1+hEC2 is larger than that of hEC1 or hEC2 alone" (lines 269-273).

2. Given an overlapping fraction of TFs in GRNs orchestrating limb and craniofacial development, and in light of the reproducible limb activity of hEC3, it would be interesting to analyze epigenomic marks of developing limbs. Are limb-enhancer signatures present at hEC1, 2 and 3?

Response: Thank you for raising this interesting question. Indeed, there are many similarities between the gene regulatory networks of branchial arch and limb development, as reported in numerous studies^{3, 4, 5, 6}.

To address your question about "Are limb-enhancer signatures present at hEC1, 2 and 3", we conducted a literature search and found a relevant paper published in *CELL* (Cotney *et al.*, 2013)⁶. This study performed H3K27ac ChIP-seq analysis on human embryonic limb development at stages E33, E41, E44, and E47 (**Figure R6, Upper panel**). We downloaded their enriched H3K27ac bed files (GSE42413), which revealed H3K27ac marks in all three enhancer regions (**Figure R6, Bottom panel**)⁶. These findings are consistent with our observation of enhancer activity that was detected in the limb. This consistency further indicates that the gene regulatory network governing craniofacial development, particularly the development of neural crest cells (NCCs), shares a certain degree of similarity with the gene regulatory network controlling limb development.

[Figure redacted]

Figure R6: H3K27ac ChIP-seq Stages of Human Limb Development and Enhancer Activity Analysis Using CNCC ATAC-seq Data. Upper panel comes from Cotney *et al.* (2013) that shows four stages of human limb development for H3K27ac ChIP-seq⁶. Bottom panel shows the result of six IGV genome browser tracks (CNCC ATAC-seq rep1, rep2 bw file tracks and E33, E41, E44, E47 bed file tracks). Three red rectangles represent locations of three enhancers (hEC1, hEC2, and hEC3).

3. Line 201: “We employed the whole-mount *in-situ* hybridization (WISH, E9.5) and transgenic Hmx1-P2A-EGFP mouse reporter line (E11.5 and E14.5) to elucidate the spatiotemporal dynamics of Hmx1 expression during embryonic ear development”: Here the authors should highlight in the main text that the Hmx1-P2A-EGFP line is a new mouse line they generated and also provide information about the exon targeted (in Fig. 4j), and whether Hmx1 levels and/or functions are affected by addition of the P2A-EGFP moiety.

Response: Thanks for your professional suggestions. In the revised manuscript, we have added the description of the Hmx1-P2A-EGFP mouse line in lines 221-225. Additionally, we provided the detailed transgenic information (gene knock-in schematic diagram) of this new mouse line in **Fig. 3g**. In order to elucidate whether spatiotemporal *Hmx1* expression is affected by addition of the P2A-EGFP moiety, we performed whole-mount *in-situ* hybridization (WISH) at E10.5, E11.5 and E14.5, and compared them with corresponding GFP signal (**Figure R7a, b, c**). From the images, we can see that the GFP signal pattern and intensity largely coincide with the endogenous gene's spatiotemporal expression pattern at each stage. Due to the large size of the embryos at E14.5, whole-embryo *in-situ* hybridization is challenging, which affects probe penetration. As a result, the signal intensity may vary; however, most of the signals are consistent. Therefore, we conclude that we have successfully established a mouse model that faithfully recapitulates the *in vivo* gene expression pattern, and we have added these results in the **Extended Data Fig.4e, f, g**.

Figure R7: Spatiotemporal pattern of *Hmx1* expression level. **a, b, c,** Left panel shows image of Hmx1-P2A-EGFP, right panel shows image of *WISH* in each stage E10.5 (**a**), E11.5 (**b**) and E14.5 (**c**). CM, craniofacial mesenchymal; OP, otic placode; DRG, dorsal root ganglia; bp, basal pinna; lp, lower part of pinna; dp, distal pinna; pmp, the proximal region of the mandibular prominence.

4. Line 249: “hEC1 is responsible for most transcriptional output, while hEC2 and hEC3 provide the supplementary positional information for the transcriptional output”. What about repressive sites, given that it seems that hEC1 activity is suppressed in hEC1+hEC2 (+hEC3) embryos?

Response: Thank you for highlighting this important observation. Indeed, we also noticed this pattern when analyzing our results, especially in the lateral nasal process (LNP), frontonasal prominence (FNP), forelimbs and hindlimbs. When hEC1 was present alone, strong staining signals were observed in the FNP and LNP regions. However, in the presence of hEC1+hEC2, the signals in these regions were significantly reduced. Furthermore, when all three elements (hEC1+hEC2+hEC3) were present, the signal intensity in the LNP region was almost completely abolished, suggesting that hEC2 and hEC3 exert repressive effects on hEC1 activity in these cranial regions.

On the other hand, hEC1 activity was detected mainly in the anterior part of forelimbs and hindlimbs, while hEC2 activity was predominantly observed in the posterior hindlimbs. In the hEC1+hEC2 combination, the signal was localized to the posterior hindlimbs, demonstrating the repressive effects of hEC2 on hEC1 activity in the limb regions. hEC3 activity was detected in the posterior forelimbs and distal forelimbs. When all three elements (hEC1+hEC2+hEC3) were present, the signal was detected throughout most of the limb regions; however, there was a notable reduction in signal intensity in the anterior parts of both forelimbs and hindlimbs (regions where hEC1 alone showed strong activity), indicating that these elements exert repressive effects on hEC1 activity. Therefore, we modified our conclusion in the revised manuscript lines 280-285.

5. The results shown in panels 4h and 4i are based on or reflecting publicly available data of, in my

opinion, only supporting value. These panels should therefore be moved to the supplementary data section. In line 305 the source of these data should also be referenced.

Response: We appreciate your professional suggestions. We have moved the content of **Fig. 4h** and **4i** to **Extended Data Fig. 7d, e**. Consequently, we relocated the LC-MS content from Extended Data Fig. 6c, d to **Fig. 4c, d**. The overall layout of **Fig. 4** and **Extended Data Fig. 6** has been reorganized to better align with the narrative flow of the manuscript.

We have included the relevant references for the source of these data in the revised manuscript (line 340). We appreciate your help in refining and improving our manuscript.

6. Line 311: “To investigate human BCE development mechanisms, we created a transgenic mouse model, mEC1dup/dup, containing four copies of mEC1 sequences”: To separate the mouse line generated clearly from LacZ reporter transgenics (also mentioned in the subsequent sentence), the authors should refer here to an “endogenous/knock-in mouse model” instead.

Response: We appreciate your attention to this important clarification. In the revised manuscript, we have incorporated the term "endogenous/knock-in mouse model" (lines 346-348) to clearly distinguish it from the LacZ reporter transgenics mentioned in the subsequent sentence.

We appreciate your feedback in helping us improve the precision and clarity of our manuscript.

7. Ext Data Fig. 4j should be placed before panels d-I (enhancer-reporter transgenesis) to align with the narrative in the main text/figures.

Response: We have implemented the following changes:

1) Moved **Extended Data Fig. 4j** to become **Extended Data Fig. 5d** to better align with the narrative in the main text and figures.

2) Incorporated the requested *in-situ* hybridization data into **Extended Data Fig. 5e, f, g**.

We appreciate your help in enhancing the readability and organization of our manuscript.

8. In the Methods section (line 617): “hEC2.1-chr4: 8705075-8706475; hEC2-chr4: 8705075-8706475”: coordinates of elements are identical (?)

Response: We appreciate your attention to this error. We have corrected the coordinates from "hEC2-chr4: 8705075-8706475" to "hEC2-chr4: 8705075-8707510" and updated the corresponding parts in the revised manuscript (line 708).

We appreciate your careful review in improving the accuracy of our manuscript.

Reviewer #2 (Remarks to the Author):

The authors present a comprehensive study of a specific group of *cis*-regulatory elements that modulate bilateral constricted ear (BCE) malformation, an outer-ear phenotype within the overall area of auricular dysregulation. The authors present a compelling collection of results that fine map the specific enhancer cluster involved in outer ear malformation in affected pedigrees of East Asian ancestry, demonstrating with *in vivo* and *in vitro* assays the spatiotemporal activity of this cluster in a relevant cell type, namely CNCCs, identify the specific transcription factors bound to the most important enhancer of the cluster, and model the human defect in mice upon targeted enhancer element duplication. In this process, the authors establish *HMX1*, a transcriptional repressor important for craniofacial development including the eye and ear, as the target gene of this enhancer cluster and subsequently perform a multifaceted interrogation of *HMX1* activity and regulation during early animal development which guides them to pinpoint the specific pharyngeal arch and cell-type, namely fibroblast, that could be important for the phenotype in an affected human population.

This work augments the authors' previous work [(Quina, L, et al. *Disease Models & Mechanisms* 2012) and (Rosin, J, et al. *Development* 2016)], and the work of others [for example (Schorderet et al. *AJHG* 2008)] in solidifying the enhancer cluster and *HMX1* as important genetic and epigenetic contributors of auricular dysregulation. The results from this work are noteworthy for several reasons. First, it could lead to establishing a low-cost prophylactic PCR strategy for genetic screening of the disorder in affected human populations. Second, it positions the authors to take the next steps in characterizing the molecular mechanisms by which *HMX1* hyperactivation leads to craniofacial defects that extend to the inner ear and eyes.

Major comments

1. It is questionable if the BCE locus defined by the authors is defined correctly or as inclusively enough to include other enhancers within the cluster, besides EC1/2/3, that could contribute to BCE malformation. This question emerges from Figure 5K, where the outer-ear phenotypes observed from *Wnt1::Cre;HMX1^{fl/fl}* conditional knockout animals look much more severe, specifically in the helix area, compared to *mEC1^{dup/dup}* mice. This suggests that other enhancers, maybe *mEC2* and *mEC3* or potentially other distal-acting enhancers outside the author-defined BCE locus, may contribute to the phenotype. There is motivation for this in the authors' other figures. First, as suggested in Promoter-Capture HiC enhancer-promoter interaction data of Figure 2B and Extended Figure 2C, there exist other distal-acting enhancers that influence *HMX1*'s expression. Second, in Figure 1C, the authors show that some pedigrees contain copy number duplication outside of the authors' defined BCE locus. The current BCE locus definition by the authors appears to be the common intersection of all pedigree-specific BCE regions.

Response: We sincerely appreciate your insightful comment regarding the ambiguity in our BCE

definition. Our defined BCE locus represents the core regulatory elements shared across the seven pedigrees, although we recognize that some pedigrees (particularly pedigrees 3 and 5) contain duplications outside this region. We have analyzed these additional potential enhancers, as shown in **Figure R8**.

Figure R8: Analysis of potential candidate enhancers (pECs) outside of our defined BCE locus.

Our analysis identified several potential regulatory elements (pECs) defined by *in vivo* epigenomic data from human craniofacial tissue (CS13-CS17):

- pEC1: present in pedigrees 5, 4, 1, 7, and 2
- pEC2: present in pedigrees 5, 1, 2, and 3
- pEC3: present in pedigrees 5, 2, and 3
- pEC4 cluster: exclusive to pedigree 5

These regulatory elements were characterized as weak enhancers with inconsistent detection across developmental stages (CS13-CS17) and showed no active epigenetic signals in our *in-vitro* derived hCNCCs model. Given their variable presence across pedigrees and weak regulatory potential, we focused our analysis on the shared characteristics of the BCE locus.

The BCE locus, consistently present in all seven families, proved to be functionally significant. Our mouse models demonstrated that the three enhancers within this locus accurately reproduced *Hmx1* gene expression in the PA2 region. Moreover, the mouse model with increased EC1 copy number successfully replicated the external ear malformation phenotype observed in patients.

We acknowledge that other candidate enhancers may influence *Hmx1* expression and external ear phenotypes. The phenotypic variations among our seven families could result from:

- 1) Differences in other genetic factors affecting ear development
- 2) Variations in the length of BCE locus copy number alterations

While generating mouse models for all patient variations would be technically challenging, we plan

to conduct more comprehensive studies of this region in the future (as demonstrated in PMID: 28846100)⁷. To improve clarity, we have renamed the BCE locus as the "**BCE core locus**" and included these considerations in the revised manuscript's discussion section (lines 466-473).

2. Are “PA-like CNCCs” truly the most relevant and representative cell type for performing *in vitro* assays? The authors demonstrate by single-cell RNA-seq of the second pharyngeal arch, that HMX1⁺ cells are enriched within fibroblast cells and to a lesser extent, in chondrocytes. Differentiating hCNCCs to these two lineages is feasible (for example, myofibroblast in (Lee, G. et al. *Nat Biotechnology* 25, 1468-1475 (2007)) and chondrocytes in (Long, H.K. et al. *Cell Stem Cell* 27, 765-783 e714 (2020)). It seems plausible that these terminal cell-type derivatives may be a better *in vitro* model for luciferase, LC/MS, ChIP-, and RNA-seq assays conducted in this study rather than the loosely defined “PA-like CNCCs”.

Response: We sincerely appreciate your professional recommendations. In our initial investigation, identifying the most representative cellular model posed a significant challenge. Our rationale for selecting neural crest cell (NCC)-derived mesenchymal cells as our disease model was supported by multiple lines of evidence: Firstly, as illustrated in the single-cell analysis of adult human ears presented in Quiat *et al.* (2022)⁸, the external ear primarily comprises chondrocytes (CHON), skeletal muscle1/2 (SKM1/2), fibroblasts (FIB), endothelial cells (EC) (**Figure R9a**) and smooth muscle cells (SMC). Notably, this study also employed NCCs as a cellular model to investigate microtia pathogenesis. Two candidate pathogenic genes, *ROBO1* and *ROBO2*^{9, 10}, are both highly expressed in fibroblasts and chondrocytes (**Figure R9b**). Secondly, our single-cell sequencing data also revealed that fibroblasts, chondrocytes, and neurons in the external ear are all derived from CNCC-derived mesenchymal cells. Therefore, it is reasonable to utilize these terminally differentiated cells' common progenitor for investigating this pathology.

Figure R9: scRNA-seq of adult human pinna (images come from Quiat *et al.*, 2022⁸).

Furthermore, previous studies and recent lineage tracing techniques have consistently demonstrated the intimate association between mouse external ear development and CNCCs. A recent publication (Allen *et al.* 2023) has specifically shown that mouse external ear fibroblasts and cartilage originate from CNCCs using *Wnt1-cre;ROSA^{mT/mG}* mice¹¹. Additionally, our epigenomic data demonstrated

that enhancers within the BCE locus become activated in CNCCs, coinciding with a significant upregulation of *HMX1* expression. Given this convergence of literature evidence and experimental findings, we opted to employ CNCC-derived mesenchymal cells (post migratory CNCCs) as our cellular model. Our current results validate this choice as appropriate, at least within this developmental window. While we cannot exclude the possibility of additional regulatory elements affecting *HMX1* expression in terminally differentiated cells (such as fibroblasts and chondrocytes), we maintain that utilizing NCCs for disease modeling during this developmental period represents a well-justified approach. Additionally, we have generated comprehensive lacZ results from E9.5 to E14.5. These high-quality results are more convincing than attempting to use the most ideal cell line. Given the microenvironmental differences between *in vivo* and *in vitro* systems, there is currently no perfect cellular model that can fully replicate *in vivo* results. Therefore, we tend to use the hCNCC model here as a reference for our specific developmental window. Although it is a suboptimal cellular model, the experimental results have established a strong connection between the enhancers within the BCE core locus and *HMX1* expression levels.

Nevertheless, we did explore the possibility of differentiating CNCCs into terminal cell types (such as chondrocytes and fibroblasts). Following the differentiation protocol described in the referenced paper (Long *et al.* 2020)¹², we firstly extract their RNA-seq results and reanalyze the *HMX1* expression level. However, we observed an unexpected significant decrease in *HMX1* expression (**Figure R10a**), which contradicted our anticipated results. Subsequently, we analyzed the epigenomic data (ATAC-seq) from their chondrocyte differentiation experiments and discovered that the active epigenetic marks at the BCE locus were indeed lost during this process (**Figure R10b**). Therefore, differentiating CNCCs into chondrocytes is not a suitable approach using this protocol.

Figure R10: Analysis of *HMX1* expression and epigenetic signals in the BCE locus between late hCNCC and chondrocytes. a, *HMX1* expression between late hCNCC and Chon (chondrocyte). b, ATAC-seq of late hCNCC, D5 Chon, D9 Chon (data from Long *et al.*, 2020¹²).

Regarding the suggestion of myofibroblast differentiation, while we had reviewed the relevant literature, our detailed examination revealed that their differentiation protocol differed substantially from our current EB-based differentiation method. We could not ensure that this approach would accurately recapitulate the disease phenotype. Therefore, as previously stated, we maintained our

strategy of using CNCCs - the common progenitor of these terminally differentiated cells - as our disease model. This approach has proven both reasonable and effective, at least within the developmental window of CNCC postmigration. Looking forward, we are considering the implementation of cutting-edge organoid models to simulate ear morphogenesis, similar to the well-established inner ear organoid models^{13, 14}, but it needs a comprehensive exploration.

We have incorporated this discussion regarding our choice of cellular model into the revised manuscript's discussion section (lines 476-481).

3. The “PA-like CNCC” of this study should be better characterized. I am skeptical if the authors have gotten the right “PA-like CNCCs” that are most representative of PA2, the specific prominence in which the BCE-associated enhancer cluster is active and where *HMX1* is expressed. This questioning stems from the incongruence observed for hEC1’s D1 and D3 deletion between *in vitro* luciferase assay done in “PA-like CNCCs” and *in vivo* LacZ transgenic mouse (see Figure 4A). Retinoic acid treatment has been a common morphogen used for posteriorizing NCCs along the body axis, such as for enteric or trunk locales (see Frith, T.J.R. et al. *Stem Cell Reports* 15, 557-565 (2020), and Frith, T.J. et al. *Elife* 7 (2018); reviewed in Williams, A.L. & Bohnsack, B.L. *Genesis* 57, e23308 (2019)). This ‘posteriorization’ process is dependent on the dose and timing of the morphogen delivery. The authors should rationalize clearly their usage of 100nM RetA treatment and should explain if higher or lower concentrations were tested. At this concentration and administration window, which pharyngeal arch are the cells pushed toward (PA2, PA3, PA4, or more posterior)? The authors show a panel of genes in Ext Figure 3D and Figure 4H of PA2 enriched genes, however, the authors should show a detailed characterization of Hox gene expression, either through RT-qPCR or RNA-seq. They should include *Hoxa3*, *Hoxb3*, *Hoxd3*, and *Hoxd4*.

Response: We sincerely appreciate your professional recommendations. The retinoic acid (RA) treatment approach was primarily based on the literature you referenced, especially considering the limited available methods for differentiating NCCs into region-specific craniofacial populations. As shown in the figure (**Figure R11a**), the RA concentration at the *Hmx1*-expressing posterior PA2 region (In fact, based on our WISH results and transgenic GFP mouse model, *HMX1* expression domain extends partially from PA2 into the region that develops from PA3) (**Extended Data Fig. 5f**) maintains an intermediate level (higher than PA1 but lower than PA4). In conjunction with another study we examined, which employed 100 nM RA treatment and conducted RT-PCR analysis, we observed a characteristic expression pattern: downregulation of *OTX2* (PA1 marker) accompanied by upregulation of *DLX1*, *Hoxa2* (PA2 markers), and *Hoxa3* (PA3 marker) (**Figure R11b,c**). These findings demonstrate that RA treatment can effectively modulate the regionalization of NCCs into specific pharyngeal arch identities.

Figure R11: Supporting evidence that RA signaling modulates the regionalization of NCCs into specific pharyngeal arch identities. **a**, RA concentration gradient in the PAs axis (image comes from Williams, *et al.*, 2019)¹⁵. **b**, **c**, 100 nM RA treatment can modulate the regionalization of NCCs into specific pharyngeal arch identities (images come from Fukuta *et al.*, 2014)¹⁶.

We greatly appreciate your suggestion to test different concentrations. We initially tested a concentration of 200 nM RA and found that high concentrations of RA were detrimental to the growth of late hCNCCs (**Figure R12a**), leading us to abandon this concentration. Subsequently, we conducted a more comprehensive analysis using four concentration gradients (0 nM, 1 nM, 10 nM, 100 nM). We performed mRNA extraction followed by quantitative PCR analysis of multiple genes including *HOXA2*, *MEIS1*, *PBX1*, *DLX6*, *HOXA3*, *HOXB3*, *HOXD3*, and *HOXD4* (**Figure R12b**). The results showed that treatment with 100 nM RA led to the most significant upregulation of genes associated with PA2/PA3, while 10 nM RA caused a slight increase in gene expression. In contrast, PA4-associated genes (e.g., *HOXD4*) showed no upregulation. This indicates that the application of 100 nM RA can temporarily confer PA2/PA3-like characteristics to late hCNCCs, aligning well with the objectives of our study. We have added these results in the **Extended Data Fig. 4a**.

Figure R12: *In vitro* differentiation system from late hCNCCs to PA-like hCNCCs state. a, 200 nM RA concentration is detrimental to the growth of late hCNCCs. **b,** RT-PCR analysis of PA-related genes in different RA-treated late hCNCCs (0 nM, 1 nM, 10 nM, 100 nM).

As for your mention of “the incongruence observed for hEC1’s D1 and D3 deletion between the *in vitro* luciferase assay performed in ‘PA-like CNCCs’ and the *in vivo* LacZ transgenic mouse (see Figure 4A),” we first acknowledge that the D1 deletion indeed shows inconsistency with the *in vivo* experiments. However, the D3 deletion partially aligns with our *in vivo* LacZ experiments, as the staining in the PA2 region decreases following the D3 deletion (**Fig. 4a and Extended Data Fig. 7a**). The enhanced staining primarily occurs in the PA1 region, which itself does not align with the PA2/PA3-like characteristics of late hCNCCs we are studying. As for why the D1 deletion results differ between the two systems, we hypothesize that this discrepancy is likely due to differences in the microenvironment between *in vivo* and *in vitro* systems. As we noted earlier, there is currently no perfect *in vitro* system that fully replicates the *in vivo* environment. To ensure the most robust results, we therefore chose to focus our subsequent analyses on the D4 region, where the results from both systems were most consistent.

We sincerely appreciate your highly professional and valuable suggestions, which have prompted us to reflect on how to construct models that better represent *in vivo* systems in future studies. This is especially critical for congenital genetic diseases, where conducting experiments *in vivo* is often challenging. Thanks very much again for your insightful feedback.

4. The mEC1 duplication animal contains a 2-copy number duplication of mEC1, which may not be a faithful model of the human pedigrees’ copy number variation. The authors do not comment on the copy number variability of hEC1 segregating within affected individuals of their cohort. This information should be noted in Figure 1 or within a supplementary table. If the copy number of affected individuals is greater than two, then the authors are encouraged to generate mice with

duplications at the upper bound of copy number duplication seen in the human cohort or explore the full spectrum of cohort-specific copy number duplications (i.e 2, 3, ..., etc).

Response: Thank you for raising this important point. We apologize for not providing the copy number information for our pedigrees. In all our pedigrees, the copy number is 3 (line 140), indicating a heterozygous state with one copy on one chromosome and two on the homologous chromosome. Our *mEC1^{dup/dup}* model has 4 copies (homozygous state). We have developed and characterized mouse models representing the full spectrum of mEC1 copy numbers:

- 0 copies (*mEC1^{del/del}*)
- 1 copy (*mEC1^{del/+}*)
- 2 copies (*mEC1^{+/+}*, wild type)
- 3 copies (*mEC1^{dup/+}*)
- 4 copies (*mEC1^{dup/dup}*)

Notably, no ear defects were observed in mice with 1 or 3 copies. This may be due to the insensitivity of mice to *HMX1* dosage and the activation of genetic compensation mechanisms, a phenomenon commonly observed in mouse models of human diseases^{17, 18}.

We acknowledge that generating mouse models with duplication patterns specific to each cohort would be ideal. We believe that our current evidence substantially elucidates the underlying mechanisms of BCE disease. In this manuscript, we have already established four novel mouse models that successfully mimic the features of outer ear malformation. Given the significant technical challenges associated with creating additional models involving large fragment insertions, we have decided to address these aspects in a separate ongoing project focused on comprehensively studying the regulatory mechanisms of this locus. We sincerely appreciate your valuable suggestion and thank you for highlighting this important aspect.

5. The paper focuses heavily on EC1, to a lesser extent on EC2, and seems to introduce and then abandon EC3 in their study. However, the authors try to convince the reader that duplications of the BCE locus (i.e. all 3 enhancers in the PI-HEC) are what is responsible for spatiotemporal dysregulation of *HMX1* and ultimately pinna malformation. Specifically, in the latter half of their paper (in figures 4-6) they diverged entirely to a focus of EC1. Given that EC2 also contributes to *HMX1* expression in the proximal (alpha) position of the PA2/pinna region (Figure 3C), the authors should show comparable motif analysis and mouse phenotype due to mEC2 duplication. Finally, the authors say on line 188 that hEC3 is a structural element. What evidence is there to support this? Are there CTCF motifs or CTCF-ChIP enrichment at hEC3?

Response: We sincerely appreciate your careful attention to the overstatements in our manuscript. We acknowledge that, as you pointed out, our manuscript primarily employed the LacZ reporter gene to analyze three enhancers in the BCE locus. Our main objective was to demonstrate the strong correlation between the spatiotemporal expression patterns of these enhancers and *HMX1*'s intrinsic expression pattern, with detailed analysis and mouse model construction focusing on the

predominant EC1 enhancer. The primary purpose of this approach was to verify that enhancer duplication in the BCE locus could indeed produce disease phenotypes similar to those observed in family patients, which our results confirmed. We are currently unable to predict whether duplications of EC2 or EC3 alone could alter mouse outer ear phenotypes. We suspect that duplication of mEC2 or mEC3 alone might not be particularly effective, and it may be necessary to duplicate the entire BCE locus to observe changes in mouse outer ear development. This is supported by the staining patterns of hEC1+hEC2+hEC3, which collectively simulate *Hmx1* expression and reflect the conditions in affected family members (Fig. 3m).

To assess whether mEC2 exhibits the same capacity as human hEC2 in driving gene expression in PA2, we tested a new cohort of embryos. The staining results revealed that mEC2 does not drive gene expression in PA2, but primarily drives expression in dorsal root ganglia (Figure R13). This evidence suggests potential differences in *HMXI* expression regulation between humans and mice. Rosin's previous mEC1 staining results (Figure R13) indicate that its gene-driving capability matches that of combined hEC1+hEC2. This finding highlights differences between human and mouse outer ears, as the more complex structure of human ears likely demands more refined regulatory mechanisms.

Figure R13: Comparative LacZ staining and conservation analysis of hEC2, mEC2, hEC1+hEC2, and mEC1 enhancers. a, LacZ staining comparison between hEC2 and mEC2; **b,** Conservation analysis of EC1 and EC2; **c,** LacZ staining comparison between hEC1+hEC2 and

mEC1 (Rosin *et al.*, 2016)².

We hypothesize that hEC3 serves as a structural element based on two key observations:

- 1) PCHi-C results show clear interaction between hEC3 and the *HMX1* promoter, potentially facilitating physical contact between BCE locus enhancers and *HMX1* to drive gene expression.
- 2) While hEC3 shows weak activity in both LUC and LacZ experiments independently, it significantly enhances the gene-driving capability when combined with hEC1+hEC2, similar to enhancers described in the Molecular Cell paper (Chen *et al.*, 2023)¹⁹. In response to your query about CTCF motifs or CTCF-ChIP enrichment at hEC3, hCNCC ChIP results indicate no such enrichment (**Fig. 2a**).

We have revised potentially overstated descriptions and conclusions in our manuscript (lines 486-491 and 506-509) and modified our article title to "**Hierarchical and Synergistic Interactions of Enhancers within Clusters Lead to Ear Malformations: A Dominant Enhancer Mimics Human Phenotypes**" to better reflect the core findings of our study. We greatly appreciate your professional and valuable questions. While this locus is indeed complex and may require systematic mouse models (similar to the approach in the Will *et al.*, 2017⁷) for full elucidation, we believe our current mouse model adequately addresses the disease mechanism of BCE. We plan to initiate a new project for more comprehensive and systematic analysis of this locus in the future.

Minor Comments:

1. *HMX1*, also known as *NKX5-3*, is closely related in homology to the broader family of homeodomain *NKX5* family members. It could be helpful to briefly introduce the similarities and dissimilarities of *HMX1* from other *NKX5* paralogs (i.e. *HMX2*, *HMX3*, etc.) early in the paper regarding its function, developmental role, and spatiotemporal expression.
Response: Thanks for your professional recommendations. We have added the introduction of similarities and dissimilarities of *HMX1* from other *NKX5* paralogs (i.e. *HMX2*, *HMX3*) in the revised manuscript (lines 77-84) regarding its function, developmental role, and spatiotemporal expression.

2. Are there SNPs associated with normal-range ear morphology variation in humans that intersect with the BCE locus? For example, see [(Adhikari, K. et al. Nature Communications 2015) or (Li, Y. et al. PLoS Genetics 2023)] for ear morphometry genome-wide association studies.

Response: Thank you for your questions. Upon review, we found that the population used in Adhikari, K. *et al.* (2015) was included in Li, Y. *et al.* (2023)^{20, 21}. Therefore, we downloaded the data from Li, Y. *et al.* and reproduced the Manhattan plot shown in their paper (**Figure R14a,b**). Subsequently, we analyzed the association signals at the BCE locus (**Figure R14c**). The minimum *P*-value for rs7683992 was 0.022, which does not reach genome-wide significance. As a result, we did not identify any SNPs within the BCE locus that are significantly associated with normal-range

ear morphology.

Figure R14: Manhattan plot of variants associated with normal-range ear morphology. a, Original Manhattan plot from Li, Y. *et al.* (2023)²⁰. **b,** Reproduced Manhattan plot using data from Li, Y. *et al.* to confirm the accuracy of the downloaded data²⁰. **c,** Regional plot of variants surrounding the BCE locus.

3. In Figures 3D and 3E, the luciferase reporter assay of the 3 enhancers, and control sequences should be conducted at D11 and P4 hCNCCs. Such figure(s) can be added as a supplement. This information can provide insight into the time at which these enhancers start becoming active in an *in vitro* context.

Response: Thanks for your professional recommendations. In the revised manuscript, we have performed additional luciferase reporter assay at D11 and P4 hCNCCs (**Figure R15**).

Figure R15: Luciferase assays evaluating candidate enhancers at the BCE locus in D11 hCNCC (a), and P4 hCNCC (b), including SV40 enhancer (positive control) and empty vector (negative control). Results from two independent experiments, each with three technical replicates (n = 6), are shown.

From the results, hEC1 exhibits only weak activity in D11 hCNCC, but becomes strongly active in subsequent late hCNCC/PA-like hCNCC. In contrast, hEC2 already shows activity in D11 hCNCC and continues to maintain strong activity in late hCNCC/PA-like hCNCC. This result is consistent with the expression level of the *HMX1* gene (Fig. 3C), which begins to rise slightly in D11 hCNCC and reaches high expression levels in late hCNCC/PA-like hCNCC. Furthermore, this result suggests that hEC1 is the dominant enhancer regulating *HMX1* expression, which aligns with the *in vivo* LacZ staining results. We have added these results in the **Extended Data Fig. 4b, c** and modified the description of this result in lines 201-209 in the revised manuscript.

Once again, thank you for helping us improve the completeness of our manuscript and for providing such insightful feedback.

4. Line 301- Figure 4G indicates that hEC1-D3 could be a repressor. That is, deletion of D3 compared to wild-type hEC1-D3+D4 results in an increase in reporter activity. Why is this?

Response: We greatly appreciate your careful review of our manuscript and for raising this interesting question. Upon revisiting the previously published literature, we indeed noticed similarities with their mouse 32 bp deletion results (as shown in the **Figure R16**). The first column is complete dmECR and the third column is 32 bp deletion of dmECR, we could see that after 32 bp deletion, the luciferase activity showed an increase.

Figure R16: Luciferase assay in the dmECR (mouse) (images comes from Rosin *et al.*, 2016)².

Based on these observations, we propose the following explanations: First, there might be protein competition among transcription factors binding to D3+D4 regions. The elimination of D3-1 might actually facilitate gene expression driven by D3+D4. Second, based on our LacZ staining results, D3-1's primary function appears to be restricting the spatial expression pattern of the gene for more precise localization, while D4-1's role is mainly to confer the enhancer's ability to drive gene expression (**Fig. 4b, bottom panel**). These two factors might explain the observed phenomenon. However, we acknowledge that even with organoid models, we cannot guarantee complete consistency between *in vivo* and *in vitro* results with current methodologies. We sincerely thank you for raising this question, as it prompts us to further consider the interactions between enhancers and proteins. This insight will guide our future research in determining which technical approaches could be employed to fully elucidate the enhancer model and deepen our understanding of gene regulation complexities.

5. On all figures or figure legends, please specify the p-value, or the p-value scale attributable to each asterisk, and the statistical test used. This applies specifically to Figure 3C, 3F, 4A, 4G, 5F, 5H, and corresponding extended figures.

Response: Thank you very much for pointing out this issue. We sincerely apologize for not including detailed statistical information.

In the revised manuscript, we have replaced the asterisks in **Fig. 3c, 3f, 4a, 4g, 5f, 5h**, and the corresponding extended figures (**Extended Data Fig. 3c, 4, 5c**) with the specific *P*-values. Additionally, we have specified the statistical methods in the figure legends. We greatly appreciate your feedback in helping us improve the clarity and transparency of our manuscript.

6. Line 336- *HMX1* ectopic activation in mEC1 duplication leads to suppression of numerous genes. Do you find enrichment of the *HMX1* motif within the *cis*-regulatory elements of these suppressed targets?

Response: We greatly appreciate your careful review of our manuscript and raising this interesting question. To address this, we extracted the promoter regions of all downregulated genes (284 in

total), including the 5' UTR and 2000 bp upstream flanking sequences. Using these 766 sequences, we performed motif enrichment analysis to assess Hmx1 motif enrichment within the promoter elements of these suppressed targets. The analysis revealed that the Hmx1 motif is not enriched in the promoter regions of these suppressed genes (**Supplementary Table 8, promoter region sheet**).

Furthermore, we used previously reported E14.5 pinna ATAC-seq peaks (± 1 Mb around suppressed gene bodies) and performed the same analysis (a total of 21,629 peaks). Results also showed that the Hmx1 motif is not enriched in the cis-regulatory elements (cREs) of these suppressed genes (**Supplementary Table 8, cRE region sheet**).

Therefore, we speculate that this repression is not direct, but rather a result of HMX1 expansion leading to gene downregulation due to adjacent tissue dysplasia. We have included this section in the revised manuscript (lines 389-392). Thank you very much for raising this interesting question.

7. Figure 3C and 3F- "Relative Expression" is relative to which gene or condition? Please describe clearly in the figures' y-axes or legends.

Response: Thank you very much for pointing out this issue. We sincerely apologize for the oversight. In Fig. 3C, "Relative Expression" represents the expression levels in H9, early hCNCC, late hCNCC, and PA-like hCNCC relative to H9 status. In Fig. 3F, "Relative Expression" represents the expression levels in KO cell lines relative to WT cell lines.

We have revised the manuscript to include clear descriptions on the y-axes and in the corresponding figure legends for both figures. We deeply appreciate your feedback, which has greatly contributed to improving the clarity and rigor of our manuscript.

8. Line 1192- Correct Figure 5g to Figure 4g

Response: Thank you for pointing this out. We have corrected **Fig. 5g** to **Fig. 4g** in the revised manuscript.

Reviewer #3 (Remarks to the Author):

This manuscript by Xu and colleagues presents significant progress to the understanding of how non-coding variants can alter phenotypes with relevance to human disease. Overall, this is a large and impressive body of work, which should be well viewed by the Developmental Biology and Gene Regulatory Fields. For further impact in Genetics and Medical fields additional supporting cases in other genetic backgrounds may be warranted.

Major comments:

1. The authors should consider better controls for many of their experiments. One particular stand out is the proteomics approach which is poorly described and has no control included at all. The protein IDs from this experiment mainly include novel unknown proteins, but without a negative control what does the data really mean.

Response: Thank you for your professional recommendations. We apologize for not providing detailed information about the proteomics approach. Here are the specific experimental procedures and steps:

1) Proteomics Approach Experimental Design

We designed three DNA-protein interaction lanes with the following components:

Components	Negative Control (NC)	Probe control	Sample
Unlabeled probe	-	+	-
Biotin-11-dUTP labeled probe	-	-	+
Nuclear protein from PA2 (E11.5)	+	+	+
Bioeast Mag-SA	+	+	+

2) Experimental Procedure

- Performed SDS-PAGE analysis
- Used silver staining to confirm DNA-protein binding
- Performed gel excision and protein recovery for the first lane (NC) and third lane (Sample).
- Performed LC-MS analysis

3) Data Processing Notes

- The proteins listed in the original supplementary table correspond to the differences between the Sample and the Negative Control
- We have replaced the novel unknown proteins with their corresponding uniprot IDs

4) Manuscript Revisions

- We added a detailed experimental description to the methods section (lines 730–732)
- Added two new sheets to the **Supplementary Table 4**:
 - Negative Control data

- Sample data

- Renamed the original sheet to "Sample-NC"

These modifications will provide greater clarity and completeness to our experimental methods and data processing procedures. We again thank you for professional suggestions, which have significantly improved the quality of our manuscript.

2. It is unclear what the authors mean by single cell lineage tracing. Analysis of scRNAseq data from multiple ages may infer expression trajectory but has no spatial context. This needs better description.

Response: Thank you for your insightful comment. We agree that the term 'single-cell lineage tracing' may have been misleading and requires further clarification. In our study, we utilized Hmx1-P2A-GFP reporter mice to isolate GFP-positive cells at three pivotal developmental stages (E10.5, E12.5, and E14.5) via fluorescence-activated cell sorting (FACS). These cells were then subjected to single-cell RNA sequencing (scRNA-seq) to capture their transcriptional profiles at each stage. While this approach does not involve traditional lineage tracing methods, which provide direct spatial and clonal information, it enables us to infer developmental trajectories of GFP-positive cell populations across these time points by analyzing stage-specific changes in gene expression profiles. This analysis allowed us to reconstruct the temporal dynamics of Hmx1-expressing cells and identify potential regulatory pathways involved in their development.

To address your concern, we modified our description in the revised manuscript to more accurately describe our methodology (lines 421-423). Specifically, we replaced "single-cell lineage tracing" with a more precise description, "transcriptional trajectory analysis of GFP-positive cells isolated from Hmx1-P2A-GFP embryos at three developmental stages (E10.5, E12.5, and E14.5)." This revision will clarify that our approach focuses on transcriptional dynamics rather than direct spatial or clonal lineage tracing.

Minor comments:

1. Better description of the pedigrees.

Response: Thanks for your suggestion. We have added more information about the pedigree used in this study. Please refer to lines 129-135 in the revised manuscript.

2. Labels 5I.

Response: Thanks for pointing it out. We have added labels to **Fig. 5I** in the revised manuscript for better clarity.

3. Spatial in situ of HMX1 is mentioned but only E9.5 shown.

Response: Thank you very much for raising this question, and we apologize for this oversight. The reason we did not include *WISH* images for other time points (i.e., E11.5 and E14.5) in **Fig. 3g**, and

instead used our newly constructed Hmx1-P2A-EGFP transgenic mice to demonstrate the spatiotemporal expression pattern of Hmx1, mainly because larger embryos are not washed thoroughly during WISH, which can result in some background signals. To effectively compare *Hmx1* expression with enhancer regulatory patterns, we decided not to include WISH images in **Fig. 3g**. However, to demonstrate the consistency between Hmx1-P2A-EGFP and endogenous Hmx1 expression, we included **Extended Data Fig. 5e–g** to demonstrate the consistency between EGFP signals and endogenous Hmx1 expression. (**Figure R17**).

Figure R17: Spatiotemporal pattern of *Hmx1* expression level. **a, b, c,** The left panel displays images of Hmx1-P2A-EGFP, while the right panel shows WISH images at each stage. **(a)**, E11.5 **(b)** and E14.5 **(c)**. CM, craniofacial mesenchymal; OP, otic placode; DRG, dorsal root ganglia; bp, basal pinna; lp, lower part of pinna; dp, distal pinna; pmp, the proximal region of the mandibular prominence.

Reviewer #4 (Remarks to the Author):

In this work, Xu and colleagues dissect the *Hmx1* regulatory regions responsible for gene expression in the pharyngeal arch and pinna. They characterize in detail three regulatory modules that synergistically create an *Hmx1* expression pattern. They link duplications of these modules, found in seven families, to bilateral constricted ears. To achieve this, they produce a mouse model that accurately reproduces the human phenotypes and perform further analyses to understand the transcriptional differences between normal and altered pinna development.

This work elegantly provides a well-described molecular mechanism linking non-coding variants to congenital ear malformation. The interplay between *in vitro* and *in vivo* approaches robustly supports all points made by the authors. Nevertheless, I have a few concerns for the authors to address. Pending these changes, I strongly endorse the publication of this work in *Nature Communications*.

1. Generally, chromatin structure data should be presented differently:
 - In Figure 1C, the Hi-C track spans a much broader region than the one under investigation, and the lines indicating the region of interest are difficult to discern initially. I would suggest the authors directly zoom in on the region of interest. Additionally, the displayed TAD structure (black and red arrows) is not well-supported by the data presented. Indeed, a Hi-C dataset of higher quality, even from a different tissue, could be more informative than the one presented here. The display of H3K26me3, a marker of transcribed gene bodies, also seems inappropriate.
 - To further describe the locus's 3D organization, a CTCF ChIP-seq track could be very informative in Figures 1C and/or 2B.
 - In Figure 2B, the Promoter Capture Hi-C interaction profiles should be displayed atop the called loops.

Response: Thank you very much for your professional advice. We acknowledge that the quality of this Hi-C data is suboptimal. To better illustrate the TAD structure at this locus, we selected a public Hi-C dataset from eye tissue, which is the site of *HMX1* gene expression, which ensures the results are more reliable. Following your suggestion, we focused specifically on the region containing *HMX1*, excluded the H3K36me3 track and incorporated the hCNCC CTCF-ChIP track to more clearly present the 3D organization of this locus (**Figure R18**).

Figure R18. Copy number duplication in each family, located in the intergenic region downstream of the *HMX1* gene, within the same TAD. The top panel displays Hi-C data (25 kb resolution) from human eye tissue, with the black dotted line marking the TAD boundary (5' TAD and 3' TAD boundary). In the middle panel, the CTCF signal (hCNCCs), a marker frequently observed at TAD boundaries, is depicted across this region. The lower panel specifies the duplication regions for each family, highlighting the minimum overlapping genomic area (chr4: 8691119-8728565, termed BCE core locus) and the reference genes across this region. The *HMX1* gene within the TAD is highlighted.

In **Fig. 2b**, we have added the Promoter Capture Hi-C interaction profiles atop the called loops (**Figure R19**).

Figure R19. Differential chromatin interactions link CE enhancers to *HMX1* promoter.

Promoter Capture Hi-C (PC-HiC) analysis demonstrates the differential physical interactions between the CE locus and the *HMX1* promoter in hESCs compared to hCNCCs. For each cell line, the upper panel illustrates the interaction profile, the middle panel depicts the identified loops with interaction strength indicated by line color, the bottom panel shows ATAC-seq track. The TAD boundary identified in **Fig. 1c** is depicted. The ATAC-seq and PC-HiC datasets for hESCs were obtained from GSE14532728 and GSE8682180, respectively, to ensure accurate and reliable data representation.

Thank you once again for your suggestions. They have made our data presentation clearer and more comprehensible, which better supports our conclusions.

2. In Figures 3H-K, the dotted black line drawn on the facial structures of E11.5 LacZ transgenic embryos is neither defined in the text nor in the figure legend. Additionally, the correspondence between the upper right schematic of pinna structure and the embryo staining is not straightforward. The authors should consider providing a clearer description or a zoomed-in view of the region of interest, similar to the *Hmx1*-GFP fluorescence images.
Response: We appreciate your insightful comments regarding the clarity of Figures 3H-K. To address these concerns, we updated the figure legend to clearly define the dotted black line on the facial structures of E11.5 LacZ transgenic embryos. The line demarcates the boundary of the second pharyngeal arch (PA2).

To enhance the alignment between the upper-right schematic of the pinna structure and the embryo staining, we have included a zoomed-in view of the region of interest (**Fig. 3h'-m'**), similar to the *Hmx1*-GFP fluorescence images, to provide a clearer depiction of the relevant structures. This enhancement facilitates clearer visualization and comparison of the staining patterns. We are grateful for your suggestions, which have significantly enhanced the quality of this figure.

3. The description of the hEC3 enhancer as “primed” is unclear, particularly as there is no visible staining shown at later developmental stages. Can the authors clarify what they mean? Additionally, is this region bound by CTCF?

Response: We sincerely appreciate your valuable question and apologize for the lack of clarity in our definition of the hEC3 enhancer as “primed.” In the manuscript, we described hEC3 as “primed for activation but it is not currently active or it may serve as a structural element to facilitate the contact between the BCE locus enhancer cluster and the *HMX1* promoter.” This description was based on the following observations:

1)PChi-C results: These results demonstrate a clear interaction between hEC3 and the *HMX1* promoter, suggesting that hEC3 may play a role in facilitating physical contact between the BCE locus enhancers and the *HMX1* promoter to facilitate the regulatory interactions required for gene expression.

2) Chromatin accessibility: While hEC3 lacks active histone modifications (e.g., H3K27ac), it

exhibits a strong ATAC-seq signal (**Fig. 2b** and **Fig. 3a**), indicating open chromatin conformation. This suggests that hEC3 may require specific transcription factors to become fully active and acquire enhancer functionality.

Additionally, we have added hEC3 LacZ staining at E14.5 stage (**Fig. 3k**), relatively stable yet weak signals were observed in the pinna region, suggesting it's an enhancer functioning in early pinna development but not in early CNCC-derived mesenchymal.

Regarding your question about CTCF binding, CTCF-ChIP data from hCNCCs (**Fig. 1c**) reveal no CTCF enrichment at hEC3, indicating that this region does not function as a CTCF-mediated structural element.

We hope this clarification addresses your concerns. If further elaboration is required, we would be happy to provide additional details. Thank you again for your thoughtful feedback, which has helped us refine the interpretation and presentation of our findings.

4. In the discussion, the absence of eye-related phenotypes in patients, in contrast to what is observed in mice, should be explored. Could this be attributed to the differences between homozygosity and heterozygosity in mouse models and human patients?

Response: Thank you for pointing out the differences in eye phenotypes between BCE patients and *mEC1* knock-in mice. We find this observation intriguing, although an in-depth exploration of these differences has not been conducted. Potential reasons for the observed discrepancies may include, but are not limited to, the following: 1) Copy number differences: BCE patients exhibit a CNV of 3, while phenotypes in mice only emerge at a CNV of 4. This aligns with the distinction between heterozygotes and homozygotes, as you have noted. 2) Variation in CNV regions: The CNV region in humans is larger compared to the *mEC1* knock-in mice, where only the segment encompassing *mEC1* is included. 3) Temporal and spatial differences in enhancer-driven gene expression: Enhancer activity may differ between humans and mice in the eye region. For instance, as demonstrated in **Fig. 3m**, the EC1+EC2+EC3 enhancers exhibit marked temporal and spatial specificity in mice, with strong signals around the eye region at E14.5. Similarly, as shown in **Fig. 5j**, *mEC1^{dup/dup}* mice display stronger WISH signals for *HMX1* expression not only in the ear region but also prominently around the eyes compared to wild-type mice. Unfortunately, due to the lack of data on the temporal and spatial activity of EC1+EC2+EC3 in human embryos, we are unable to determine the status of these enhancers in human eyes. Additionally, single-cell RNA sequencing data from human embryos (9th to 24th week) and spatial RNA-seq from mouse embryos (E16.5) reveal that *HMX1* is expressed in eye-related cell populations (**Figure R20**)^{22, 23}. Notably, *HMX1* expression occurs earlier in human embryos compared to mice (E14.5–E16.5 in mice, corresponding to the 7th–8th week in humans). This suggests that the observed species-specific differences in *HMX1* expression patterns may stem from temporal rather than spatial variations.

We hope this response addresses your concerns. Additionally, as per your suggestion, we have included speculative explanations for the human-mouse differences in eye phenotypes in the

Discussion section (lines 547-551).

Figure R20. The expression pattern of *HMX1* in human and mouse eyes. a-b: Single-cell RNA-seq analysis of human embryonic eyes from the 9th to the 24th week of development. c-d: *HMX1* is primarily expressed in retinal progenitor cells and H2 horizontal cells. e: *HMX1* expression is most prominent after the 15th week of human embryonic development. f: Spatial RNA-seq analysis of mouse embryos reveals that *Hmx1* is expressed at eye region (labeled by red triangle) at embryonic day 16.5 (E16.5), corresponding to the developmental stage of the 9th week in human embryos (other time points were not shown for no *Hmx1* expression signal).

Minor points:

L61: "Developmental biology, a discipline that has recently seen significant progress." While we agree with this statement, reference #1 (Atchley et al., 1991) may not be the most current example to demonstrate recent advancements in the field.

Response: Thank you very much for pointing out the inappropriate citation. We have re-evaluated the relevant literature and found a recent review (Liberali *et al.* 2024) that describes the contributions of new technologies and methods in developmental biology to areas such as gene regulation, pattern formation, morphogenesis, organogenesis, and stem cell biology²⁴. We believe this review provides a more appropriate reference for citation. Therefore, we have updated the citation on line 64 in the manuscript.

L118: Suggest using "GeneChip" instead of "genechip".

Response: We apologize for the inaccurate terminology used in the original manuscript. In the revised manuscript on line 135, we have corrected it to "GeneChip".

L154: "Promoter Capture Hi-C (PC-HiC)." The acronym PCHi-C is commonly used in the field, as in L158.

Response: We greatly appreciate your professional suggestion. In the revised manuscript on line 173, we have corrected the expression to "PCHi-C".

L211: "In the LacZ assays assessing enhancer activity..." This sentence could be reformulated to clarify the identity of the enhancers tested. For example: "In the LacZ assays assessing BCE enhancers' activity."

Response: Thank you for your valuable suggestion. In the revised manuscript on lines 234-235, we have reformulated the sentence for clarity as follows: "In the LacZ assays assessing BCE enhancers' activity."

In Figure 2A, the color code of CS13 to CS17 tracks should be described rather than merely referenced on a website.

Response: We sincerely apologize for not including the color code description. In the revised manuscript, we have added the relevant annotations to the legend of **Fig. 2a** on lines 1218-1219.

Figure 3A-F: The colors used to represent the three different EC enhancers, the three stages of hCNCCs sampled, and the two genotypes tested are all relatively similar and complicate the graphical understanding of Figure 3F.

Response: Thank you for pointing out the issue with the color scheme in **Fig. 3F**. We agree that The similar colors may hinder the clarity of the graphical representation. In the revised manuscript, we have adjusted the color scheme of two genotypes tested in **Fig. 3F** to improve the clarity of the figure.

Figure 3G: An "r" is missing in "CNCCs early development."

Response: Thank you for pointing out this error. In the revised manuscript, we corrected the typo in 'CNCCs early development' in **Fig. 3g** by adding the missing 'r'.

Figure 6E-F: Could the authors specify in the figure how the seven clusters from panels E-F were selected from panel D? Our understanding is that they are found E10.5, but this should be acknowledged in the main figure.

Response: Response: Thank you very much for raising this question. We selected these seven clusters because they all represent CNCC-derived fibroblast types, the primary cell populations expressing the *Hmx1* gene. Among them, C6 represents CNCC-derived mesenchymal cells,

primarily identified at E10.5, while the remaining clusters (C1, C2, C3, C4, C5, and C7) represent fibroblast types found at E10.5, E12.5, and E14.5 (mainly at E12.5 and E14.5).

To better illustrate our purpose, we manually drew a curve in Figure 6D (now **Fig. 6i**) to outline these seven cell types and added a "fibroblast" label. Additionally, we also included a "fibroblast" label in Figure 6E (now **Fig. 6j**). We have included detailed explanations in the figure legend (lines 1356-1357).

Thank you again for your question. We hope our revisions help readers easily extract the relevant information from the figures.

References

1. Long HK, Prescott SL, Wysocka J. Ever-Changing Landscapes: Transcriptional Enhancers in Development and Evolution. *Cell* **167**, 1170-1187 (2016).
2. Rosin JM, *et al.* A distal 594 bp ECR specifies Hmx1 expression in pinna and lateral facial morphogenesis and is regulated by the Hox-Pbx-Meis complex. *Development* **143**, 2582-2592 (2016).
3. Hirsch N, *et al.* Unraveling the transcriptional regulation of TWIST1 in limb development. *PLoS Genet* **14**, e1007738 (2018).
4. Sumiyama K, Tanave A. The regulatory landscape of the Dlx gene system in branchial arches: Shared characteristics among Dlx bigene clusters and evolution. *Dev Growth Differ* **62**, 355-362 (2020).
5. Osterwalder M, *et al.* HAND2 targets define a network of transcriptional regulators that compartmentalize the early limb bud mesenchyme. *Dev Cell* **31**, 345-357 (2014).
6. Cotney J, *et al.* The evolution of lineage-specific regulatory activities in the human embryonic limb. *Cell* **154**, 185-196 (2013).
7. Will AJ, *et al.* Composition and dosage of a multipartite enhancer cluster control developmental expression of Ihh (Indian hedgehog). *Nat Genet* **49**, 1539-1545 (2017).
8. Quiat D, *et al.* An ancient founder mutation located between ROBO1 and ROBO2 is responsible for increased microtia risk in Amerindigenous populations. *Proc Natl Acad Sci U S A* **119**, e2203928119 (2022).
9. Zhang YB, *et al.* Genome-wide association study identifies multiple susceptibility loci for craniofacial microsomia. *Nat Commun* **7**, 10605 (2016).
10. Xu X, *et al.* Novel risk factors for craniofacial microsomia and assessment of their utility in clinic diagnosis. *Hum Mol Genet*, (2021).
11. Allen RS, Biswas SK, Seifert AW. Neural crest cells give rise to non-myogenic mesenchymal tissue in the adult murid ear pinna. *bioRxiv*, (2023).
12. Long HK, *et al.* Loss of Extreme Long-Range Enhancers in Human Neural Crest Drives a Craniofacial Disorder. *Cell Stem Cell* **27**, 765-783 e714 (2020).
13. Koehler KR, Mikosz AM, Molosh AI, Patel D, Hashino E. Generation of inner ear sensory epithelia from pluripotent stem cells in 3D culture. *Nature* **500**, 217-221 (2013).
14. Liu XP, Koehler KR, Mikosz AM, Hashino E, Holt JR. Functional development of mechanosensitive hair cells in stem cell-derived organoids parallels native vestibular hair cells.

Nat Commun **7**, 11508 (2016).

15. Williams AL, Bohnsack BL. What's retinoic acid got to do with it? Retinoic acid regulation of the neural crest in craniofacial and ocular development. *Genesis* **57**, e23308 (2019).
16. Fukuta M, *et al.* Derivation of mesenchymal stromal cells from pluripotent stem cells through a neural crest lineage using small molecule compounds with defined media. *PLoS One* **9**, e112291 (2014).
17. Ma T, Yang B, Gillespie A, Carlson EJ, Epstein CJ, Verkman AS. Generation and phenotype of a transgenic knockout mouse lacking the mercurial-insensitive water channel aquaporin-4. *J Clin Invest* **100**, 957-962 (1997).
18. El-Brolosy MA, Stainier DYR. Genetic compensation: A phenomenon in search of mechanisms. *PLoS Genet* **13**, e1006780 (2017).
19. Chen LF, Long HK, Park M, Swigut T, Boettiger AN, Wysocka J. Structural elements promote architectural stripe formation and facilitate ultra-long-range gene regulation at a human disease locus. *Mol Cell* **83**, 1446-1461 e1446 (2023).
20. Li Y, *et al.* Combined genome-wide association study of 136 quantitative ear morphology traits in multiple populations reveal 8 novel loci. *PLoS Genet* **19**, e1010786 (2023).
21. Adhikari K, *et al.* A genome-wide association study identifies multiple loci for variation in human ear morphology. *Nat Commun* **6**, 7500 (2015).
22. Zuo Z, *et al.* Single cell dual-omic atlas of the human developing retina. *Nat Commun* **15**, 6792 (2024).
23. Chen A, *et al.* Spatiotemporal transcriptomic atlas of mouse organogenesis using DNA nanoball-patterned arrays. *Cell* **185**, 1777-1792 e1721 (2022).
24. Liberali P, Schier AF. The evolution of developmental biology through conceptual and technological revolutions. *Cell* **187**, 3461-3495 (2024).

REVIEWER COMMENTS

Reviewer #1 (Remarks to the Author):

The authors have convincingly addressed all major points raised by this reviewer and included substantial additional experimental data and analysis, leading to an overall improvement of the manuscript which I consider ready for publication in Nature Communications.

I would however recommend to still consider the following points:

- The new title should be more accurate and clear. The authors found that HMX1 is controlled by hierarchical and synergistic interactions among the clustered enhancers in the wildtype situation. The title does however not reflect that a mutation leads to the ear malformation.

Response: Thanks for your suggestion. We've revised the title to "Auricular Malformations Driven by Copy Number Variations in an Enhancer Cluster with Hierarchically Synergistic Interactions: A Dominant Enhancer Recapitulates Human Pathogenesis" to clarify that a mutation causes the ear malformation while reflecting the hierarchical and synergistic enhancer interactions in the wildtype context.

- Figure 3 (panels h-m) would still gain from clearer indication that the embryos in vertical order are the result of the same construct (at different timepoints) with indications of the timepoints on the left of the h panel. It is not immediately clear that the timepoints shown in g also account for panels h-m.

Response: Thanks for your recommendations. We've updated Figure 3 (panels h-m) by adding timepoint labels on the left of panel h, ensuring it's clear that these embryos, shown in vertical order, result from the same construct at different stages and align with the timepoints indicated in panel g.

- Line 283: The sentence should include that "...hEC1 and hEC2 likely function as synergistic enhancers..." as the result is based on transgenic reporter analysis out of endogenous context. Therefore, the actual endogenous enhancer properties could still be subject to additional modification.

Response: Thanks for your suggestion. We've revised Line 283 to state, "...hEC1 and hEC2 likely function as synergistic enhancers based on transgenic reporter analysis out of endogenous context," acknowledging that additional modifications might affect their endogenous properties.

- Line 369: The reference (Fig. 5l) should be placed after the previous sentence describing the result. The respective figure legend of panel 5l (line 1328) should mention what type of expression analysis this is.

Response: Thanks for your suggestion. We've moved the reference (Fig. 5l) to follow "...a ninefold decrease" in Line 368 for better alignment with the described result, and added a description in Line 1333 of the figure legend to specify the type of expression analysis used in panel 5l.

Reviewer #2 (Remarks to the Author):

First, I must thank the authors for addressing each point of my critique thoroughly with additional figures that have been made using pre-existing data, and by conducting further experiments. These revisions add to the monumental amount of evidence already presented prior and strengthen the authors' manuscript.

Second, the manuscript's narrative is clearer, especially in the presentation of the ssRNA-seq data. Specifically, adding ssRNA-seq data from mEC1^{dup/dup} strengthens the authors' spatiotemporal determination of mEC1's effects in fibroblast cells. I also appreciate the authors' work on performing RetA titration and gene expression analysis of marker genes to demonstrate the posteriorizing of CNCC to a PA2/PA3 locale. Their presentation of this data is well rationalized, and I commend the authors for the inclusion of these experiments in the revised manuscript.

Finally, the authors have satisfactorily addressed each of my critique points and I would recommend the publication of this research article in Nature Communications. In my response to their rebuttal, I have raised a few additional questions. The authors are not compelled to address these in the manuscript, but I would be delighted to see them do so.

Major Comment #1 response to rebuttal:

I am pleased that the authors acknowledge the other enhancers outside the BCE core locus. I agree with the authors that these enhancers are indeed weak based on the chromatin tracks presented. These may become active in some unknown developmental context that is beyond the scope of the study. Though not necessary for further revision, it would be worthwhile in later studies to test these elements in luciferase assay conducted in late CNCCs and PA2-line CNCCs. Noteworthy, in future luciferase and lacZ experiments, it may be worthwhile replacing the LUC or LacZ reporter's promoter, which may be minSV40 or HSP90 promoters, with the *HMX1* basal promoter to determine if there is improved reporter response due to enhancer-promoter compatibility.

Response: Thank you for your valuable suggestions. We agree that testing the activities of these enhancers in luciferase assays conducted in late CNCCs and PA2-line CNCCs would be worthwhile for future studies. Additionally, replacing the LUC or LacZ reporter's promoter with the *HMX1* basal promoter to improve enhancer-promoter compatibility is an excellent idea. While we plan to explore the use of the *HMX1* native promoter for LUC in future experiments, we currently employ the next-generation enSERT system for LacZ, which utilizes the shh promoter¹. This system has demonstrated stability and efficiency, and there are no reports suggesting replacement with a native promoter. Hence, we do not have immediate plans to modify the LacZ system. Once again, we appreciate your rigorous scientific approach, which has helped strengthen our research.

Major Comment #2 response to rebuttal:

In their rebuttal, the authors demonstrate that *HMX1* is not expressed in *in vitro* derived

cranial chondrocytes, nor do hEC1/2/3 elements show accessibility in *in vitro* chondrocyte ATAC-seq data. This is concordant with the authors finding in their ssRNA-seq data that “the *HMX1* gene is primarily expressed in fibroblasts, and the duplication of mEC1 leads to a significant expansion of *HMX1* expression in these cells (Fig. 6c)” (line 384-386). Yet, the authors say in lines 409-411 that “differential gene analysis revealed significant downregulation of genes such as *Msx1*, *Fgf18*, and *S100a4* in the chondrocyte subpopulation” (see Figure 6e). However, Figure 6c shows very little *HMX1* expression in chondrocytes (cluster 9). If there are *HMX1*-expressed cells in cluster 9, then they must be rare because I can only see a few red points. Therefore, I am confused by the discrepancy between the paucity of *HMX1* expression in chondrocytes and the authors’ finding of DEG in chondrocytes. Is this due to organismal and/or *in vitro* vs *in vivo* differences? That is, the CNCC *in vitro* data is human and the ssRNA-seq *in vivo* data is mouse? If there is a tiny subpopulation of chondrocytes in Figure 6c that is driving the DEGs shown in Figure 6e, I find it exciting to know what these cells are.

Response: We sincerely appreciate your insightful comments. In the mouse single-cell sequencing data, *Hmx1* is predominantly expressed in fibroblasts, with only a subset of chondrocytes showing detectable levels, as indicated in Fig. 6c and further validated by GFP lineage-tracing single-cell sequencing (Fig. 6i, cluster 13). We also observed that mEC1 duplication moderately expands *Hmx1*-expressing chondrocytes (Fig. 6c), supporting the plausibility of DEGs in the chondrocyte cluster. Another plausible explanation involves paracrine signaling: the expansion of *Hmx1* expression in fibroblasts might indirectly influence chondrocyte development via non-cell-autonomous mechanisms. Regarding the absence of *HMX1* expression in *in vitro*-derived cranial chondrocytes, we hypothesize that these chondrocytes differ from the *HMX1*-expressing auricular chondrocytes. The elastic cartilage of the external ear represents a specialized subtype of cartilage, requiring distinct signaling programs for neural crest cells to differentiate into this lineage. Thank you again for your thought-provoking questions.

Major comment #3 response to rebuttal:

First, I agree with the authors that the *in vitro* CNCC model is not a drop-in replacement for the *in vivo* mouse system. However, the readership can benefit from an explanation of why 100uM RetA was chosen in this protocol. The RT-qPCR results showing regionalized gene expression of marker genes are crucial, and I thank the authors for this addition.

Second, the authors say “In fact, based on our WISH results and transgenic GFP mouse model, *HMX1* expression domain extends partially from PA2 into the region that develops from PA3”. I examined the updated Figure 3g, Figures 5i,j, and Extended Figure 5f but it is not obvious where *HMX1*'s expression expands into the PA3. The only figure that shows this expanded expression to PA3 is in the scRNA-seq analysis presented in Extended Figure 2A, however, showing this in an animal image is preferred. Please note this expanded PA3 expression by highlighting the expression domain in the WISH or *HMX1*-GFP reporter animals presented in the main or supplementary figures. It could be worthwhile to show a zoomed-in version of the mouse if the PA3 region is unclear from the whole animal image.

Response: We thank the reviewer for their thoughtful feedback. Regarding the choice of 100

nM RetA, we appreciate your suggestion that providing an explanation is necessary, as it allows the readership to understand that the selection of 100 nM is supported by experimental evidence, which we clarified in detail in our initial rebuttal letter. We believe this explanation sufficiently addresses the concern and highlights the justification for its use in our protocol. Regarding *Hmx1* expression extending into PA3, we acknowledge that our original description in the rebuttal response may have been imprecise. The interpretation was based on *Hmx1* expression patterns at E11.5, particularly in the region outlined by the rectangular box in Extended Data Figure 5f. This area lies inferior to PA2, leading to our description that "the *Hmx1* expression domain extends partially from PA2 into the region that develops from PA3." We have included this figure with detailed anatomical annotations to clarify the relevant structures.

Extended Data Figure 5f. Comparison of the spatiotemporal pattern of *Hmx1* expression between the Hmx1-P2A-EGFP transgenic reporter line and WISH results at E11.5. The left panel shows an image of Hmx1-P2A-EGFP; the right panel shows an image of WISH result.

Major comment #4 response to rebuttal:

I appreciate the authors' specification of the n=3 BCE copy number in their affected cohort and for noting in their rebuttal that they are indeed testing 0 to 4 copies of the mEC1 enhancer in their transgenic animals. This information could also be helpful for the reader, and should be written in the results section titled "Impact of spatial *HMX1* expression on pinna development: insights from transgenic mouse models mimicking human BCE anomalies." I was puzzled why 3 copies did not show ear defects, and the authors should note in the discussion section about their hypothesis. That is, "this may be due to the insensitivity of mice to HMX1 dosage and the activation of genetic compensation mechanisms, a phenomenon commonly observed in mouse models of human diseases^{17,18}".

Response: Thanks for your comment. Currently, we have added this information in the revised manuscript (Line 375-379). Thank you for your comment. We have incorporated the information about the n=3 BCE copy number in the affected cohort (lines 140-141) and the testing of 0 to 4 copies of the mEC1 enhancer in our transgenic animals into the revised manuscript (Lines 362-363 and lines 365-366). Regarding the lack of ear defects in mice with 3 copies, we hypothesize that this could be due to the insensitivity of mice to HMX1 dosage or the activation of genetic compensation mechanisms, a phenomenon commonly observed in mouse models of human diseases^{17,18}. This hypothesis has been added to the discussion

section for the readers' benefit (lines 553-557).

Major Comment #5 response to rebuttal

I agree with the authors that EC2 and EC3 need to be explored more in *in vivo* experiments, as noted in the discussion, but the authors have adequately shown that it is the EC1 element that contributes the most to BCE malformation.

The authors write "while hEC3 shows weak activity in both LUC and LacZ experiments independently, it significantly enhances the gene-driving capability when combined with hEC1+hEC2, similar to enhancers described in the Molecular Cell paper." As noted in comment #1, in future luciferase lacZ experiments, it may be worthwhile replacing the LUC or LaxZ reporter's promoter with the *HMX1* promoter to see if EC3 does show activity due to promoter compatibility. In light of the absence of CTCF motifs and chip-seq peaks, it is exciting to understand how EC3 functions as a structural element or if it could indeed be a facilitator.

Response: Thank you for your suggestion. We agree that further exploration of EC2 and EC3 in *in vivo* experiments is necessary, as noted in the discussion. We also acknowledge the potential importance of enhancer-promoter compatibility and will consider using the *HMX1* promoter in future luciferase and LacZ experiments to evaluate EC3 activity. To further investigate EC3's function, we plan to knock out EC3 in cell lines and conduct a more comprehensive three-dimensional structural analysis. Additionally, as the LacZ results indicate that EC3 activity is primarily observed during ear development (Figure 3k, lp region at E14.5), specialized cell lines may be required for *in vitro* experiments. However, these efforts pose significant challenges and will likely require collaboration with other researchers in the field of external ear malformations.

Additional Minor comments:

Figure 3 h-i: "The orange number in the bottom right corner of each embryo indicates the count of positive LacZ staining embryos". Does this number include only LacZ-positive embryos, likely measured from PCR, or does it also include animals that show the same expression domain as the representative animal shown? Neither the methods section nor the legend makes this clear. For reproducibility, an expression domain should be consistent between at least 2 LacZ positive embryos.

Response: Figure 3 h-i: The orange number in the bottom right corner of each embryo represents the number of embryos showing a similar expression domain as the representative animal. The black number indicates the total number of LacZ-positive embryos. All patterns have been replicated in at least two embryos. This clarification has been added to the corresponding legends (Line 1263-1265).

Text is difficult to read in Figure 6D. I would recommend labeling only a few genes.

Response: We have updated Figure 6D to label only key genes for improved readability.

Superscript text is difficult to read in Figure 5M. I would recommend making the font larger.

Response: The font size of the superscript text in Figure 5m has been increased for better legibility.

Reviewer #3 (Remarks to the Author):

The authors have made significant improvements to the original manuscript addressing all of my original concerns. This publication is expected to be well received by a general audience.

Response: Thank you for your acknowledgment.

Reviewer #4 (Remarks to the Author):

The authors have satisfactorily addressed all my points.

I would, however, ask them to provide more detail in the legend of Figure 2B regarding the promoter capture Hi-C tracks. Specifically, they should clarify the meaning of the full and dotted lines, the points, and the color of the different points. This is particularly important as the current display of the data is not intuitive.

Response: Thanks for your suggestions. We have added the description of full and dotted lines, and also the meaning of different color points in the legend in the revised manuscript (Line 1232-1234).

- 1 Kvon, E. Z. *et al.* Comprehensive In Vivo Interrogation Reveals Phenotypic Impact of Human Enhancer Variants. *Cell* **180**, 1262-1271 e1215 (2020). <https://doi.org/10.1016/j.cell.2020.02.031>